# Testing the performance of field calibration techniques for low-cost gas sensors in new deployment locations: across a county line and across Colorado

Joanna Gordon Casey[1], Michael P. Hannigan[1]

[1]Department of Mechanical Engineering, University of Colorado at Boulder, Boulder, 80309, United States of America

*Correspondence to*: Joanna Gordon Casey (joanna.casey@colorado.edu)

**Abstract.** We assessed the performance of ambient ozone ($O_3$) and carbon dioxide ($CO_2$) sensor field calibration techniques when they were generated using data from one location and then applied to data

collected at a new location. This was motivated by a previous study (Casey et al., 2017) which highlighted the importance of determining the extent to which field calibration regression models could be aided by relationships among atmospheric trace gases at a given training location, which may not hold if a model is applied to data collected in a new location. We also explored the sensitivity of these methods in response to the timing of field calibrations relative to deployments periods. Employing data

from a number of field deployments in Colorado and New Mexico that spanned several years, we tested and compared the performance of field-calibrated sensors using both linear models (LMs) and artificial neural networks (ANNs) for regression. Sampling sites covered urban, rural/peri-urban, and oil and gas production influenced environments. We found that the best performing model inputs and model type depended on circumstances associated with individual case studies, such as differing characteristics of

local dominant emissions sources, relative timing of model training and application, and the extent of extrapolation outside of parameter space encompassed by model training. In agreement with findings from our previous study that was focused on data from a single location (Casey et al., 2017), ANNs remained more effective than LMs for a number of these case studies but there were some exceptions. For $CO_2$ models, exceptions included, case studies in which training data collection took place more

than several months subsequent to the test data period. For $O_3$ models, exceptions included case studies in which the characteristics of dominant local emissions sources (oil and gas vs. urban) were

significantly different at model training and testing locations. Among models that were tailored to case studies on an individual basis, $O_3$ ANNs performed better than $O_3$ LMs in 6 out of 7 case studies, while $CO_2$ ANNs performed better than $CO_2$ LMs in 3 out of 5 case studies. The performance of $O_3$ models tended to be more sensitive to deployment location than to extrapolation in time while the performance

of $CO_2$ models tended to be more sensitive to extrapolation in time than to deployment location. The performance of $O_3$ ANN models benefited from the inclusion of several secondary metal oxide type sensors as inputs in 5 of 7 case studies.

## 1    Introduction

In places like the Denver Julesburg (DJ) and San Juan (SJ) Basins, along Colorado's Front Range and in

the Four Corners Region, oil and gas production activities have been increasing with the advent of horizontal drilling that can be effectively used in conjunction with hydraulic fracturing to produce hydrocarbons from unconventional geologic formations. Public health concerns have arisen about the increasing number of people living alongside these industrial activities and emissions (Adgate et al., 2014; Mckenzie et al., 2014; McKenzie et al., 2012, 2017). We previously developed methods to

quantify ozone ($O_3$), carbon dioxide ($CO_2$), methane ($CH_4$), and carbon monoxide (CO) using low-cost gas sensors in an area where the ambient mole fractions of these species are influenced by oil and gas production activities (Casey et al., 2017). Such low-cost sensor measurements could enable greater understanding of air quality in oil and gas production basins, informing the spatial and temporal scales that people live and work in a way that current technologies used by regulatory agencies cannot feasibly

accomplish. In our previous work, we tested and compared the performance of direct and inverted linear models (LMs) as well as artificial neural networks (ANNs) as regression tools in the field calibration of low-cost sensor arrays to quantify these target gas species using month-long test datasets, training each model with approximately one month of data prior to and one month of data subsequent to this test period. ANNs are powerful pattern recognition tools. They were found to perform better than

both inverted and direct LMs in our previous study, but concerns arose when findings suggested that the performance of ANNs was being augmented by the relationships among gas mole fractions in the atmosphere at a given location. Low-cost gas sensor systems have the potential to inform spatial and

temporal variability of pollution. Calibration equations for each sensor system can be generated in one location based on co-located measurements with reference instruments, and then the sensor systems can be moved into a spatially distributed network. Since the relationships among gas mole fractions will differ at different sampling sites across a spatially distributed network, calibration models may not hold

at new sampling sites. In this work, we test calibration model performance when extended to new locations.

## 1.1   Low-Cost Sensors for Air Quality Measurements

The use of low-cost metal oxide, electrochemical, and non-dispersive infrared sensors to characterize air quality is becoming increasingly common across the globe (Clements et al., 2017; Kumar et al., 2015).

While low-cost sensors have been emerging on the market with sufficient sensitivity to resolve variations in ambient mole fractions of target gases of interest, they are also sensitive to temperature and humidity variations that occur in the ambient environment. NDIR sensors, like the ELT s300 $CO_2$ sensor employed in this study, have good selectivity, but, since pressure and temperature are not controlled in the optical cavity of ELT s300 $CO_2$ sensors, the influence of temperature on sensor signals

plays an important role. The influence of humidity is also important to address because changes in water vapor are known to influence NDIR measurements of $CO_2$ in terms of spectral cross-sensitivity due to absorption band broadening (Licor, 2010).

Both metal oxide and electrochemical type sensors operate on the principle of oxidizing or reducing

reactions at sensor surfaces. For electrochemical sensors, like the Alphasense CO-B4 sensor employed in this study, oxidizing or reducing compounds react at the working electrode, resulting in the transfer of ions across an electrolyte solution from the working electrode to the counter electrode, balanced by the flow of electrons across the circuit connecting the working electrode to the counter electrode. A linear relationship is expected between this current and the target gas mole fraction. Electrochemical

sensors can be tuned to respond more or less strongly to specific gases by adjusting the material properties of the working electrode. A membrane is located between the working electrode and the exterior of the sensor in order to control redox reaction rates. The rates at which gases diffuse through

the membrane to reach the working electrode and the electron transfer rates have been shown to increase at higher temperatures (Xiong and Compton, 2014), and since chemical reaction rates are also influenced by temperature, electrochemical sensor responses can be influenced by sensor operating temperature.  Changes in ambient humidity levels can cause sensors to lose or gain of the electrolyte

solution, by mass, also influencing electrochemical sensor response (Xiong and Compton, 2014).

For metal oxide sensors, and to a lesser extent for electrochemical sensors, resolving the response of a sensor attributable to the target gas species can also pose a challenge in the presence of interfering gas species.  Metal oxide sensors, like those used in this study, have a resistive heater circuit that warms up

the sensor surface, causing $O_2$ molecules to adsorb to the sensor surface, which leads to increased resistance across the surface of the sensor.  In the presence of an oxidizing compound, like $O_3$, more oxygen molecules are adsorbed to the sensor surface and the resistance across the sensor surface is increased further.  In the presence of a reducing compound, like CO, oxygen molecules are removed from the sensor surface, allowing electrons to flow more freely, resulting in decreased resistance across

the sensor surface. For metal oxide sensors, the resistance across the sensor surface can then be used to determine the mole fraction of a given oxidizing or reducing compound, often according to a nonlinear relationship.  Exposure to humidity has been shown to significantly lower the sensitivity of metal oxide gas sensors making it an important parameter to address in a gas quantification model (Wang et al., 2010).  Metal oxide sensor operating temperature has also been shown to strongly influence sensor

sensitivity and selectivity to different gas species (Wang et al., 2010).  Metal oxide type sensors can be tuned to respond differently from one another to oxidizing and reducing gas species by using different metal oxide materials and doping agents for the sensor surface, but selectivity is difficult to achieve.

## 1.2   Low-Cost Air Quality Sensor Quantification

Because low-cost gas sensor signals are influenced, sometimes significantly, by interfering gas species

and changing weather conditions in the ambient environment, field normalization methods to quantify atmospheric trace gases using low-cost sensors have been found to be more effective than lab calibration (Cross et al., 2017; Piedrahita et al., 2014; Sun et al., 2016).  Our previous study and several

others have compared the performance of field calibration models generated using LMs (simple and multiple linear regression) relative to supervised learning methods (including ANNs and random forests), all finding that ANNs (Casey et al., 2017; Spinelle et al., 2015, 2017) and random forests (Zimmerman et al., 2017) outperformed LMs in the ambient field calibration of low-cost sensors. Like

earlier laboratory based studies (Brudzewski, 1999; Gulbag and Temurtas, 2006; Huyberechts and Szeco, 1997; Martín et al., 2001; Niebling, 1994; Niebling and Schlachter, 1995; Penza and Cassano, 2003; Reza Nadafi et al., 2010; Srivastava, 2003; Sundgren et al., 1991), ANN-based calibration models, incorporating signals from an array of gas sensors with overlapping sensitivity as inputs, have been able to effectively compensate for the influence of interfering gas species and resolve the target

gas mole fraction.

ANNs are known to be able to very effectively represent complex, nonlinear, and collinear relationships among input and output variables in a system (Larasati et al., 2011). ANNs are useful in the field calibration of low-cost sensors because, through pattern recognition of a training dataset, they are able

to effectively represent the complex processes and relationships among sensors and the ambient environment that would be very challenging to represent analytically or based on empirical representation of individual driving relationships. In practice though, the reason multiple gas sensors are able to improve the performance of calibration models may be in part the result of correlation between mole fractions of target gases themselves that hold for one model training location, but might

not remain effective at alternative sampling sites or during other time periods.

## 1.3    Summary of Previous Study

Our previous study was carried out using sensor measurements collected over the course of several months in the spring of 2017, in Greeley, Colorado, which lies within the Denver Julesburg oil and gas production basin. Others had recently demonstrated the utility of machine learning methods in the

25 quantification of atmospheric trace gases using arrays of low-cost sensors in urban (De Vito et al., 2008, 2009; Zimmerman et al., 2017) and rural (Spinelle et al., 2015, 2017) areas. Our previous study tested the relative performance of machine learning methods and LMs in the quantification of $CH_4$, $O_3$, $CO_2$,

and CO in an area strongly influenced by oil and gas production activities, where enhanced levels of hydrocarbons and other industry related pollutants could potentially confound measurements. The previous study was also the first to compare machine learning regression techniques with LMs toward the quantification of $CH_4$ using arrays of low-cost sensors in any setting. The study tested and compared calibration models using data from two U-Pod sensor systems containing arrays of low-cost gas sensors; these systems were co-located with optical gas analysers at a Colorado Department of Public Health and Environment monitoring site. ANNs and LMs were trained using a variety of sensor signal input sets from a month of co-located data collected prior to and following a month long test period. The performance of each model was then evaluated relative to reference instrument measurements during the test period. For quantification of all four trace gases that we tested in this oil and gas-influenced setting, we found that ANNs performed better than LMs. The better performance of ANNs over LMs was likely largely attributable to the ability of ANNs to more effectively represent complex and nonlinear relationships among sensor responses, environmental variables, and trace gas mole fractions than LMs. However, the performance of these powerful regression methods could be aided by relationships among atmospheric trace gases specific to the training location, which would not necessarily hold at different sampling sites.

## 1.4 Spatially Distributed Networks of Sensors and Spatial Extension of Calibration Models

Distributed spatial networks of low-cost sensor systems have the potential to inform air quality with high spatial and temporal resolution. As such, studies have begun to deploy spatial networks of low-cost sensor systems. These, studies rely on the spatial transferability of quantification techniques. In the present work, we test model performance under conditions of spatial transferability, wherein a model is trained using data from one location then applied to a test dataset using data from a new location. In testing spatial extension of a model, we investigate how well the field calibration of low-cost sensors can inform target gas mole fractions when sensors are deployed in a new location and a new microenvironment of oxidizing and reducing compounds. We also test model performance under conditions of temporal extension, wherein a model is trained using data that was collected only prior or subsequent to the model application period. In testing temporal extension of models, we investigate

how model performance is influenced by sensor drift over time. We opportunistically utilize measurements collected with low-cost sensors in Denver, Boulder County, and the DJ and SJ oil and gas production basins in recent years. This effort focuses on the analysis for $O_3$ and $CO_2$ using both LMs and ANNs, including a comparison of models with a number of different input sets. In previous
work (Casey et al., 2017), we have additionally addressed the quantification of CO and $CH_4$ using arrays of low-cost sensors together with field normalization methods, but these species are not included in the present analysis because analogous reference data to those we present for $O_3$ and $CO_2$, were not available CO and $CH_4$.

**1.5   Oil and Gas Production and Air Quality**

Oil and gas production related emissions, namely nitrogen oxides ($NO_X$) and volatile organic compounds (VOCs), have been shown to influence tropospheric ozone ($O_3$), which is particularly relevant in regions that are in non-attainment of the United States Environmental Protection Agency (USEPA) National Ambient Air Quality Standards (NAAQS) for ozone, like the Colorado Front Range
where the DJ Basin is situated. $NO_X$ and VOC emissions, including those from oil and gas production activities, react in the atmosphere in the presence of sunlight to form tropospheric $O_3$. A number of studies have demonstrated that oil and gas related emissions contribute to increased $O_3$ in the DJ Basin (Cheadle et al., 2017; Gilman et al., 2013; McDuffie et al., 2016). Mole fractions of ozone as high as 140 ppb and 117 ppb during winter months have also been observed and attributed directly to oil and
gas production emissions in the Upper Green River Basin of Wyoming and Utah's Uinta Basin, respectively (Ahmadov et al., 2015; Edwards et al., 2013, 2014; Field et al., 2015; Oltmans et al., 2016; Schnell et al., 2009). Additionally, a modeling study concluded that oil and gas production activities could significantly impact ozone near emissions sources, beginning 2 and 8 km downwind of compressor engine and flaring activities, respectively (Olaguer, 2012).

Emissions of industry related air pollutants, including $O_3$ precursors, $NO_X$ and VOCs, are expected to occur on spatially distributed scales, across components on well pads, transmission lines, transportation

routes, and gathering stations that are each distributed throughout production basins (Litovitz et al. 2013; Mitchell et al. 2015; Allen et al. 2013). Spatially distributed networks of low-cost sensors have the potential to better inform spatial variability of air quality than existing regulatory air quality monitoring stations which cannot feasibly cover such spatially resolved measurements continuously,

and may not be representative of air quality across smaller spatial scales (Bart et al., 2014; Jiao et al., 2016; Moltchanov et al., 2015). Abeleira and Farmer show that ozone production throughout much of the Front Range, outside of downtown Denver, is likely to be $NO_X$ limited implying that local $NO_X$ sources are likely influencing ozone on small spatial scales (Abeleira and Farmer, 2017). Oil and gas industry related $NO_X$ sources, such as diesel truck traffic, flaring, and compressor engines, could lead to

pockets of elevated $O_3$ throughout the DJ Basin. While emissions from truck traffic (and in some cases a generator to power a drill rig), at a given well pad are expected to be highest during the drilling, stimulation, and completion phases, industry truck traffic often persists as the contents of produced water and condensate tanks are frequently collected from well pads throughout the production phase, as do emissions from flaring and compressor engines. Low-cost $O_3$ sensors could augment the few and far

apart regulatory sites that currently monitor $O_3$ levels in places like the DJ Basin, which has better coverage than many other production basins in the United States. While elevated ambient $CO_2$ levels are not directly harmful to human health, continuous $CO_2$ measurement can provide information about nearby combustion-related pollution and atmospheric dynamics that lead to the accumulation of potentially harmful compounds associated with the oil and gas production industry during periods of

atmospheric stability.

In this work, we present and compare models designed to address the unique challenges that come with using low-cost sensors in the quantification of atmospheric trace gases of interest in oil and gas production basins, where ambient hydrocarbon mole fractions are potentially elevated, exerting

uniquely cofounding influence on low-cost gas sensors. Calibration models that were found to perform best in our previous study are applied to data collected in different locations. For the first time, we investigate how well models can be transferred from one microenvironment to another, with different dominant local emissions source characteristics, and different relative abundance of oxidizing and

reducing compounds. Microenvironments explored in this work include a basin where both natural gas and heavier hydrocarbons are produced (the DJ Basin), and a basin where prominently natural gas is produced (the SJ Basin), with much smaller proportional emissions of heavier hydrocarbons, and likely lower atmospheric concentrations of alkanes, alkenes, and aromatics. Within and bordering the DJ

Basin, additional microenvironments include an urban location, with significant mobile sources emissions ($NO_X$, CO, and VOCs), and a peri-urban site with fewer mobile emissions and closer proximity to oil and gas production activities. We explore how robust model performance is when a model is trained in one microenvironment and transferred to another; challenged by different relative abundance of oxidizing and reducing gas species. Additionally, we test how well models can represent

and address sensor stability over time and the potential for drift.

## 2    Methods

### 2.1    Sensors and U-pods

All U-Pod sensor systems (mobilesensingtechnology.com) employed in the case studies, described below, were populated with seven low-cost gas sensors, as in our previous study (Casey et al., 2017).

The gas sensors are listed in Table 1 along with their target gas and the model input codes we assigned to each. A RHT03 sensor was used in each U-Pod to measure temperature (temp) and relative humidity (rh). A Bosch BMP085 sensor was used to measure pressure in each U-Pod.

### 2.2    Case Studies

Five to ten U-Pods were deployed at sampling sites in and around the DJ and SJ Basins from 2014 -

2017. Deployments generally consisted of co-location with reference measurements prior to and following approximately one-month periods of spatially distributed measurements. During some of the distributed measurement periods, a subset of U-Pods remained co-located with reference instruments where the field calibrations took place. As well, during some distributed measurement periods, some U-Pods were deployed in new locations that were equipped with reference measurements. In between

periods of distributed sensor system deployments, sensor systems were co-located with reference instruments for as long as possible, as logistics, and coordination with other regulatory agencies and

researchers would allow.  In this way, we hoped to maximize our ability to encompass full ranges of temperature, humidity, and trace gases that occur across seasons, in order to minimize extrapolation with respect to these parameters when models were applied to measurements from distributed deployment periods.  The locations where all or a subset of U-Pods were co-located with reference instruments are indicated in Fig. 1.  In this exploratory study, we opportunistically employ data from these sensor deployments, treating them as case studies in order to characterize the performance of field calibration models when they are extended to new locations. For each case study, described below, data was divided into training and test periods.  Timelines for these dataset pairs are detailed in Fig. 2.  Some U-Pods employed in these case studies (indicated in grey font in Fig. 2) were constructed, populated with sensors, and deployed at field sites in the spring of 2014, approximately a year before the rest of the U-Pods were constructed, populated with sensors, and deployed at field sites in the spring of 2015. The relative age of sensor systems included in some case study comparisons could have contributed to some discrepancy in model performance, though systematic differences based on U-Pod age is not apparent.

As available data from each case study allowed, we used approximately one month of training data before and after a given test period. When training data was not available within several months of a test period, significantly longer training datasets were used in order to attempt capture and effectively represent trends in sensor drift over time, as well as to avoid extrapolation of model parameters (particularly temperature) during the test data period.  As a result, model-training durations varied across case studies and sometimes significantly exceeded model-testing durations.  Each case study is similar in representing approximately one month-long deployment of sensor systems.  This study design serves a primary goal of this work, supporting the quantification of atmospheric trace gases from low-cost gas sensor data in new locations, relative to model training locations, for periods of approximately one month at a time.

Making quantitative measurements of atmospheric trace gases with low-cost sensors is challenged by unique variations in individual sensor responses associated with variations in the manufacturing

process, sensor age, and sensor exposure history. For these reasons, we generated unique calibration models using data from sensors in each individual U-Pod sensor system. The closest available data prior and/or subsequent to a test data period was used for model training to avoid complications associated with significant sensor drift and degradation in sensor sensitivity to target gas species over

time. Table 2 lists the $O_3$ and $CO_2$ reference instruments that were co-located with U-Pods at each sampling site, along with instrument operators, calibration procedures, and reference data time resolution. The selected case studies, described in sections 2.2.1 through 2.2.7 below, were aimed to support methods to quantify atmospheric trace gases during the distributed deployments we carried out from 2014 through 2017 as well as future distributed sensor network measurements. Fig. 1 shows

sampling site locations in context with urban areas and oil and gas production wells. Fig. 2 shows the timeline of each of these deployments, highlighting the training and testing periods defined for both $O_3$ and $CO_2$.

### 2.2.1   Dawson Summer 2014

The first distributed measurement campaign took place during the summer of 2014 when five U-Pods

were sited at locations around Boulder County, with four distributed along the eastern boundary of the county, adjacent to Weld County where dense oil and gas production activities were underway. A background site, further from oil and gas production activities was also included to the west, near a busy traffic intersection on the north end of the City of Boulder. Co-locations with reference measurements that were used for field calibration of the sensors took place at the Continuous Ambient Monitoring

Program (CAMP) Colorado Department of Health and Environment (CDPHE) air quality monitoring site in downtown Denver. One of the distributed sampling sites, Dawson School, was also equipped with a Thermo Electron 49 $O_3$ reference instrument operated by Detlev Helmig's research group from the Institute for Artic and Alpine Research (INSTAAR). In this work, a case study is developed using data from one U-Pod located at the CAMP site in downtown Denver for $O_3$ model training, and data

from one U-Pod, located at the Dawson School for $O_3$ model testing. This case study is used to test model performance when extrapolated in terms of $O_3$ mole fractions and applied in a new location, transferred from an urban to a peri-urban environment.

### 2.2.2  SJ Basin Spring 2015

In the spring of 2015, we augmented our original fleet of five U-Pods (BA, BB, BD, BE, and BF) with five more (BC, BG, BH, BI, and BJ) and deployed these sensor systems in the SJ Basin while a targeted field campaign was underway to understand more about a $CH_4$ 'hot spot' that was discovered from
satellite based remote sensing measurements (Frankenberg et al., 2016; Kort et al., 2014). The primary goal of this sensor deployment was to inform spatial and temporal patterns in atmospheric trace gases like $CH_4$, $O_3$, CO, and $CO_2$ across the SJ Basin. Most U-Pods were located at existing air quality monitoring sites operated by the New Mexico Air Quality Bureau (NM AQB), the Southern Ute Indian Tribe Air Quality Program (SUIT AQP), and the Navajo Environmental Protection Agency (NEPA),
which supported validation of sensor measurements for $O_3$. After this deployment period, all U-Pods were moved to the BAO site in the DJ Basin for approximately one month, and were co-located with reference instruments there that were operated by National Oceanic and Atmospheric Administration (NOAA) researchers. A case study is developed with data from the BAO site to train $O_3$ models for four U-Pods, and data from SJ Basin sites to test $O_3$ models for four U-Pods. This case study is used to
test model performance when extrapolated in temperature and time, and applied in a new location, extended from one oil and gas production basin to another across Colorado.

### 2.2.3  SJ Basin Summer 2015

In the summer of 2015, after an approximately month-long co-location with reference instruments at the BAO site, seven U-Pods were deployed again at existing regulatory monitoring sites for approximately
one month, after which they were moved back to the BAO site for another month of co-location with reference instruments there. We equipped two of the regulatory monitoring sites in the SJ Basin with LI-COR LI-840A $CO_2$ analysers to provide reference measurements for $CO_2$. A case study is developed with data from the BAO site, pre and post of the SJ Basin summer 2015 deployment to train models, and data from SJ Basin sites during the summer deployment period to test models. Data from
seven U-Pods were used to train and test $O_3$ models and data from two U-Pods were used to train and test $CO_2$ models. This case study is used to test model performance when training took place both pre

and post of the test period, and when extended to a new location, from one oil and gas production basin to another across Colorado.

### 2.2.4  BAO Summer 2015

During the SJ Basin Summer 2015 deployment period, two U-Pods remained at the BAO site. A case study is developed using data from those two U-Pods that remained at the BAO site. This case study is used to test model performance when training took place both pre and post of the test period, and when the model was tested on data that was collected in the same location as model training.

### 2.2.5  BAO Summer 2016

U-Pods were deployed at the BAO site again in 2016 for several months during the summer. In August of 2016 the U-Pods were moved to the Greeley Tower (GRET) CDPHE air quality monitoring site in Greeley, Colorado, a location which, like the BAO site, is also strongly influenced by DJ Basin oil and gas production activities. The U-Pods remained there for one year. For the GRET co-location period, CDPHE shared reference measurements for $O_3$. Additionally, Jeffrey Collett and Katherine Benedict of Colorado State University (CSU) shared $CO_2$ reference measurements from an instrument they operated at the site before October $1^{st}$ in 2016 and after March $7^{th}$ in 2017, when the instrument was located at the GRET site. A case study is developed using data from two U-Pods. Data from the yearlong deployment at the GRET site was used to train models for $O_3$, and data from the BAO site during the summer 2016 deployment was used test models for $O_3$. Because reference data for $CO_2$ was not available at the GRET site during winter months, only eight months of data from these two U-Pods during the GRET deployment was used to train models for $CO_2$, but again, data from BAO summer 2016 deployment was used to test models for $CO_2$. A significantly longer training duration is implemented in this case study because the training period took place more than several months after the model testing period. We reasoned that a longer training duration would be better able to represent patterns in sensor drift over time, as well as encompass the temperature range of test dataset period. Significantly less training time is needed when training occurs directly pre and/or post of a given model application period. This case study is used to test model performance when extrapolated significantly

(more than several months) in time and extended to a new location, from one location in the DJ basin to another.

### 2.2.6 GRET Fall 2016

In order to test model performance, under similar circumstances in terms of relative model training and
5   testing durations and timing to the BAO Summer 2016 case study, but with no extension of models to a new location, we developed another case study. This time, models for $O_3$ and $CO_2$ were trained using data from two U-Pods at GRET over the course of eight months and models for $O_3$ and $CO_2$ were tested using data from two U-Pods at GRET over the course of approximately a month in the fall of 2016. This case study is used to test model performance when extrapolated significantly (more than several
10   months) in time and applied in the same location as training took place.

### 2.2.7 GRET Spring 2017

We include findings from our previous work as a case study in order to provide context. Models for $CO_2$ and $O_3$ were tested using data from two U-Pods collected over the course of approximately one month at the GRET site in the spring of 2017. Data from two U-Pods during approximately month-long
periods pre and post of the test period were used to train $O_3$ and $CO_2$ models. This case study provides another example of model performance when training took place both pre and post of the test period, and testing took place in the same location as training.

### 2.3 Reference and Sensor Data Preparation

Each of the U-Pod sensor signals was logged to an onboard micro SD card. For metal oxide type
sensors, voltage signals were converted into resistance, and then normalized by the resistance of the sensor in clean air, $R_0$. A single value for $R_0$ was used for each sensor across the study duration. This $R_0$ value was taken as the resistance of each sensor during the GRET Spring 2017 field deployment period, when the target pollutant had approached background levels (at night for the metal oxide $O_3$ sensors and midday for all other metal oxide sensors), and when the ambient temperature was approximately 20° C
and relative humidity of approximately 25%. Relative humidity, temperature, and pressure measured in each U-Pod were used to calculate absolute humidity. Over the course of multiple field deployments,

relative humidity sensors in four of the U-Pods drifted down, causing the lower humidity levels to be cut off or 'bottomed out'.  RH sensors were not replaced during field deployments in order to preserve consistency across different deployment periods, allowing for the possibility of a single comprehensive model to apply to all data from a single U-Pod.  After some experimentation in generating a 'master model' that could be applied to data from a given U-Pod for all collected field measurements, across several years, we determined that individual models for each deployment would be more effective, and replacing RH sensors that had drifted down would have been appropriate in support of the methods presented here.  We have since upgraded to Sensirion AG SHT25 sensors, which appear to be more robust and consistent over the course of long-term field deployments.  For measurements collected in the spring and summer of 2015 and the spring of 2017, we replaced the relative humidity (RH) signal of U-Pods with malfunctioning humidity sensors with signals from the closest U-Pod with a good humidity sensor and complete data coverage as noted in Table S1. Temperature and RH sensor measurements are usually collected from within each U-Pod sensor system in order to gain representative information about the environment the gas sensors are being operated in. Using an alternative source for RH data that are not onboard an individual U-Pod has the potential to increase uncertainty of quantified gas mole fractions.  We used replacement RH data from the closest available U-Pod instead of ambient measurements in order to more closely approximate humidity at the operating temperature within a U-Pod enclosure. The closest U-Pod with good humidity sensors ranged from several feet, when U-Pods were co-located during deployments in the DJ Basin at the BAO and GRET sites, to approximately fifty miles during deployments in the San Juan Basin.

When the U-Pods were initially deployed at the GRET site, on August 23[rd] of 2016, the RH sensors in all ten U-Pods malfunctioned, logging an error code of -99 instead of the relative humidity.  This malfunction seemed to coincide with the implementation of radio communication from each U-Pod to a central node in an effort to reduce trips to the field site to download data and to identify issues with data acquisition promptly.  No other impacts to sensor systems were observed in connection with radio communications.  RH signals in the U-Pods began logging correctly again in October when we stopped remote communication.  We replaced RH values for the U-Pods during this time period by utilizing data

from the Picarro Cavity Ring-Down Spectrometer that was co-located at GRET with the U-Pods. Water mole fractions measured by the Picarro were converted into mass-based mixing ratios to match the units of the absolute humidity signal in the U-Pod data. We applied an adjustment to this absolute humidity signal so that it matched observations in U-Pods during the following month when good RH sensor data was available, to account for the fact that temperatures were higher in U-Pod enclosures than the ambient environment. We then replaced the relative humidity signal in each U-Pod from August 23[rd] through October 1[st] in 2016 with the mixing ratios derived from Picarro measurements. Using the temperature and pressure logged in each U-Pod along with the absolute humidity from the Picarro, relative humidity was calculated for each U-Pod during this period.

To perform regressions toward field calibration of sensors, the reference and U-Pod data needed to be aligned. When reference measurements with minute time resolution were available for both training and corresponding testing periods, minute median data from the U-Pods were used. Medians were used as opposed to averages in order to reduce the potential influence of sensor noise as well as to remove short duration spikes in the reference and sensor data that resulted from air masses that may not have been well mixed across the reference instrument inlets and the U-Pod enclosures. When reference data were instead available with only 5-minute or 60-minute time resolution, U-Pod medians were calculated for to match that time step. In order to test models using the same time resolution they were trained with, the time resolution of reference and sensor measurements for corresponding training/testing datasets were matched, if necessary, by taking medians of the dataset with higher time resolution to match the data with the longer time resolution. The first 15 minutes of data after any period that the U-Pods had not recorded data for the previous 5 minutes was removed in order to filter transient behavior associated with sensor warm-up. During a given deployment, the data removed to avoid sensor warm-up transients constituted less than 1%.

When time was included in a model as an input, the absolute time was used. Specifically, we used the datenum value from the MATLAB environment, which is defined by the number of days that have elapsed since the start of January 1[st], in the year 0000. A model was extrapolated in time when ever

training data does did take place both before and after a given test deployment period. In several case studies we present, model training only took place after the test deployment period, comprising a 'post only' calibration. In Colorado, and more broadly in the western United States, ambient temperatures change significantly across the seasons throughout the year, so if a model is extrapolated in time,

extrapolation in temperature often results as well.

### 2.4 Calibration Model Techniques

In this work, we explore how well field calibration methods hold up in new locations, a topic which has not yet been sufficiently addressed by the scientific community. As in (Casey et al., 2017), direct LMs and ANNs were trained with a number of different sensor input sets to map those inputs to

10 target gas mole fractions measured by reference instruments. Direct LMs implemented were multiple linear regression models given by

$$r = \ p_1 + \ p_2 s_1 + \ p_3 s_2 + ... + p_n s_{n-1}, \tag{1}$$

where $r$ is the target gas mole fraction (measured by a reference instrument) $s_1 - s_{n-1}$ are sensor signals from U-Pods that are included as model predictor variables, and $p_1 - p_n$ are corresponding predictor

coefficients.

ANNs designed for regression tasks, like those employed in this work, generally consist of artificial neuron nodes that are connected with weights. Weights are initiated with randomly assigned values. An optimization algorithm is then employed iteratively adjust the values of these weights in order to

20 map a given set input values to corresponding target values. An example of a very simple feed forward neural network, and how weights are propagated through it are depicted in Fig. 3. In this work, ANNs were designed by assigning U-Pod sensor signals to artificial neurons in an input layer and assigning target gas mole fractions for an individual gas species, measured by a reference instrument to a single output neuron. Nonlinear, tansig, artificial neurons in one hidden layer for $O_3$ or two hidden layers for

$CO_2$ (accordance with our earlier findings for each target gas species (Casey et al., 2017)) were then added between input layer and the network output neuron. Additionally, bias neurons, each assigned a value of 1, were connected to neurons in the hidden layer(s) so that individual connecting weights could

be activated or deactivated during the optimization process. The number of neurons in each hidden layer was set equal to the number of inputs included in a given ANN. Fig. 4 shows a diagram of an ANN architecture employed in this work, when there were five inputs.

For ANN training we employed the Levenberg Marquardt optimization algorithm with Bayesian Regularization (Hagan et al., 1997). The Levenberg-Marquardt algorithm combines the Gauss-Newton and Gradient Decent methods, towards incremental minimization of a cost function, which is defined by the summed squared error between the ANN output and target values as a function of all of the weights in the network. Training begins according to the Gauss-Newton method, in which the Hessian matrix,

the second order Taylor series representation of the local shape of the error surface, is approximated as a function of the Jacobian matrix and its transpose, significantly reducing required training time. Network weights are adjusted accordingly during each training step to reduce error. If the cost function is not reduced in a given training step, an algorithm parameter is adjusted so that optimization more closely approximates the gradient decent method (a first order Taylor series representation of the local

shape of the cost function), providing a guarantee of convergence on a cost function minimum. Since local minima may exist across the error surface, it is important to train the same network multiple times, with different randomly assigned starting weights, in order to assess the stability of ANN performance. In this work, each ANN was trained 5 times.

In the implementation of Bayesian Regularization, a term is added to the sum of squared error cost function as a penalty for increased network complexity in order to guard against over fitting. A two level Bayesian inference framework is employed, operating on the assumptions that the noise in the training data is independent, normally distributed, and also that all of the weights in the ANN are small, normally distributed, and unbiased (Hagan et al., 1997). In preliminary ANN tests we found that over

fitting occurred even when Bayesian Regularization was used, so we additionally implemented early stopping, which proved to be effective in the reduction of over fitting. To implement early stopping, a portion of training data is set aside as validation dataset, and during training. Training continues so long as the error associated with the validation dataset is reduced. When the error associated with the

validation dataset is no longer being reduced, training stops early. For ANNs, training datasets were divided in half on an alternating 24-hr basis, with half used for training and half used as validation data for early stopping. Input signals for both LMs and ANNs were normalized so that they ranged in magnitude from -1 to 1 since this practice is recommended for the ANN optimization algorithm used

(Hagan et al., 1997).

## 2.5  Calibration Model Evaluation and Testing

To evaluate the performance of each of the ANN and LM models that were generated using training data then applied to test datasets, we explored residuals, the coefficient of determination ($r^2$), root mean squared error (RMSE), mean bias error (MBE), and centered root mean squared error (CRMSE). The

CRMSE is an indicator of the distribution of errors about the mean, or the random component of the error. The MBE, alternatively, is an indicator of the systematic component of the error. The sum of the squares of the CRMSE and the MBE is equal to the square of the total error, the square root of which is defined by the RMSE.

First, we generated and applied the best performing model, as determined in our previous work (presented in Table 3), to data from each new case study. Each new case study was selected to challenge models in different ways in order to evaluate the resiliency of the findings from our previous study when challenged by different circumstances. Then we tested LMs for $CO_2$ and $O_3$ that contained only the primary target gas sensor for each species, as well as temperature and absolute humidity as

inputs. Finally, we generated, applied, and evaluated the performance of a number of LMs and ANNs with different sets of inputs for each case study in order to see which specific model performed the best for each individual case study. The $r^2$, RMSE, and MBE for each of these alternative models when applied to test data are presented in the supplemental materials (SM) in Fig. S2 through Fig. S7, along with representative scatter plots and time series comparing the performance LMs and ANNs for a given

set of inputs. In Fig. S2 through Fig. S7, the best performing model inputs for each train/test data pair are shaded in purple. The type of model that performed the best (ANN vs. LM) is indicated in the caption of each figure. We discuss both the performance of the previously determined best fitting

model (generated using data from the GRET Spring 2017 case study) when applied and generated to data from new case studies, and the performance of models that were tuned to perform the best for each individual case study. From these comparisons, we draw insight into circumstances that challenge model performance in terms of relative local emissions characteristics, location, and timing between 5 model training and testing pairs. Table 4 lists the relative timing and parameter coverage between model training and testing periods for dataset pairs, highlighting instances of incomplete coverage during training that led to model extrapolation during testing.

## 3    Results and Discussion

### 3.1    BAO and SJ Basin Summer 2015

10 The set of deployments we conducted in the summer of 2015 is particularly useful to the objective of characterizing how well field calibration models can be extended to a new location relative to their performance where they were trained. During the testing period, two U-Pods were located at BAO, where training took place, while seven U-Pods were co-located with reference measurements for $O_3$, and two U-Pods were co-located with reference measurements for $CO_2$ in the SJ Basin, across Colorado 15 and over the state line in New Mexico. Sampling sites at BAO, in the DJ Basin, and throughout the SJ Basin were all influenced by oil and gas production activities and their associated emissions to some extent, but the composition of the production stream is different in each basin. In the SJ Basin, particularly the northern portion of the basin where all our sampling sites were located production is dominated by coalbed methane. In contrast, many wells in the DJ Basin produce both oil and gas 20 leading to greater relative abundance of heavier hydrocarbons in emissions. The DJ Basin air shed is also more strongly impacted by urban emissions than the SJ Basin air shed, and is more strongly influenced by mobile sources with Denver, Boulder, Fort Collins, Greeley, and many other smaller communities in its midst and along its borders. The Four Corners region, where the SJ Basin is situated, has a much smaller population density. Additionally, while there are some agricultural activities and 25 associated emissions in and around the SJ Basin, there is a significantly larger agricultural industry in and around the DJ Basin. SJ Basin sampling sites spanned a range of elevations, including some that

were higher and some that were lower than the BAO Tower, coinciding with a wide range of atmospheric pressure at the distributed sampling sites.

We began by testing the best-performing $CO_2$ model, as determined in our previous work (Casey et al., 2017), on data from this case study, during the summer of 2015. ANNs were trained for each U-Pod using data from the BAO Tower with the following inputs from each U-Pod: eltCO2 (ELT S300 CO2 sensor), temp (temperature), and absHum (absolute humidity), then tested on data collected at the BAO Tower and at sampling sites in the SJ Basin. The performance of these ANNs when applied the test data are presented in Fig. 5 and Fig. 6. Fig. 5 shows scatter plots of U-Pod $CO_2$ vs. reference $CO_2$ during the test data period for sensors located at BAO as well as sensors that were located at distributed sampling sites throughout the SJ Basin. The scatter plots show that while there was generally a smaller dynamic range of $CO_2$ at the SJ Basin sites relative to BAO, model performance did not appear to be impacted or degraded by spatial extension to these locations in the SJ Basin. The line of best fit for Fort Lewis site (periwinkle) is even closer to the 1:1 than the lines of best fit for two U-Pods located at BAO (black and grey). Overlaid histograms of residuals in the bottom right corner of Fig. 55 show that $CO_2$ residuals from each of the SJ Basin U-Pods are generally centered and evenly distributed about zero with similar spread.

U-Pod $CO_2$ average residuals during this test period, using the best performing ANNs from our previous study, are plotted according to time of day and date in Fig. 6. While the use of ANNs in place of LMs reduces U-Pod $CO_2$ residuals significantly with respect to temperature, some daily periodicity in the residuals for all four U-Pods is apparent in the upper plot in Fig. 6 that shows residuals by date. The lower plot in Fig. 6, showing residuals by time of day, demonstrates that $CO_2$ from three of four U-Pods was generally under predicted during early hours of the morning and generally over predicted during afternoon and evening hours. Interestingly, this trend in residuals by time of day is more pronounced for the two U-Pods that remained at BAO. Upon examination of overlaid histograms showing distributions of parameters during model testing and training periods, in Fig. S12, and model time series and residuals plots in Fig. S3, there is no indication of model extrapolation at the BAO site, nor the sites

in the SJ Basin (with the exception of pressure due sampling site altitudes) and no significant trends of concern with respect to residuals and model inputs.

Next we evaluated the best model type and set of inputs for $CO_2$ based on this specific case study.
Differing from our previous findings, for this group of training and testing data pairs from the summer of 2015 at the BAO and SJ Basin sites, the inclusion of the e2vVOC (e2v MiCs-5521) and alphaCO (Alphasense CO-B4) sensor signals noticeably improved the RMSE in the quantification of $CO_2$. While the inclusion of these two secondary sensor signals didn't result in the best performance in our previous study, using data from the GRET site (Casey et al., 2017), their inclusion did not degrade performance
relative to the models that included just eltCO2, temp, and absHum signals as inputs, so including these sensor signals may be appropriate as a general rule, in areas that are strongly influenced by oil and gas production activities. Generally, using rh vs. absHum signals as ANN inputs did not have a measurable impact on model performance, though linear models were sometimes found to perform better when the absHum signal is used instead of the rh signal. From Fig. S2, it is apparent that inputs including e2vCO
(e2v MiCs-5525), temp, rh, e2vVOC, and alphaCO sensor signals as model inputs resulted in the lowest RMSE for U-Pods at BAO as well as at the two SJ Basin sites. Plots analogous to those presented in Fig. 5 and Fig. 6, but with this best performing set of inputs for the present data set pairs are presented in the SM, in Fig. S24 and Fig. S25 respectively.

For $O_3$, we similarly began by testing the model that was found to perform the best from our previous study on data from this case study. $O_3$ was quantified using data from the two U-Pods deployed at BAO and seven of the U-Pods deployed at SJ Basin sampling sites using ANNs with the following inputs: e2vO3 (e2v MiCs-2611), temp, absHum, e2vCO, e2vVOC, figCH4 (Figaro TGS 2600), and figCxHy (Figaro TGS 2602). These same inputs and model configuration were also found to be the best
performing for the U-Pods at the BAO site and the majority of SJ Basin 2015 dataset pairs as noted in Fig. S2. Interestingly though, LMs with this same set of inputs performed competitively well for three of the seven U-Pods in the SJ Basin in terms of RMSE and $r^2$. The observation that LMs performed competitively well at a subset of SJ Basin sites is likely connected to the relative abundance of

hydrocarbons and other potentially interfering oxidizing and reducing gas species at individual sampling sites, diverging from conditions present during model training at the BAO site. ANNs can better represent the influence of these interfering species than LMs during training, but appear to have lose their ability to do so for this subset of microenvironments in the SJ Basin.

Scatter plots and trends in residuals are presented in Fig. 7 and Fig. 8 for $O_3$. These plots show the performance of U-Pods at BAO relative to those at SJ Basin sites in the quantification of $O_3$ during the test data period. U-Pod $O_3$ measurements at Fort Lewis, Navajo Dam, and the Sub Station did not agree with reference measurements as well as U-Pod $O_3$ measurements from the other four SJ Basin sites. As

noted earlier, U-Pods at the Navajo Dam and Sub Station sites had faulty relative humidity sensor data, so humidity from the U-Pod located at the Ignacio site was used in place of their humidity signals. Since the Ignacio site was located approximately twenty-two and fifty miles away from the Navajo Dam and Sub Station sites respectively, this could have introduced some additional error into the application of a calibration equation, particularly since we showed earlier that $O_3$ ANNs like the ones we employed

here are very sensitive to humidity inputs (Casey et al., 2017). Spatial variability in humidity across tens of miles could be significant as isolated storms (which are on average 15 miles in diameter) propagate throughout the region in the summer. At the Fort Lewis site, a 2b Technologies model 202 $O_3$ analyser was employed as a reference instrument, differing from the Thermo Scientific 49i, Thermo Scientific 49is, and Teledyne API T400 instruments utilized for reference measurements, elsewhere in

the SJ Basin, and the Thermo Scientific 49c that was operated at the BAO site and used for model training. Of all the reference instruments, only the 2b Technologies model 202 $O_3$ at the Fort Lewis site was operated in a room that was not temperature controlled, as such, some bias may have been introduced to the Fort Lewis $O_3$ reference measurements. Different instruments, operators, calibration and data quality checking procedures could have contributed to observed discrepancies. It is also

possible that the microenvironment at each of these three sites contributed to lower model performance. Fig. S1 shows that differences among U-Pod $O_3$ performance during the test deployment period were larger than those observed during the training period among the same U-Pods. Therefore, the incongruous field calibration performance phenomena we observed seems to be connected to unique

characteristics associated with humidity sensor signal replacement or individual sampling site characteristics; possibly relative abundance of oxidizing and reducing molecules in the local atmosphere, which could interfere with sensor responses to their target gas species, as opposed to the quality of individual gas sensors in each of those U-Pods.

All SJ Basin U-Pod $O_3$ measurements systematically over estimate lower levels of $O_3$ each night, a trend apparent in the scatter plots in Fig. 7 and in the residuals by time of day plot in Fig. 8. Upon examination of the scatter plots in Fig. 7, U-Pods at some sampling sites had positive bias for higher $O_3$ measurements as well (Shiprock, Ignacio, Sub Station, and Bloomfield), while for others, bias at the

10 higher end of $O_3$ distributions did not appear to be present (Navajo Dam, Fort Lewis, and Bondad). The residuals by time of day plot in Fig. 8 shows that the two U-Pods at BAO did not have significant trends in their residuals according to the time of day, but that U-Pods deployed at SJ Basin sites consistently over estimated nighttime $O_3$. The residuals are also plotted with respect to temperature in Fig. 8, where all U-Pods, even those at BAO to a lesser extent, appear to over predict $O_3$ at lower temperatures, which

generally occurred at night. In general, the times of day that correspond to the highest $O_3$ levels had the lowest residuals, with some exceptions at the Fort Lewis and Navajo Dam sites.

Fig. 8 includes a plot of the residuals across the duration of the deployment period, showing no significant sensor drift in measurements for any of the U-Pods. This plot also shows that the highest

residuals observed generally occurred over short periods in time, particularly for the Fort Lewis (blue) and Sub Station (magenta) sites. In order to further explore the performance of field calibration models for $O_3$ at SJ Basin sites relative to BAO, the combined parameter space of temperature with $O_3$ reference mole fractions and temperature with absolute humidity are presented in Fig. 9. The combined temperature and reference $O_3$ parameter space appears to be similar for all of the U-Pods, both at BAO

and the SJ Basin sites. However, there appears to be some outlying combined temperature and humidity parameter space at the Sub Station site and at the Navajo Dam site. Brief excursions, lasting approximately 2 – 4 hours, of high humidity (up to 0.025 kg/kg, relative to the upper bound of absolute

humidity observed at other sampling sites of 0.013 kg/kg) may be connected to some of the large short-term residuals observed at these two sampling sites.

The majority of U-Pods stopped logging data, unfortunately, at one point or another during these deployments. Periods of missed data during the month-long deployment included approximately one day at the Shiprock site, two days at the Bloomfield site, four days at the Sub Station site, nine days at the Fort Lewis site, and seventeen days at the Navajo Dam site. We carried out frequent sampling site visits (on a weekly or biweekly basis as logistics and travel to remote locations in some cases allowed) in order to identify and fix problems as they arose during field deployments. Operational issues were predominantly attributable to power supply problems associated with BNC bulkhead fittings and corrupted micro SD cards. The periods of missing data are reflected in the plots of residuals by date in Fig. 6 for $CO_2$ and in Fig. 8 for $O_3$. Fortunately, no drift over the course of the deployment period was observed in these plots.

## 3.2 Insight from Additional Case Studies of Field Calibration Extension to New Locations

### 3.2.1 Urban calibration moved to rural/peri-urban setting: Dawson Summer 2014

The Boulder County deployment in the summer of 2014 was used to test how well a field calibration for sensors in one U-Pod, generated in a busy urban area (at CAMP in downtown Denver), could be extended to a peri-urban setting (at Dawson School in eastern Boulder County). Training took place at CAMP for several days each month, before and after each approximately month-long deployment period at Dawson School over the course of four months. Fig. S7 shows the performance of a number of ANN and LM-based CAMP field calibrations with different sets of inputs at this Dawson School test site. In this case study, LMs performed better than ANNs across all sets of sensor inputs. Unlike findings from our previous study (Casey et al., 2017), including secondary metal oxide type sensors as inputs didn't help to improve model performance. The best performing set of inputs included just e2vO3, temp, and absHum signals. The very different relative abundance of various oxidizing and reducing compounds in downtown Denver relative to the Dawson School site, surrounded by open grassy fields, and in closer proximity to oil and gas production activities, may be the reason why

including additional gas sensors as model inputs and the use of ANNs failed to improve the quantification of U-Pod $O_3$ in this case. Relatively short training durations could also contribute to this finding, based on findings from our previous work (Casey et al., 2017).

The fact that LMs performed better than ANNs in this case (with an r2 of .95 and RMSE of 0.35 ppb for LMs, as opposed to an r2 of .9 and an RMSE of 5.1 ppb for ANNs) may have to do with the general expectation that LMs be more resilient to extrapolation than ANNs. Notably though, neither ANNs nor the LMs captured the highest levels of $O_3$ at Dawson School well. We attribute the poor performance at high levels of $O_3$ at this site, those in exceedance of about 70 ppb, to extrapolation of the $O_3$ mole
fractions encompassed during the training period. The LM generally performed well within the $O_3$ levels covered during the training period. Across applications, ANNs have been found to be unreliable when extrapolated, due to the nonlinear nature and complexity of the relationships they represent. Though they are generally expected to be more robust to extrapolation that ANNs, increased uncertainty in measurements can also be introduced to LMs when parameters are extrapolated. In order to have
high confidence in measurements of uncommonly high mole fractions of a target gas, the model training period has to encompass the full possible range. Combining both field calibration and lab calibration data together in a training dataset could accomplish this type of coverage. If extrapolation is expected to occur with respect to the target gas mole fraction, as in this case study, the use of an inverted LM may yield better results than LMs or ANNs. We describe inverted LMs and their potential advantages
in our previous work (Casey et al., 2017). Keeping in mind this finding about $O_3$ extrapolation, for ambient measurements in the DJ Basin, for subsequent deployments, we selected field calibration sites that were more representative of distributed sampling site locations, outside of the dense urban environment in downtown Denver, where $O_3$ did not get as high, likely due to increased titration of $O_3$ at night in connection with abundant $NO_X$ compounds.

**3.2.2    Post only calibration moved across the state:  SJ Basin Spring 2015**

We also examined model performance that was subject to extrapolation in time and temperature. We present $O_3$ model performance data from four U-Pods that were co-located with reference instruments

in the SJ Basin in the spring of 2015, at the Navajo Dam, Sub Station, and Bloomfield sites. Two U-Pods at the Bloomfield site provide a set of duplicate measures. Fig. S4 shows the performance of a number of ANN and LM-based BAO field calibrations with different sets of inputs at this SJ Basin test sites in the spring of 2015, just prior to the summer 2015 BAO training period. U-Pod $O_3$ was quantified for these deployments using training data from the same co-location period at BAO that was used toward quantification of the summer 2015 SJ Basin deployment, described in section 3.1.

The addition of time as a model input didn't seem to improve the performance of either ANNs or LMs and ANNs generally outperformed LMs. Gas sensor manufactures don't clearly define sensor lifetimes, but sensors are generally expected to loose sensitivity over time. For example, Alphasense CO-B4 electrochemical sensors are expected to have 50% of their original sensitivity after two years (Alphasense, 2015). The heater resistance in a given metal oxide type sensor is expected to drift over time, influencing sensor measurements (e2v Technologies Ltd., 2007). Masson and colleagues observed a significant drift in a metal oxide sensor heater resistance over the course of a 250 day sampling period in a laboratory setting (Masson et al., 2015). While we did not measure and record metal oxide sensor heater resistance for sensors included in U-Pods, we have investigated eltCO2 and e2vO3 sensor signal drift from the summer of 2015 through the summer of 2017. These data are presented in Fig. S26. Systematic downward drift in all eltCO2 sensor signals is apparent over this time frame. A clear and consistent pattern of systematic drift over this time period is less apparent for e2vO3 sensors. Since the training data was collected immediately after, the test data period, and since the test data period was relatively short (approximately one month) sensor drift could be negligible across the combined training/testing time frame. U-Pods experienced colder temperatures during this spring deployment than were encompassed subsequently in the BAO training period. Linear models generally resulted in more bias than ANNs. Again the model for $O_3$ that was found to perform best in our previous (Casey et al., 2017), an ANN with temp, absHum and all metal oxide sensor signals as inputs, performed the best at sites included in this case study, with one exception. At the Sub Station site the inclusion of the figCxHy sensor signal decreased model performance. Additionally, the performance of all models tested at the Sub Station site during the SJ Basin Spring 2015 deployment was significantly

worse in terms of MBE than model performance at other sites, both LMs and ANNs with different sets of inputs. Since this sensor signal input augmented model performance at the same sampling location during the summer deployment period, this finding could be attributable to the extrapolation with respect to temperature that occurred during the test period of this case study. As discussed in the introduction, metal oxide sensor sensitivity to different gas species can vary along with sensor surface temperature. Models were trained to use the figCxHy sensor signal, across the ambient temperatures in encompassed by the training data, to help account for the influence of confounding gas species at the BAO site. We think it is possible that the different temperatures in combination with the unique mix of gas species present at the Sub Station site, which the figCxHy sensors are highly sensitive to, caused the ANN to perform worse. The Sub Station site is close to two large coal-fired power plants, indicated in Fig. 11 by orange markers in the SJ Basin pane. It is possible that emissions from the San Juan Generating Station (north) and/or the Four Corners Power Plant (south) uniquely influenced the response of this particular Figaro sensor in ways that are not well represented at BAO in the DJ Basin, or present at other SJ Basin sampling sites. Several-hour long enhancements or spikes are apparent in the raw eltCO2 and alphaCO sensor signals in the U-Pod deployed at the Sub Station site, indicating the presence of a near-by combustion-related emissions source. Another indication of a near-field power plant plume across the deployment area is apparent, in the form of several-hour long enhancements of reference measurements of NO and $NO_2$ at the site.

### 3.2.3 Post only calibration moved 40 miles across the DJ Basin: BAO Summer 2016

In testing the performance of field calibrations that were generated using data collected at the GRET site in 2017 and applied for the quantification of $O_3$ at BAO in the 2016, across the DJ Basin, we were interested to find that again, the inclusion of time as a model input did not yield any improvements in calibration equation performance, even though model training took place several months after the test period. Fig. S5 shows the performance of a number of ANN and LM-based GRET field calibrations with different sets of inputs at this BAO test site the previous summer. Another interesting finding from this training/testing dataset pair was that the addition of secondary metal oxide type gas sensors, didn't seem to help improve the performance of field calibration equations either. Fig. S5 shows that ANNs

performed better than LMs and that the most useful set of inputs included just e2vO3, temp, and absHum. Similarly, the performance of field calibration equations for $CO_2$ generated at GRET in 2017 and applied to data from BAO in the summer of 2016, did not seem to be augmented by the inclusion of additional gas sensor signals, though the inclusion of time as a predictor was useful. In the case of $CO_2$,

LMs outperformed ANNs, which could be largely attributable to notable instability associated with the performance of ANNs when time was included as an input. For $CO_2$, we expected the inclusion of time as an input to be a useful to model performance across this time frame, owing to observed trends of decreased $CO_2$ sensor sensitivity in time. To keep the power requirements for the U-Pod sensor systems low, and to keep systems quiet, fans were used to exchange air in the enclosures as opposed to pumps.

As a result, the air entering the enclosures was not filtered, and sensors were exposed to some dust over time. This dust exposure is likely largely responsible for observed decreases in $CO_2$ sensors sensitivity over time, shown in Fig. S26. Decreases in infrared lamp intensity over time may also play a role. In the case of $CO_2$ sensors, the implementation of pumps to draw new, filtered air into sensor enclosures could likely significantly reduce lose rates in the sensitivity of an individual sensor over periods of

continuous deployment in ambient environment. While we are not sure why ANN performance tended not to benefit from the addition of a time input, while LM performance did, it is likely attributable to the extrapolation of the time input, since only data that was collected significantly subsequent to the test data period was used for training. ANNs are expected to be able to better represent time decay trends if data from measurements both prior and subsequent to the test period are used in training, so that there is

no extrapolation with respect to the time input.

### 3.2.4  Post only calibration applied to the same location:  GRET Fall 2016

To investigate if reduced performance from these GRET to BAO field calibration tests were more connected to the new deployment location or to the significant extrapolation with respect to time of the calibration models, we generated calibration equations based on similarly long training periods at

GRET and applied them to data collected prior to the training period at GRET in the fall of 2016. We couldn't draw strong conclusions from this comparison, unfortunately, because of an issue with humidity sensors, described in the methods section and below. Fig. S6 shows the performance of a

number of ANN and LM-based GRET field calibrations with different sets of inputs at the GRET test site during fall of the previous year.  For $O_3$ models, the best performing ANN inputs for this dataset pair were the same ones that we found in our previous study (Casey et al., 2017), with the exception of the  humidity signal.  The fall 2016 GRET test period coincided with the time period U-Pod absolute

humidity was replaced using mixing ratios from a co-located Picarro due to missing humidity sensor data. Interestingly, when this 'borrowed' humidity signal was not included as an input, the model performance markedly increased and became competitive with other 'same location' test deployment case studies.  In our previous work, we showed that $O_3$ models were very sensitive to the humidity signal input  (Casey et al., 2017).  In this case study, it seems that replacing actual humidity signals with

closely approximated humidity signals, negatively influenced model performance.   In order to investigate this observation further, we tested the influence of replacing humidity data in the same manner, using mixing ratios from the same co-located Picarro, on test data from the GRET Spring 2017 case study.  A comparison of model performance under normal and this 'borrowed RH' circumstance is presented in Fig. S27 in the SM.  $O_3$ model performance was negatively impacted when 'borrowed' RH

values based on Picarro data replaced U-Pod RH sensor signals.  From these findings, it seems likely that the inclusion of multiple metal oxide type sensors as inputs in the model, which all respond strongly to humidity fluctuations, helped the ANN to effectively represent the influence of humidity in the system, more so than including a 'borrowed RH' signal from another instrument.  We tested models with multiple gas sensor signals and no humidity signal as inputs for a number of other case studies as

well (as seen in Fig. S2, Fig. S4, and Fig. S5), when good humidity data from U-Pod enclosures was available, but they did not turn out to be the best performing model in any of these other tests.

### 3.3   Evaluation of models across training/testing dataset pairs

For each of the case studies, we compare the relative model performance under three governing model-training paradigms.  The first of these paradigms includes linear models with only the primary gas

sensor signal, along with temperature, and absolute humidity signals as inputs.  Performance of these models is shown in Fig. 10.  The next paradigm includes models that were found to perform best for each trace gas in our previous work.  Performance of these models is shown in Fig. 11.  The third

paradigm includes models that were optimized for each case study specifically. Performance of these models is shown in Fig. 12. Table 5 and Table 6 show the mean and standard deviation of model performance metrics for each of the case studies presented. Table 7 shows the percent change in model performance metrics when one model-training paradigm is used in place of another, highlighting relative benefits associated with the implementation of different models for $O_3$ and $CO_2$.

Fig. 10, Fig. 11, and Fig. 12 contain target plots showing the MBE and CRMSE of models from each dataset pair in terms of absolute mole fractions and mole fractions normalized uniformly by the standard deviation of reference data during the spring 2017 GRET deployment. In the SM, Fig. S23 contains target diagrams equivalent to those presented in Fig. 12, but with individually normalized MBE and CRMSE, according to the standard deviation of reference measurements during each individual test period. The outer circle's radius in each of these target diagrams denotes an error-to-signal ratio of 1. The inner circle's radius in each of these target diagrams encompasses the performance of models when they were tested at the same location that they were trained and when training data bookended the test period, so that there was no extrapolation of the model across time or deployment location. We present our findings in the form of these target diagrams in order to compare our findings with those presented in several particularly relevant previous studies focused on the field calibration of low-cost sensors (Spinelle et al., 2015, 2017; Zimmerman et al., 2017).

Fig. 10 and Fig. 11 show that for $CO_2$, ANN models generally performed slightly better than LM models with the same set of inputs, though models that were extrapolated more than several months in time were the exception. For $O_3$, ANNs that included multiple secondary metal oxide sensor signals as inputs were also found to generally perform slightly better than the relatively simple LMs that didn't include any secondary gas sensors as inputs over all (with exceptions for individual case studies). This can be seen in Table 7 and in Fig. 10 and Fig. 11, with all plot markers falling within the outer radius in Fig. 11 (ANNs) but some plot markers falling outside the outer radius in Fig. 10 (LMs). Models that were not moved to a new location for the test period gained the most benefit in their performance when ANNs were used instead of LMs, resulting in a smaller inner radius in the target plots in Fig. 11 relative

to Fig. 10 for both $O_3$ and $CO_2$. The target diagrams in Fig. 10 and Fig. 11 show some degradation in performance when models were applied to data in new locations and when training data took place only after the test period. The of the target plots in Fig. 10 and Fig. 11 demonstrate that bias was introduced when field calibration models were extrapolated in terms of time, when training periods only encompassed data after the test data period and not prior. Interestingly, there are noticeable similarities between the target plots for $CO_2$ in Fig. 10 and 11 and the target plots for $O_3$ in Fig. 10 and 11.

The relative performance of models among each training/test dataset pair remained fairly consistent across the different models employed in data quantification. These systematic trends highlight the importance of model training and testing circumstances relative to specific field calibration model types and inputs. For the BAO Summer 2016 case study, when time was extrapolated significantly, and when models were moved across the DJ Basin, $CO_2$ and $O_3$ were both better represented by LMs than ANNs. $CO_2$ and $O_3$ models did not benefit from additional gas sensors added as inputs either for this case study. In Fig. 11, of models that performed best for each species as determined in our previous study, models that were not extrapolated in time for $CO_2$, and all $O_3$ models, plot markers fall within the outer radius, meeting performance standards framed by previous studies (Spinelle et al., 2015, 2017; Zimmerman et al., 2017). In Fig. 12 the best field calibration model performances for each case study all fall within the outer radius, showing good performance, and indicating that incomplete coverage of parameter space in terms of atmospheric chemistry, weather patterns, sampling location, and sampling timing, can be addressed to some extent by tailoring field calibration models and their inputs to specific training/testing datasets pairs.

For $CO_2$ we found that field calibration models generally extended with good performance to new locations. ANNs outperformed LMs when training took place pre and post of a test deployment. When training only took place after a test deployment LMs performed better. LMs seem to be better at extrapolating in time. Over time, ELT NDIR $CO_2$ sensors seem to lose sensitivity and/or drift. When $CO_2$ models were extended back in time, significant bias resulted when time was not included as an input. ANNs were not able to extrapolate in time with any success and their performance became very

unstable when time was added as an input, an occurrence that is apparent in Fig. S5 and Fig. S6. Models performed better when they were extended spatially, all the way across Colorado from the DJ Basin to the SJ Basin, than they did when they were extended back in time. Extension of a LM back in time and across the DJ Basin (from GRET in 2017 to BAO in 2016) resulted in significant MBE relative to the other case studies. The inclusion of multiple additional gas sensors augmented model performance when extended back in time at the same location as training took place, but not at a new location.

For $O_3$ we found that ANNs with the same set of inputs worked best across a number of case studies, informed by all the metal oxide sensor signals as well as temperature and humidity. The extension of models to new locations often resulted in increased MBE or systematic error, and in some cases increased CRMSE or random error. Some observed bias in the extension of models to new locations could be attributable to different reference instruments with different operators and/or different calibration and data quality measures employed. $O_3$ model extension to new locations seemed to be more impactful than extension back in time. Interestingly, additional metal oxide sensor signals remained helpful when models were extended all the way across Colorado, from BAO to the SJ Basin, but these additional gas sensor signals did not remain helpful when $O_3$ models were extended across a county line, from Adams County (CAMP) to Boulder County (Dawson) or from Weld County (GRET) to Boulder County (BAO). ANNs generally performed better than LMs for $O_3$, with the exception of these two Front Range case studies (Dawson Summer 2014 and BAO Summer 2016). We found in our previous study that shorter training times led to decreased performance in ANNs and sometimes increased performance in LMs. The training time used in the CAMP to Dawson case study was relatively short, which could have contributed to the superior performance of LMs over ANNs. For the BAO Summer 2016 case study, both ANN and LM markers are included (each with the same inputs: e2vO3, temp, and absHum). LMs had smaller random error but ANNs had smaller bias, highlighting an important consideration in the application of one or the other to inform specific research or measurement goals.

## 4   Conclusions

Several previous studies have shown that multiple gas sensor signals and the implementation of supervised learning techniques can improve the performance of field calibration of low-cost sensors in the quantification of a number of atmospheric trace gas mole fractions.  We investigated how well a supervised learning technique (ANNs) held up when sensors were moved to a new location, different from where calibration model training took place.  We tested the spatial and temporal transferability of field calibration models for $O_3$ and $CO_2$ under a number of different circumstances using data from multiple reference instrument co-locations, using the same sensors over the course of several years, when sensors were deployed in two oil and gas production basins, along with urban and peri-urban sites.  We found that the best performing field calibration models for both $O_3$ and $CO_2$ were not consistent across all training and testing deployment pairs, though some patterns emerged in terms of model type and inputs in association with the spatial and temporal extension of calibration equations, from training to testing performed in oil and gas production areas.  The performance of $O_3$ models generally benefited from the inclusion of multiple metal oxide sensor signals in addition to the primary e2vO3 sensor signal, while the performance of $CO_2$ models relied more heavily on temperature and humidity inputs.  $CO_2$ model performance was impacted more by temporal extension than spatial extension.  In contrast, $O_3$ model performance was impacted more by spatial extension than temporal extension.

While ANNs and other supervised learning techniques have been shown to consistently out perform linear models in previous studies when training and testing took place in the same location, we find that this trend does not always hold when field calibration models are applied in a new location, with significantly different local emissions source signatures for $O_3$ models, or when model training data takes place more than several months subsequent to the model application period for $CO_2$ models.  We find that the implementation of calibration models that were well suited to specific training and test data pairs resulted in generally good test performance in terms of centered root mean squared error and mean biased error, scaled by reference measurement standard deviation, reported in target diagrams in previous studies.  For example, when models were significantly extrapolated in time and transferred to a

new location, a well-suited set of sensor inputs would generally not include secondary gas sensor signals.

LMs with just one primary gas sensor signal as well as temperature and humidity were found to outperform ANNs when models were applied to a location with different dominating sources of pollution in the case of $O_3$, like Downtown Denver relative to eastern Boulder County. These three-input LMs also outperformed ANNs when models were significantly extrapolated in time. While these LMs seemed to be more stable under circumstances of significant extrapolation in terms of local air chemistry and timing, we found that they did not extrapolate well in terms of the $O_3$ mole fraction, resulting in underproduction of $O_3$ values during the test period that exceeded those encompassed in the training data.

Field calibration models tested in new locations often resulted in the introduction of additional bias relative to field calibration models that were tested in the same location they were trained in. As seen in Fig. 12, plot markers from all case studies have very similar CRMSE values, but plot markers from case studies in which models were tested in new locations have larger MBE values than models that were tested in the same location as they were trained. Finding ways to effectively mitigate bias associated with new field deployment locations would further improve the performance of field calibrations toward quantification of atmospheric trace gases using arrays of low-cost sensors. Such improvements in the field of low-cost sensors will help to enable dense distributed networks of low-cost sensors to inform air quality in oil and gas production basins. The following findings from this work, and associated recommendations, are made to help inform the logistics of future studies that employ field calibration methods of low-cost gas sensors.

1. **Finding:** For $O_3$ models, LMs perform better than ANNs when the chemical composition of local emissions sources is significantly different in the model-training location relative to the model-application location. We found that when models were trained in an urban area with

significant mobile sources, then tested in a peri-urban area, more strongly influenced by oil and gas emissions, the differences in local sources of pollution were significantly different enough that LMs outperformed ANNs. Alternatively, when models were trained in one oil and gas production region and tested in another the different composition of local emissions (lighter vs. heavier hydrocarbons) was not significant enough for LM performance to surpass the performance of ANNs, though some positive bias was evident in predicted $O_3$ mole fractions.

**Explanation:** ANNs are very effective at compensating for the influence of interfering gas species through pattern recognition of a training dataset. However, if different patterns, in terms of the relative abundance of various oxidizing and reducing compounds in the air, are present in the testing location relative to the training location, ANNs may not able to compensate for the influence of interfering gas species as effectively. The relative abundance of interfering oxidizing and reducing compounds are not included as model parameters, but ANN performance is challenged by these circumstances.

**Recommendation:** When measuring $O_3$ or other gas species with a metal oxide type sensor, if the nature of dominant emissions sources at the model training location is significantly different than the nature of dominant emissions sources in the model application location, use a LM instead of an ANN. For the best performance, try to train models in locations with similar emissions sources to a desired sampling location. If the nature of dominant emissions sources at the model training and application locations are similar, signals from an array of multiple unique metal oxide sensors will likely augment model performance.

2. **Finding:** For $CO_2$ models, LMs perform better than ANNs when model training occurs significantly (more than several months) prior to or subsequent to the model application period.

**Explanation:** $CO_2$ sensors drift over time in terms of sensitivity and baseline response. When models are extrapolated in time (when training takes place more than several months prior or subsequent to the model application period), ANN performance can be compromised to a greater extent than LM performance. ANNs are able to represent relationships during training very effectively, and with significant more complexity and nonlinear relationships among time and

other model inputs than LMs.  The more complex the model, the less likely it can be extrapolate effectively.  LMs, with no interaction terms like we employ in this work, are not able to fit data and potentially complex patterns inherent in sensor drift over time during training as closely as an ANN, but the simple linear relationships they represent between the time input and the target gas mole fraction over the course of training are more likely to hold prior or subsequent to the training period.

**Recommendation:** When measuring $CO_2$ with a NDIR sensor, if model-training data is only available more than several months prior or subsequent to the model application period, use a LM instead of an ANN.  For the best model performance, use training data that is collected directly pre or post of the model application period, and preferably data from both pre and post of the model application period.  Training models using data from both pre and post of a given model application period helps models to encompass sensor drift over time as well as increases the likelihood of covering the full range of environmental parameter space that occurs during the model application period so that extrapolation of these parameters is avoided.

3. **Finding:** Extrapolation of an $O_3$ or $CO_2$ model in time, and especially significant extrapolation in time, can change both the type of model that is most effective, as well as the specific model input signals that are most effective.

**Explanation:** Low-cost sensors change over time, both in terms of their baseline response and in terms of their sensitivity to target and interfering gas species.  Different sensor types drift due to different physical phenomenon so further a generalization across sensor types is difficult.

**Recommendation:** Use training data collected directly pre and post of the model application period in order to implement a 'best performing model' for each gas species that can be applied using data from different model training and application pairs.

4. **Finding:** ANNs yield less bias and more accurate gas mole fraction quantification than LMs, even when transferred to a new location under the following circumstances: (a) extrapolation of training parameters is avoided during the model application period, (b) training takes place for

several weeks to a month prior and subsequent to the model application period, and (c) the dominant local emissions sources are similar in the model training and application locations.

**Explanation:**  Our previous study and multiple other ambient and laboratory based experiments have shown, arrays of low-cost sensors in combination with ANN regression models can support useful quantification of gases in mixtures and in the ambient environment because ANNs can more effectively represent complex nonlinear relationships among environmental variables and signals in a sensor system like a U-Pod than LMs.  With this work, we have explored limitations associated with these methods when challenged in different ways, as we present with a number of case studies.

**Recommendation:**  If minimizing error and bias in measurements of gas mole fractions using low-cost sensors systems is a primary goal, design sensor system training and field deployment experiments so that extrapolation of model training parameters is avoided during the model application period, so that training takes place for several weeks to a month directly prior and directly subsequent to the model application period, and so that the dominant local emissions sources are similar in the model training and application locations.  When these conditions are satisfied, ANNs can be robustly implemented, with better performance than LMs.

It is also imperative that sensor users keep in mind the primary importance of minimizing extrapolation of temperature, humidity and sensor signal from model training to application.  We show that field normalization trace gas quantification models can more readily be transferred across a large state from one oil and gas production to another, than from an urban to oil and gas production basin that are in closer proximity to each other.  We also show that pre and post model training, directly prior to and after field site deployment, is generally much more effective than pre or post model training alone, especially when the training takes place significantly before or after the deployment period. Along with these findings and general guidelines for future studies, we recommend further validation efforts in the extension of quantification of atmospheric trace gases using low-cost gas sensor arrays in oil and gas production basins and toward other ambient measurement applications. The findings presented here may be applicable and generalizable in the use of low-cost metal oxide, electrochemical, and non-

dispersive infrared sensor arrays in various configurations and sampling regions to characterize mole fractions of a number of atmospheric trace gases.  Future studies exploring the sensitivity of our findings to these factors are recommended.  In order to account for unique variations in sensor responses, in each individual sensor system, due to variations in manufacturing along with elapsed time

5 and specific exposure subsequent to manufacturing, we present models that are generated for each sensor system on an individual basis.   Future studies exploring the potential for universal calibration models would be very useful to the field.

The authors declare that they have no conflict of interest.

**Acknowledgements**

The many low-cost sensor and reference instrument measurements that facilitated this study were made possible with the gracious help of a number of agencies and individuals.  We'd like to thank Bradley Rink and Erick Mattson of CDPHE, Katherine Benedict and Jeff Collett of CSU, Detlev

Helmig and Jacques Hueber of INSTAAR, Gaby Pétron, Jon Kofler, Audra McClure, Bruce Batram, and Daniel Wolfe of NOAA, Michael King of NEPA, Christopher Ellis and Andrew Switzer of the SUIT AQP, along with Joe Cotie and Roman Szkoda of the NM AQB for sharing reference instrument data with us and facilitating our U-Pod measurements.  We thank Jana Milford for assistance in the preliminary analysis and reporting of results from the Boulder County Study.  Thanks to John Ortega

for preparing U-Pods and arranging permissions and logistics for the Boulder County project in 2014, including the Dawson School site we use data from in this work. Thanks are also due to include Brianna Yepa, Victoria Danner, Tasha Nez, Rebecca Bullard, Madeline Polmear, and Bryce Goldstien for helping to maintain U-Pods in the field as well as downloading and organizing data.  Bryce Goldstien made some maps in ArcMap of the SJ and DJ Basins with data from the Colorado Oil and Gas

Conservation Commission that were adapted and presented here.  It was a pleasure working with all these interesting and helpful people.  We thank Jana Milford for assistance in the preliminary analysis and reporting of results from the Boulder County Study as well as useful feedback and suggestions in review of this manuscript.  Many thanks also to Shelly Miller, Marina Vance, and Christopher Ellis for

kindly reviewing this work which was funded by Boulder County and the National Science Foundation Air Water Gas Sustainability Research Network under grant number CBET-1240584.

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

**Table 1: Gas sensors included in U-Pods along with the model input codes for each. The input code is an abbreviation for the make of the sensor, followed by the target gas species(s).**

| Sensor Type | NDIR | Metal Oxide | | | | | Electrochemical |
|---|---|---|---|---|---|---|---|
| Target Gas(s) | $CO_2$ | $CH_4$ * | CxHy ** | $O_3$ | VOCs | CO | CO |
| Model | S300 | TGS 2600 | TGS 2602 | MiCs-2611 | MiCs-5521 | MiCs-5525 | CO-B4 |
| Make | ELT | Figaro | Figaro | e2v/SGX | e2v/SGX | e2v/SGX | Alphasense |
| Code | eltCO2 | figCH4 | figCxHy | e2vO3 | e2vVOC | e2vCO | alphaCO |

**\*Light hydrocarbons**          **\*\*Heavy Hydrocarbons**

**Table 2**: **Reference instrument measurements at U-Pod sampling sites**

| Deployment | Reference Instrument | Calibration | Operator | Res |
|---|---|---|---|---|
| **Ozone** | | | | |
| **CAMP** | Teledyne API 400E | quarterly cal/nightly quality checks | CDPHE | 1 |
| **Dawson** | Thermo Electron 49 | pre cal/post cal check | INSTAAR | 5 |
| **BAO\*** | Thermo Scientific 49c | annual cal/monthly quality checks | NOAA | 60 |
| **Navajo Dam** | Thermo Scientific 49i | quartertly cal/weekly quality checks | NM AQB | 1 |
| **Bloomfield** | Thermo Scientific 49i | quartertly cal/weekly quality checks | NM AQB | 1 |
| **Sub Station** | Thermo Scientific 49i | quartertly cal/weekly quality checks | NM AQB | 1 |
| **Ignacio** | Thermo Scientific 49is | monthly cal/weekly quality checks | SUIT AQP | 1 |
| **Bondad** | Thermo Scientific 49is | monthly cal/weekly quality checks | SUIT AQP | 1 |
| **Shiprock** | Teledyne API T400 | quarterly cal/monthly quality checks | NEPA | 60 |
| **Fort Lewis** | 2b Technologies 202 | factory cal/post cal check | CU Boulder | 1 |
| **GRET** | Teledyne API T400E | quarterly cal/nightly quality checks | CDPHE | 1 |
| **Carbon Dioxide** | | | | |
| **BAO** | Picarro G2401 | | NOAA | 1 |
| **SJ Basin** | LI-COR LI-840A | pre + post cal:  zero precision span | CU Boulder | 1 |
| **GRET** | Picarro G2508 | periodic zero stability checks | CSU | 1 |

\*(McClure-Begley et al., n.d.)          Res = Time resolution of measurements in minutes

**Table 3: Best performing models, as determined for each gas species, in the previous study (Casey et al., 2017)**

| Gas Species | Model Type | Sensor Signal Model Inputs | |
|---|---|---|---|
| $CO_2$ | ANN | eltCO2 | (ELT S300 CO2 Sensor) |
| | | temp | (temperature) |
| | | absHum | (absolute humidity) |
| $O_3$ | ANN | e2vO3 | (e2v MiCs-2611) |
| | | e2vCO | (e2v MiCs-5525) |
| | | e2vVOC | (e2v MiCs-5521) |
| | | figCH4 | (Figaro TGS 2600) |
| | | figCxHy | (Figaro TGS 2602) |
| | | temp | (temperature) |
| | | absHum | (absolute humidity) |

**Table 4**: Relative timing and parameter coverage between model training and test deployment dataset pairs. Incomplete coverage of time occurred if training only took place before or after the test data period and not before and after (pre and post). Incomplete coverage of location occurred when training took place in one location and testing took place in another. Incomplete coverage of parameters, or extrapolation of models, including the target gas mole fraction, temperature, time, and pressure occurred when the values observed during training did not encompass the values observed during testing. Extrapolation in time occurred when training only took place after the test period (post model training timing). Extrapolation in location occurred when a model was trained in one location then applied to data collected in a new location.

| Case Study | Summary | Training Timing | Extrapolation During Test |
|---|---|---|---|
| Dawson Summer 2014 | Urban calibration moved to rural/peri-urban setting | Pre/Post | Location, O |
| SJ Basin Spring 2015 | DJ Basin calibration moved across the state to SJ Basin sampling sites | Post | Location, Pressure, Time |
| SJ Basin Summer 2015 | DJ Basin calibration moved across the state to SJ Basin sampling sites | Pre/Post | Location, Pressure |
| BAO Summer 2015 | DJ Basin calibration applied to same location | Pre/Post | None |
| BAO Summer 2016 | DJ Basin calibration moved 40 miles across the DJ Basin | Post | Location, Time |
| GRET Fall 2016 | DJ Basin calibration applied to same location | Post | Time |
| GRET Spring 2017 | DJ Basin calibration applied to same location | Pre/Post | None |

**Table 5**: O₃ model performance metrics.

| Case Study | N | $R^2$ | RMSE (ppb) | MBE (ppb) | Standard Deviation $R^2$ | Standard Deviation RMSE | Standard Deviation MBE |
|---|---|---|---|---|---|---|---|
| O₃ Models | | | | | | | |
| **Best O₃ Model (Casey et al., 2017)** **ANN with inputs: e2vO3 temp absHum e2vVOC e2vCO FigCH4 FigCxHy** | | | | | | | |
| Dawson Summer 2014 | 1 | 0.83 | 6.46 | -0.91 | 0.00 | 0.00 | 0.00 |
| SJ Basin Spring 2015 | 4 | 0.86 | 7.74 | 3.69 | 0.05 | 3.82 | 5.78 |
| SJ Basin Summer 2015 | 7 | 0.85 | 7.03 | 4.89 | 0.10 | 1.10 | 1.73 |
| BAO Summer 2015 | 2 | 0.93 | 4.26 | 1.45 | 0.00 | 0.31 | 0.07 |
| BAO Summer 2016 | 2 | 0.92 | 12.21 | -11.14 | 0.00 | 0.31 | 0.07 |
| GRET Fall 2016 | 2 | 0.96 | 12.87 | 12.02 | 0.01 | 2.30 | 2.35 |
| GRET Spring 2017 | 2 | 0.98 | 2.59 | 1.49 | 0.00 | 0.69 | 1.02 |
| **Simple Model (Single Gas Sensor)** **LM with inputs: e2vO3 temp absHum** | | | | | | | |
| Dawson Summer 2014 | 1 | 0.95 | 3.59 | -0.46 | 0.00 | 0.00 | 0.00 |
| SJ Basin Spring 2015 | 4 | 0.83 | 17.95 | 16.09 | 0.06 | 6.10 | 5.83 |
| SJ Basin Summer 2015 | 7 | 0.86 | 6.30 | 3.53 | 0.06 | 1.40 | 2.06 |
| BAO Summer 2015 | 2 | 0.87 | 5.50 | 0.94 | 0.00 | 0.78 | 1.56 |
| BAO Summer 2016 | 2 | 0.89 | 5.78 | -2.71 | 0.00 | 0.78 | 1.56 |
| GRET Fall 2016 | 2 | 0.93 | 12.73 | 11.92 | 0.01 | 0.62 | 0.88 |
| GRET Spring 2017 | 2 | 0.89 | 6.00 | -3.19 | 0.00 | 0.73 | 1.38 |
| **Models Optimized For Case Studies** | | | | | | | |
| Dawson Summer 2014 | 1 | 0.95 | 3.59 | -0.46 | 0.00 | 0.00 | 0.00 |
| SJ Basin Spring 2015 | 4 | 0.86 | 7.74 | 3.69 | 0.05 | 3.82 | 5.78 |
| SJ Basin Summer 2015 | 7 | 0.85 | 7.03 | 4.89 | 0.10 | 1.10 | 1.73 |
| BAO Summer 2015 | 2 | 0.93 | 4.26 | 1.45 | 0.02 | 0.51 | 1.54 |
| BAO Summer 2016 | 2 | 0.87 | 6.25 | -0.20 | 0.02 | 0.51 | 1.54 |
| GRET Fall 2016 | 2 | 0.95 | 3.99 | 2.14 | 0.00 | 0.28 | 0.89 |
| GRET Spring 2017 | 2 | 0.98 | 2.59 | 1.49 | 0.00 | 0.69 | 1.02 |

**Table 6**: $CO_2$ model performance metrics.

| Case Study | N | $R^2$ | RMSE (ppm) | MBE (ppm) | Standard Deviation $R^2$ | Standard Deviation RMSE | Standard Deviation MBE |
|---|---|---|---|---|---|---|---|
| \multicolumn CO$_2$ Models | | | | | | | |
| **Best CO$_2$ Model from (Casey et al., 2017) ANN with inputs: eltCO2 temp absHum** | | | | | | | |
| SJ Basin Summer 2015 | 2 | 0.65 | 8.42 | -0.62 | 0.00 | 1.81 | 1.41 |
| BAO Summer 2015 | 2 | 0.75 | 9.98 | -2.60 | 0.05 | 13.00 | 13.89 |
| BAO Summer 2016 | 2 | 0.69 | 54.38 | 48.37 | 0.05 | 13.00 | 13.89 |
| GRET Fall 2016 | 2 | 0.74 | 42.37 | 39.58 | 0.02 | 2.44 | 2.57 |
| GRET Spring 2017 | 2 | 0.83 | 6.31 | 0.59 | 0.03 | 0.13 | 2.61 |
| **Simple Model (Single Gas Sensor) LM with inputs: eltCO2 temp absHum** | | | | | | | |
| SJ Basin Summer 2015 | 2 | 0.71 | 7.84 | 0.27 | 0.01 | 1.43 | 0.42 |
| BAO Summer 2015 | 2 | 0.69 | 10.62 | -1.26 | 0.06 | 1.52 | 10.67 |
| BAO Summer 2016 | 2 | 0.73 | 11.82 | 0.73 | 0.06 | 1.52 | 10.67 |
| GRET Fall 2016 | 2 | 0.82 | 8.62 | -3.46 | 0.00 | 0.69 | 1.45 |
| GRET Spring 2017 | 2 | 0.55 | 9.88 | -0.33 | 0.03 | 0.29 | 1.91 |
| **Models Optimized For Case Studies** | | | | | | | |
| SJ Basin Summer 2015 | 2 | 0.72 | 7.45 | -0.11 | 0.04 | 2.06 | 0.31 |
| BAO Summer 2015 | 2 | 0.80 | 8.85 | -2.29 | 0.10 | 6.47 | 7.08 |
| BAO Summer 2016 | 2 | 0.73 | 11.82 | 0.73 | 0.06 | 1.52 | 10.67 |
| GRET Fall 2016 | 2 | 0.82 | 8.62 | -3.46 | 0.00 | 0.69 | 1.45 |
| GRET Spring 2017 | 2 | 0.83 | 6.31 | 0.59 | 0.03 | 0.13 | 2.61 |

**Table 7**: **Relative benefits associated with the implementation of different models for O$_3$ and CO$_2$.**

| Case Study | Mean % Increase in R$^2$ | Mean % Decrease in RMSE | Mean % Decrease in MBE | Mean % Increase in R$^2$ | Mean % Decrease in RMSE | Mean % Decrease in MBE |
|---|---|---|---|---|---|---|
| | CO$_2$ Models | | | O$_3$ Models | | |
| **Benefit of Models Optimized for Case Studies Over the Best Models from (Casey et al., 2017)** | | | | | | |
| Dawson Summer 2014 | | | | 14.51 | 44.42 | 50.00 |
| SJ Basin Spring 2015 | | | | 0.00 | 0.00 | 0.00 |
| SJ Basin Summer 2015 | 10.56 | 11.52 | 82.60 | 0.00 | 0.00 | 0.00 |
| BAO Summer 2015 | 5.84 | 11.27 | 11.95 | 0.00 | 0.00 | 0.00 |
| BAO Summer 2016 | 5.72 | 78.27 | 98.49 | -5.01 | 48.82 | 98.19 |
| GRET Fall 2016 | 11.17 | 79.66 | 108.73 | -0.54 | 68.99 | 82.22 |
| GRET Spring 2017 | 0.00 | 0.00 | 0.00 | 0.00 | 0.00 | 0.00 |
| **Benefit of the Best Models from (Casey et al., 2017) Over Simple Linear Models** | | | | | | |
| Dawson Summer 2014 | | | | -12.67 | -79.92 | -99.99 |
| SJ Basin Spring 2015 | | | | 3.20 | 56.88 | 77.09 |
| SJ Basin Summer 2015 | -8.41 | -7.29 | 331.39 | -1.34 | -11.53 | -38.41 |
| BAO Summer 2015 | 8.70 | 6.05 | -106.48 | 6.79 | 22.48 | -53.85 |
| BAO Summer 2016 | -5.41 | -360.09 | -6543.84 | 2.57 | -111.22 | -310.71 |
| GRET Fall 2016 | -10.05 | -391.73 | 1244.99 | 2.88 | -1.12 | -0.86 |
| GRET Spring 2017 | 51.92 | 36.13 | 278.55 | 10.00 | 56.90 | 146.65 |
| **Benefit of Models Optimized for Case Studies Over Simple Linear Models** | | | | | | |
| Dawson Summer 2014 | | | | 0.00 | 0.00 | 0.00 |
| SJ Basin Spring 2015 | | | | 3.20 | 56.88 | 77.09 |
| SJ Basin Summer 2015 | 1.26 | 5.06 | 140.25 | -1.34 | -11.53 | -38.41 |
| BAO Summer 2015 | 15.04 | 16.64 | -81.80 | 6.79 | 22.48 | -53.85 |
| BAO Summer 2016 | 0.00 | 0.00 | 0.00 | -2.57 | -8.10 | 92.59 |
| GRET Fall 2016 | 0.00 | 0.00 | 0.00 | 2.33 | 68.64 | 82.07 |
| GRET Spring 2017 | 51.92 | 36.13 | 278.55 | 10.00 | 56.90 | 146.65 |

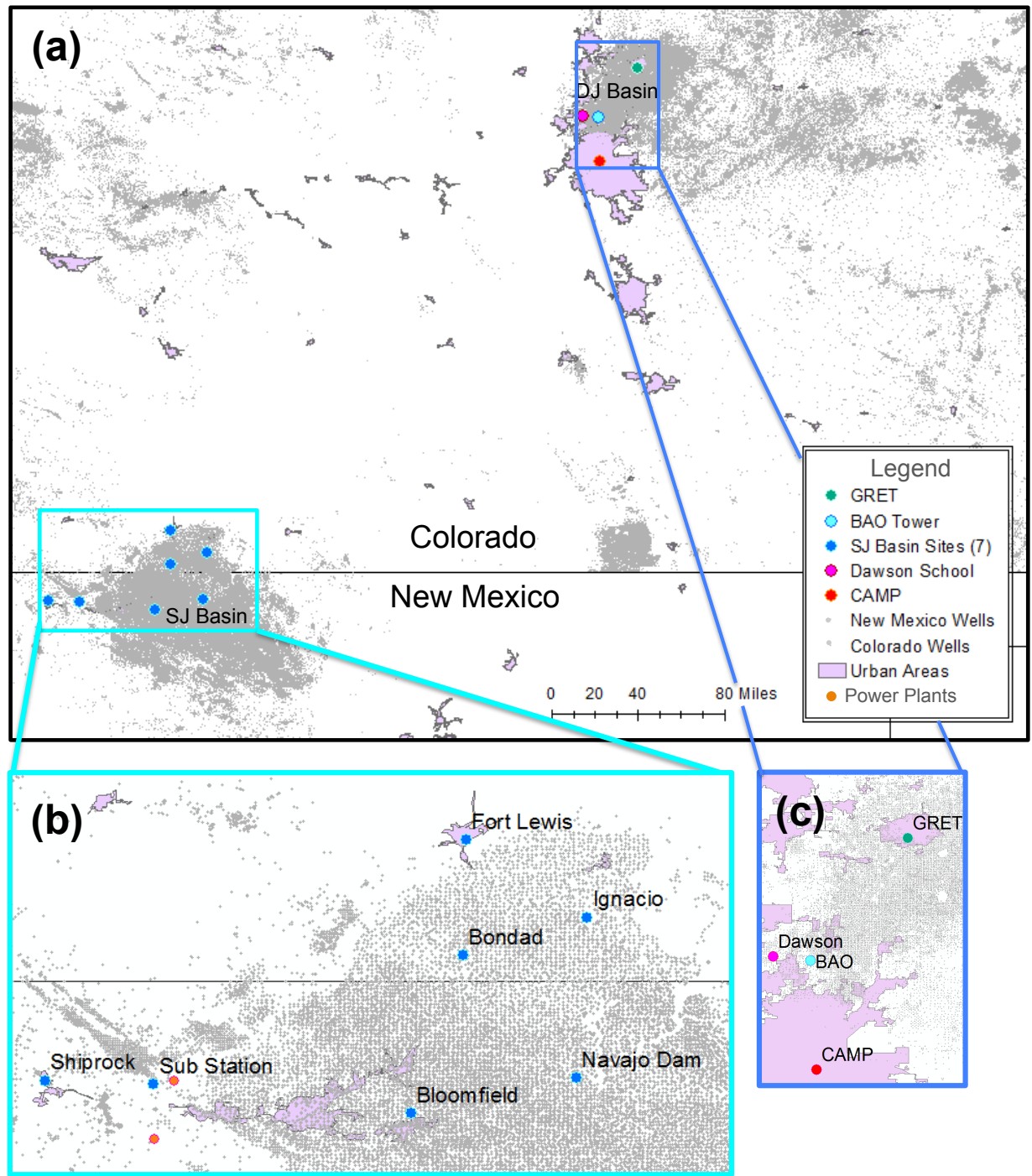

Figure 1: (a) Training and test deployment locations are identified in the SJ and DJ Basins in context with urban centers and oil and gas production wells. (b) Panel zoomed in on the SJ Basin, covering approximately 4,250 square miles (85x50 miles). (c) Panel zoomed in on the DJ Basin covering approximately 1,540 square miles (28x55 miles).

**(a)**

| Case Study | Year | Model Training and Test Deployment Timelines |
|---|---|---|
| | | Jan Feb Mar Apr May Jun Jul Aug Sep Oct Nov Dec |
| Dawson Summer 2014 | 2014 | |
| SJ Basin Spring 2015 | 2015 | |
| SJ Basin Summer 2015 | 2015 | |
| BAO Summer 2015 | 2015 | |
| BAO Summer 2016 | 2016 2017 | |
| GRET Fall 2016 | 2016 2017 | |
| GRET Spring 2017 | 2017 | |

**(b)**

| Case Study | Training Location | Test Location | O₃ # U-Pods | O₃ U-Pod Names | CO₂ # U-Pods | CO₂ U-Pod Names |
|---|---|---|---|---|---|---|
| Dawson Summer 2014 | CAMP | Dawson | 1 | BE | NA | NA |
| SJ Basin Spring 2015 | BAO | SJ Basin | 4 | BB, BD, BF, BJ | NA | NA |
| SJ Basin Summer 2015 | BAO | SJ Basin | 7 | BA, BB, BD, BE, BF, BH, BI | 2 | BB, BD |
| BAO Summer 2015 | BAO | BAO | 2 | BC, BJ | 2 | BC, BJ |
| BAO Summer 2016 | GRET | BAO | 2 | BH, BI | 2 | BH, BI |
| GRET Fall 2016 | GRET | GRET | 2 | BH, BI | 2 | BH, BI |
| GRET Spring 2017 | GRET | GRET | 2 | BH, BI | 2 | BF, BI |

**Figure 2**: **(a) ANN and LM training and test deployment timelines. The Dawson, BAO, and GRET sampling sites are all located in the DJ Basin. Model training periods for each test deployment are shown in blue, and model test periods are shown in magenta. For the BAO Summer 2016 case study, the period outlined in blue shows data that was used to train O₃ model, but not CO₂ models since CO₂ reference data was not available during winter months. (b) Information about each of the case studies presented in the above timelines, including model training and testing locations, as well as the number and names of U-Pods included in each case study for both O₃ and CO₂ models. The U-Pods with names shown in grey were constructed and deployed starting in May of 2014. The U-Pods with names shown in black were constructed and deployed starting in April of 2015.**

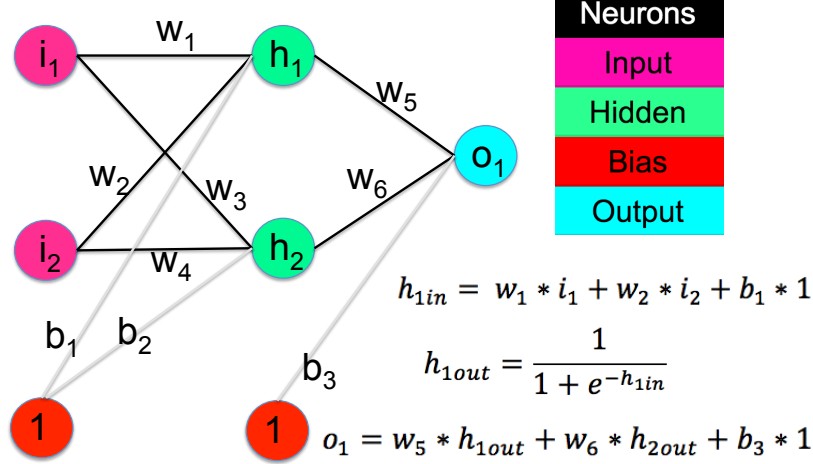

$$h_{1in} = w_1 * i_1 + w_2 * i_2 + b_1 * 1$$

$$h_{1out} = \frac{1}{1 + e^{-h_{1in}}}$$

$$o_1 = w_5 * h_{1out} + w_6 * h_{2out} + b_3 * 1$$

**Figure 3.** **Example of a simple feed forward neural network, showing how inputs are propagated through the network during each of the training iterations (Casey et al., 2017)**

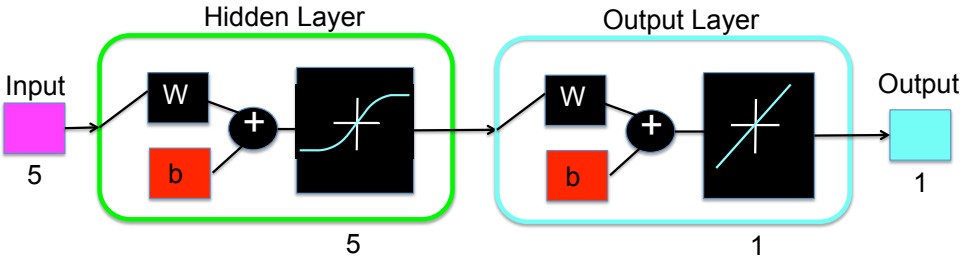

**Figure 4. Diagram of an example ANN with the same color-coded components as are presented in Figure SM3 in section 2.2 of the SM. This ANN has 5 inputs, 1 hidden layer with 5 tansig hidden neurons, and 1 linear output layer leading to 1 output. The network is fully connected with weights and biases (Casey et al., 2017).**

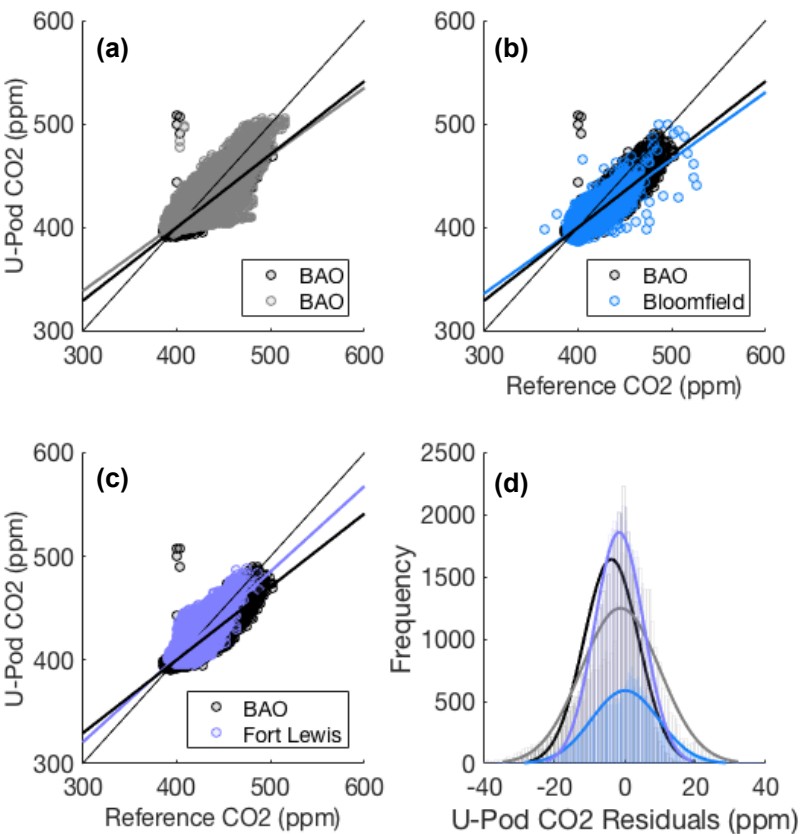

**Figure 5**: Scatter plots of U-Pod $CO_2$ vs. reference $CO_2$ and overlaid histograms of U-Pod $CO_2$ residuals for (a) BAO and BAO (b) BAO and Bloomfield (c) BAO and Fort Lewis. A 1:1 single-weight reference line is included in each scatter plot along with double-weight lines of best fit for U-Pods at each sampling location. Data from U-Pod BC at BAO is plotted in black along with U-Pods BJ, BB, and BD at BAO, Fort Lewis, and Bloomfield, respectively. Sensor signal inputs include eltCO2, temp, and absHum. (d) Overlaid histograms of model residuals with respect to reference $CO_2$.

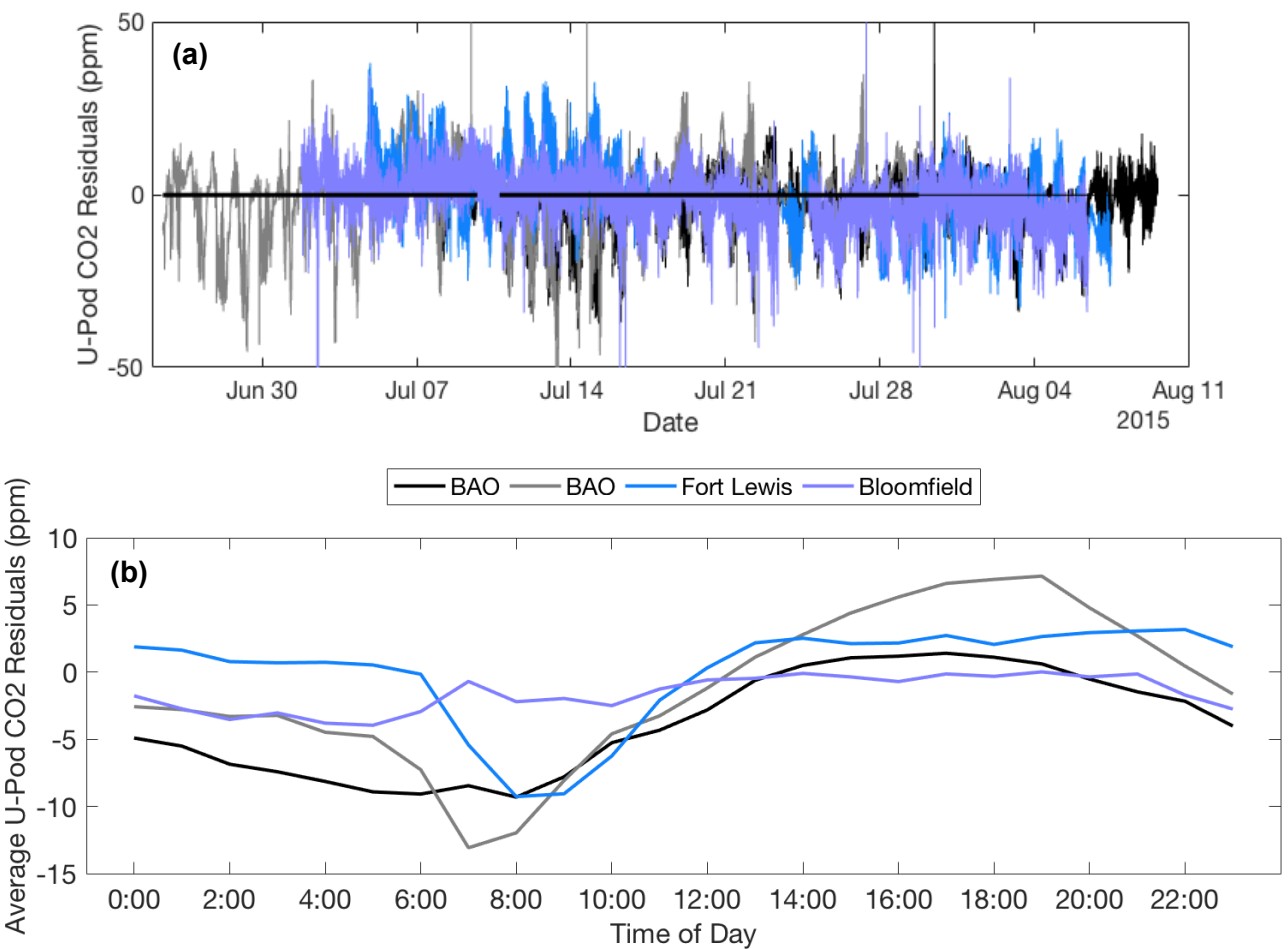

**Figure 6**: U-Pod CO$_2$ residuals by (a) data and (b) time of day and throughout the duration of the deployment period. Sensor signal inputs include eltCO2, temp, and absHum.

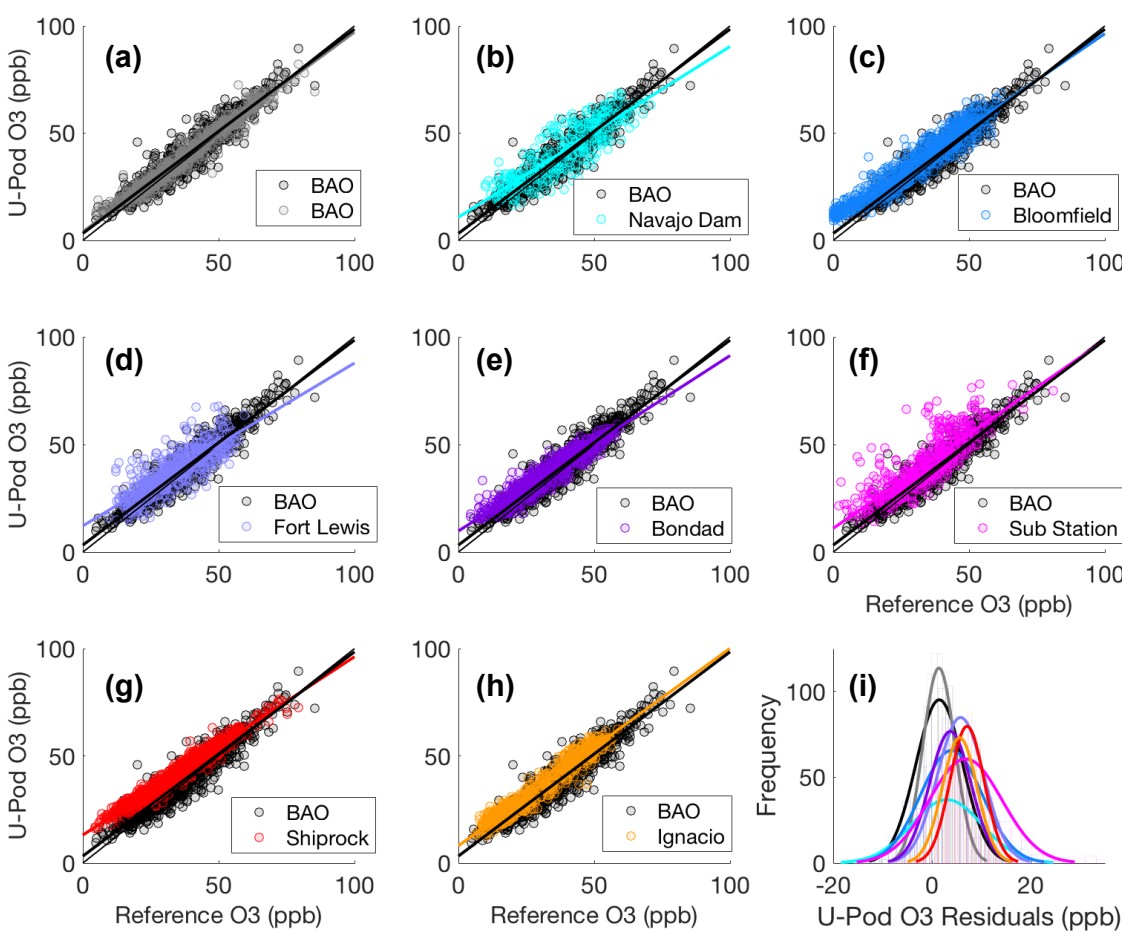

**Figure 7**: **Scatter plots of U-Pod vs reference O₃, comparing U-Pod BC at BAO, in black, with (a) U-Pod BJ at BAO (b) U-Pod BA at Navajo Dam (c) U-Pod BB at Fort Lewis (d) U-Pod BD at Bloomfield (e) U-Pod BE at Bondad (f) U-Pod BF at the Sub Station (g) U-Pod BH at Shiprock and (h) U-Pod BI at Igniacio. (i) Overlaid histograms of model residuals with respect to reference O₃.**

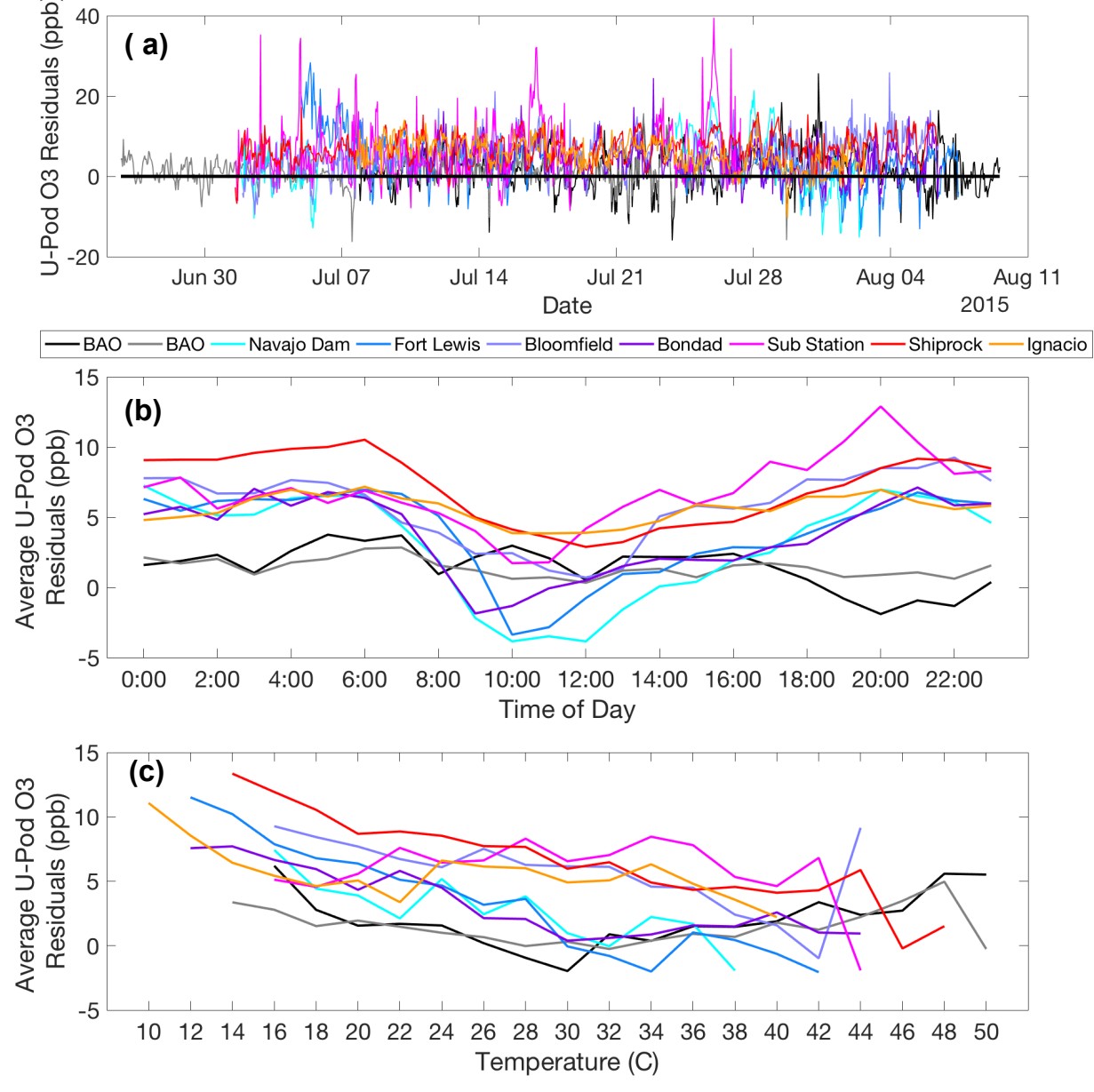

**Figure 8**: Residuals of U-Pod O₃ spanning of the deployment period, by (a) date (b) time of day and (c) temperature.

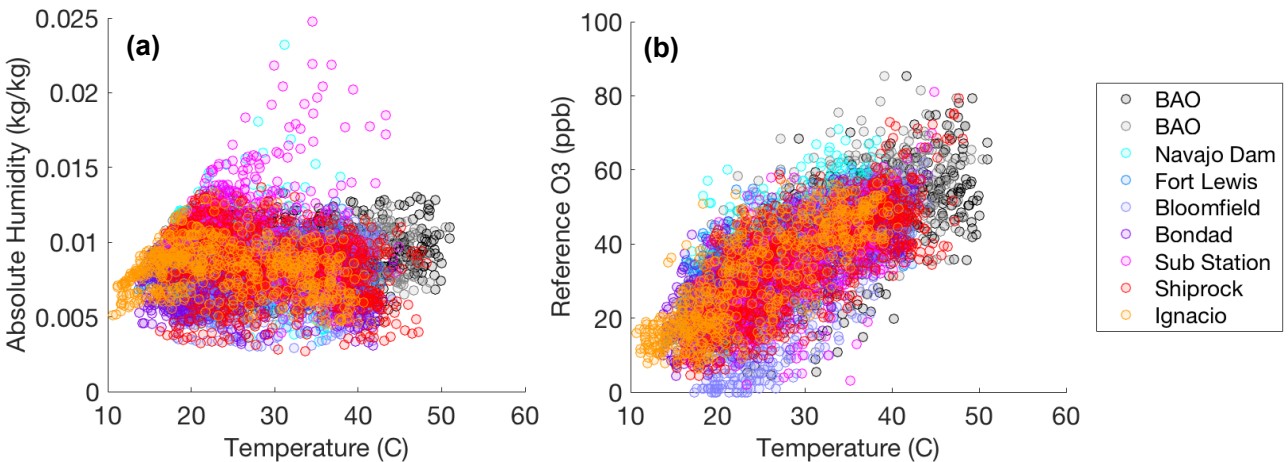

**Figure 9**:  **Scatter plots showing the combined parameter space of (a) absolute humidity with temperature and (b) reference O$_3$ with temperature for each of the U-Pod sampling sites at BAO and the SJ Basin.**

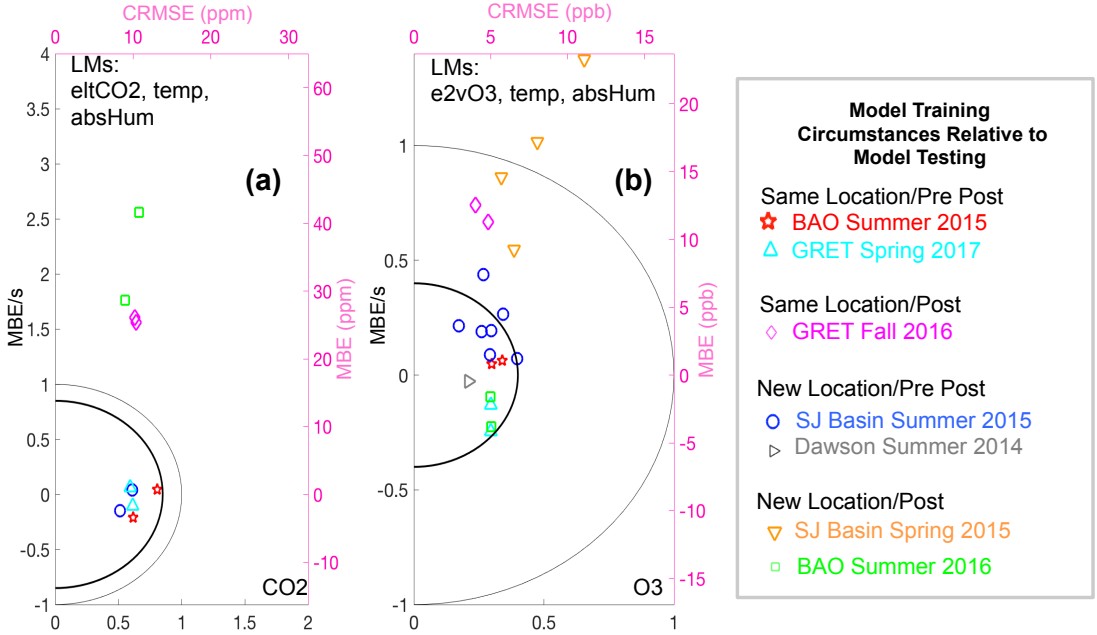

**Figure 10**: **Target diagrams demonstrating performance of a previously determined best-performing model across all new test datasets. (a) CO₂ and (b) O₃ LM performance when only the primary gas sensor, temperature and humidity are inputs. (c) CO₂ and (d) O₃ ANN performance with inputs that were found to perform best at the GRET site in the spring of 2017** (Casey et al., 2017). **Model input definitions: eltCO2 (ELT S300 CO₂ sensor), e2vO3 (e2v MiCs-2611 sensor), temp (temperature), and absHum (absolute humidity).**

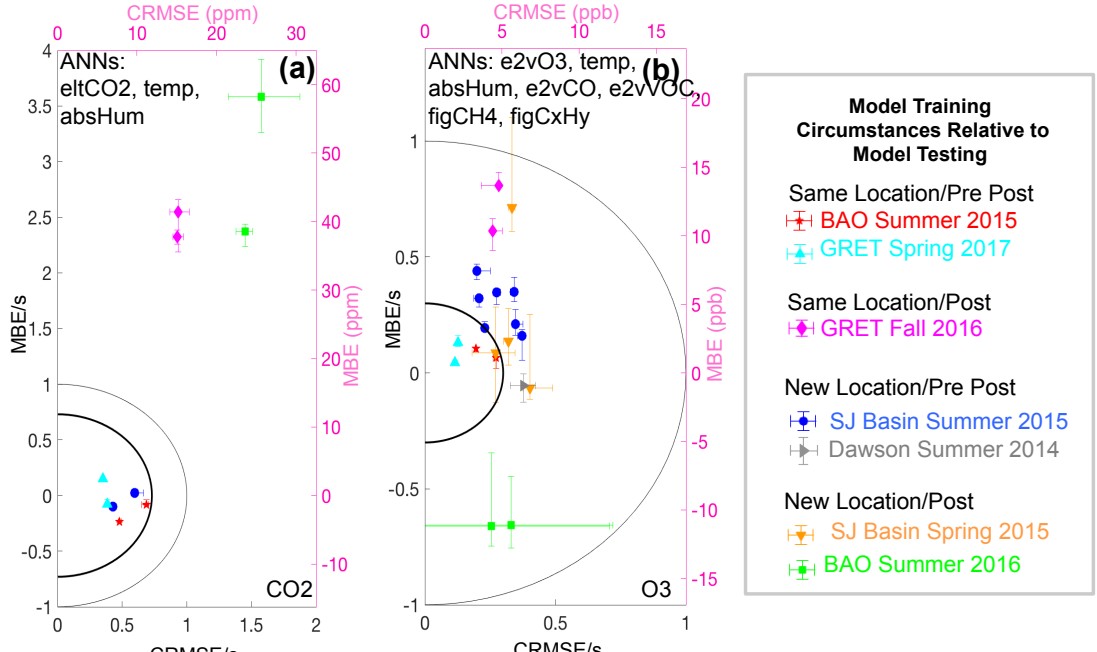

**Figure 11**: **Target diagrams demonstrating performance of a previously determined best-performing model across all new test datasets (a) CO$_2$ and (b) O$_3$ ANN performance with inputs that were found to perform best at the GRET site in the spring of 2017** (Casey et al., 2017). **Model input definitions: eltCO2 (ELT S300 CO$_2$ sensor), e2vCO (e2v MiCs-5525 sensor), e2vVOC (e2v MiCs-5521 sensor), e2vO3 (e2v MiCs-2611 sensor), figCH4 (Figaro TGS 2600 sensor), figCxHy (Figaro TGS 2602 sensor), temp (temperature), and absHum (absolute humidity).**

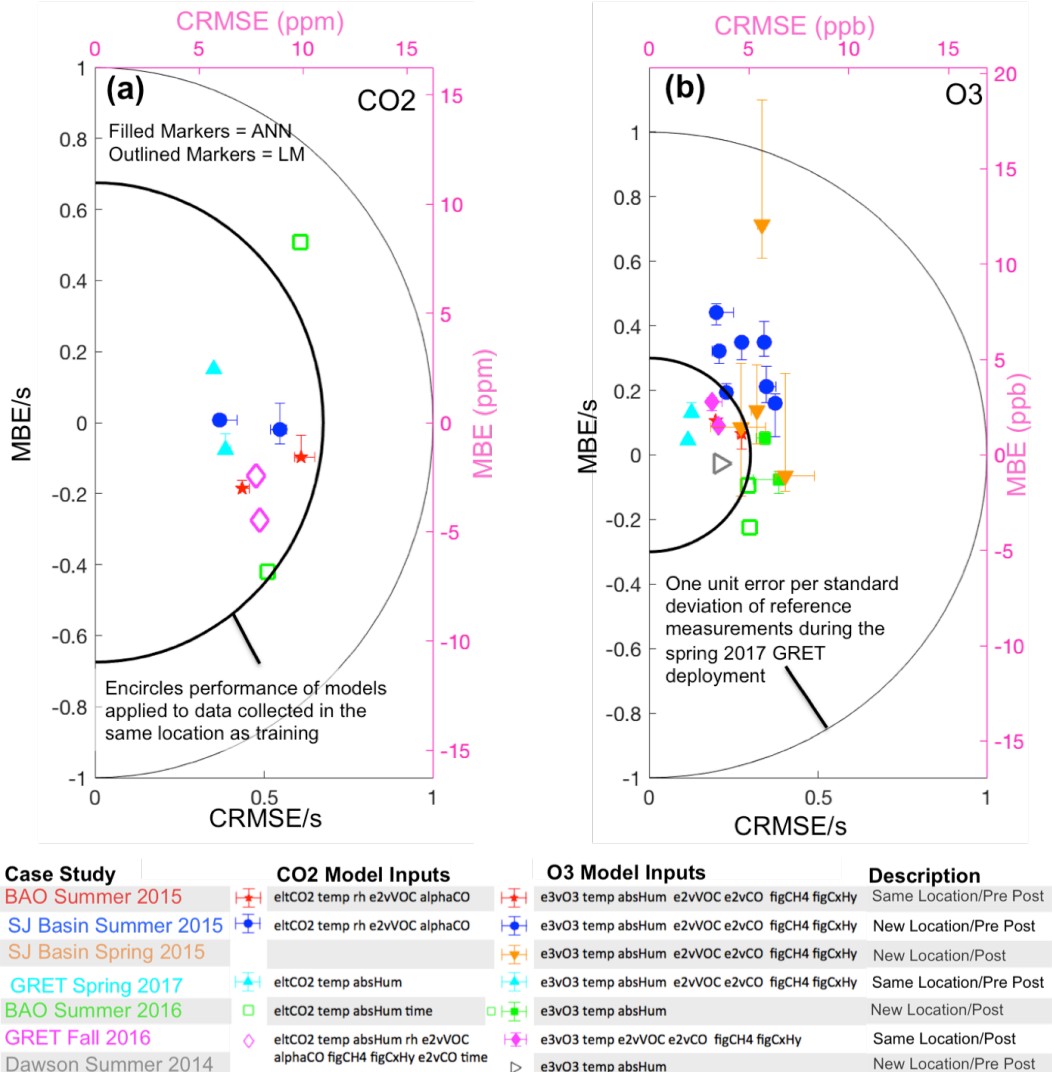

**Figure 12:** Target diagrams for (a) $CO_2$ and (b) $O_3$ calibration model performance for the best performing model for each particular case when tested on data from a number of field deployments. Model input definitions: eltCO2 (ELT S300 CO2 sensor), e2vCO (e2v MiCs-5525 sensor), e2vVOC (e2v MiCs-5521 sensor), e2vO3 (e2v MiCs-2611 sensor), figCH4 (Figaro TGS 2600 sensor), figCxHy (Figaro TGS 2602 sensor), alphaCO (Alphasense CO-B4 sensor) temp (temperature), absHum (absolute humidity), rh (relative humidity), and time (absolute time).