# Peer review of "Testing the performance of field calibration techniques for low-cost gas sensors in new deployment locations: across a county line and across Colorado"

_Atmospheric Measurement Techniques, 2018_

## Referee Comment (RC1) · Anonymous Referee #1 · 19 Jun 2018

Please see attachment for review comments.

Please also note the supplement to this comment:
https://www.atmos-meas-tech-discuss.net/amt-2018-81/amt-2018-81-RC1-supplement.pdf

---

## Referee Comment (RC2) · Anonymous Referee #2 · 7 Aug 2018

**Testing the performance of field calibration techniques for low-cost gas sensors in new deployment locations: across a county line and across Colorado**

Joanna Gordon Casey[1], Michael P. Hannigan[1]

5  [1]Department of Mechanical Engineering, University of Colorado at Boulder, Boulder, 80309, United States of America

*Correspondence to*: Joanna Gordon Casey (joanna.casey@colorado.edu)

Review -

Casey and Hannigan explore the spatial and temporal transferability of field calibration models (specifically linear models (LMs) and artificial neural networks (ANNs)) for two sensors, O3 (e2vO3) and CO2 (eltCO2), reported by the integrated U-POD sensor package. By 'spatial/temporal transferability' they mean a determination as to whether a calibration model trained from sensor co-location (with reference instrumentation measuring target species) at one location works effectively when that same sensor system is then deployed at a different location. As the authors point out, changing the micro-environment (and local air pollution source contributions to that unique environment) may pose additional complications/challenges when trying to reconcile quantitative measurements with low-cost sensors. The authors make some attempt to separately describe temporal and spatial extension to better understand whether time-alone undermines the accuracy of the calibration models or change of location.

While the topic of sensor calibration and extension of calibration models across a diverse set of deployment scenarios is of fundamental importance to the field of low-cost AQ sensing, the paper, as written, largely fails to pull together a coherent narrative from which active participants in the low-cost AQ measurement space could easily glean useful, actionable information. To be clear, the topic of sensor quantification is inherently complex, and the authors undertake an ambitious analysis spanning 3 years of data from 10 U-POD systems deployed across 4 micro-environments. There are important lessons to be learned from their efforts, but at present these lessons are not brought to the fore of the paper and as a result are easily lost to the reader.

Throughout the manuscript the authors refer back to their published work (Casey et al., 2017). In the vast majority of instances in which this reference is provided, there is little to no contextual detail explicitly drawing the lines of connectivity between the current work and the previous work. Seeking out the exact evidence that exists in the earlier work and relating its relevance to the current work is left entirely up to the reader. Overall, this referencing needs to be done in a manner that is not vague and does not require that the reader be intimately familiar with the previous work. The paper would also be strengthened if the unique and novel insights that result from the current work were more clearly differentiated from the Casey et al., 2017 effort.

There are seemingly contradictory statements throughout the text. These tend to originate from the authors' desire to provide a clear-cut answer as to whether or not a given model 'worked' in a given case study under a given environmental sampling condition. The fact of the matter is, low-cost AQ sensor quantification is extremely convoluted and often times the validity of data can be somewhat ambiguous. Faced with this level of complexity, the current manuscript fails to provide a succinct and systematic

evaluation/reporting approach, and as such main (and important) take-home lessons from their work are lost.

Specific comments:

[revised manuscript text omitted]

- L11: What is the difference between supervised learning methods and ANNs? This warrants a more detailed description / definition.
- L15: This sentence (bracketed in red) - is very important, but also very wordy and hard to follow. Related to this assertion, it is not clear how the authors disentangle the temporal and spatial domain from one another, particularly the temporal domain. Time-decay patterns in the data are going to be present whether or not the sensor system has been moved to a different location. How would one ascribe difference in that case to a spatial domain and not temporal domain?
- 'hold up' this language is too casual and used throughout the text. Consider re-wording.

Section 1.2

- L28: 'A number of enclosures..' define the number.
- If Casey et al., 2017 demonstrated the ANN results for CO2 and O3 in the Spring of 2017 in Greeley, CO; is that same data being presented as a portion of this paper (as Figure 1 suggests).
- The concluding sentences of this section nicely frame the motivation/need for the current work, consider bringing this to the fore of the paper / abstract, etc.

Section 1.3

- Final sentence: It's unclear why, if all of the U-POD sensor systems were equipped to measure CO and CH4 alongside CO2 and O3, analogous training/test matrix pairs are unavailable for these other species.

Section 1.4

- L20: 'Very high levels of ozone' – specify the actual concentration or concentration range
- L23: 'a modeling study' – is there really only one modeling study that shows this?
- Final sentence: 'pooling' avoid using words with common association different from the intended meaning. Consider re-wording. 'accumulating'?

Section 2.1

- L5: "with a number of low-cost gas sensors" – specify the actual number of sensors integrated in each U-POD

**2.2 Deployment Locations and Timelines**

10 These ten U-Pods were deployed at a number of sampling sites in and around the DJ and SJ Basins over the course of several years, from 2014 - 2017. Deployments generally consisted of co-location with reference measurements prior to and following a period of spatially distributed measurements. During some the distributed measurement periods, a subset of U-Pods remained co-located with reference instruments where the field calibrations took place. During other distributed measurement periods, U-Pods were deployed in new locations that were equipped with reference measurements. We

15 opportunistically employ data from a number of these sensor deployments, treating them as case studies in order to characterize the performance of field calibration models when they are extended to new locations. For each case study, data was divided into training and test periods.

Table 2 lists the $O_3$ and $CO_2$ reference instruments that were co-located with U-Pods at each sampling site, along with

20 instrument operators, calibration procedures, and reference data time resolution. The first distributed measurement campaign took place during the summer of 2014 when five U-Pods were sited at locations around Boulder County, with four distributed along the eastern boundary of the county, adjacent to Weld County where dense oil and gas production activities were underway. A background site, further from oil and gas production activities was also included to the west, near a busy traffic intersection on the north end of the City of Boulder. Co-locations with reference measurements toward field

25 calibration of sensors took place at the Continuous Ambient Monitoring Program (CAMP) Colorado Department of Health and Environment (CDPHE) air quality monitoring site in downtown Denver. One of the distributed sampling sites, Dawson School, was also equipped with an optical $O_3$ instrument, operated by Detlev Helmig's research group from the Institute for Artic and Alpine Research (INSTAAR). This study was funded by Boulder County with the combined aims of gaining a better understanding of how oil and gas emissions affect air quality in Boulder County and learning more about how low-

30 cost air quality measurement methods can help inform air quality in this context.

- The authors identify that 10 U-POD systems were used in the previous and current work, but the vast majority of case studies (outlined in Figure 1.) utilize just 2 U-PODs at each location. The authors need to more clearly describe in the text how the U-PODs were distributed throughout the work and whether all 10 U-PODs used in the current work had the same characteristic O3 and CO2 response when measuring the same air. The sensor system age (time since manufacture date) and environmental-hysteresis (lifetime environmental exposure of a given UPOD system) is not mentioned anywhere in the text. Do these factors not matter when analyzing the temporal extension of a given calibration model? When considering the fundamental measurement principles of these particular gas sensors, does degradation occur due to gradual (or rapid) deposition of material onto active catalytic sites within the sensors? If so, then the age of a given sensor and what's it's been exposed to over its lifetime, ought to factor in.. or at least deserve a mention.
- The explanation of the training vs test sampling periods is confusing as written. Given the nature of the experiment, doesn't each UPOD system have to be co-located with reference

instrumentation for the full duration of the period of study?  It sounds as though the authors aimed to bookend the distributed network measurements ('testing period' with a period of co-location at a reference site in the general vicinity of the deployment ('training period') – but in order evaluate their models, they would have to retain a co-located reference measurement of O3 and CO2 at all times in all locations.  Looking at the deployment timelines displayed in Figure 1, it is also evident from the Figure (but not from the text) that the vast majority (~75% or greater) of the total deployment time was used to train the nodes not test the resultant calibration models (~25% of the total time).  These train-to-test ratios appear to undermine the general applicability of the models to longer duration, distributed sensor measurements in which no co-located reference measurements are available.  The authors should make an effort to bridge the gap between how they were able to execute their experiments and how distributed low-cost AQ sensor systems will ultimately be deployed.

- Highlighted in passage above:
  - L10: define the number of sampling sites.  Eliminate vague language in the text.
  - L15: same comment.
  - L21: 5 UPOD systems are purportedly used in the Boulder / CAMP 2014 work.  Figure 1 lists 1 UPOD system as being active during that test.  Reconcile this.
  - L27: Identify the actual ref. O3 measurement in the text here
  - Last sentence:  is this relevant to the current paper/study?  Not clear what 'study' the authors are referring to in this sentence.

[revised manuscript text omitted]

- L9:  The authors claim that the SJ Basin network was similarly executed for the DJ Basin. DJ Basin is absent from Figure 1., replaced presumably by BAO. It is unclear how many UPODs were deployed to the DJ Basin. It's very confusing trying to track in time and location the distribution of the 10 UPODs. If I try and decipher the information in Figure 1, either 2 or 4 UPOD units were deployed to the DJ Basin, which on the face of it, does not constitute a similar network deployment of 10x UPODs deployed to the SJ Basin (although, it seems that only 4 and/or 7 UPOD units were deployed to the SJ Basin..
- L13: The authors identify the BAO site as the relevant co-location site for the DJ Basin-deployed UPODS, but then point out that there were NO co-located reference instrumentation accessible for any of the distributed sampling sites. What does this mean for evaluating / testing their models in the distributed network application?
- L14-16:The authors state the GRET site housed all 10x UPOD systems for a year, but Figure 1 indicates that only 2-6? UPOD systems were used at this location and only for shorter periods of time. Again, the text is extremely hard to follow and the information in Figure 1 does not make it any clearer.
- L26:  The only metal oxide sensor that's relevant to the current work is the e2vO3 sensor. The operational fundamentals of this sensor should be described: the raw signal processing, circuitry

considerations, and known theoretical operational conditions that undermine the sensitivity, selectivity, and/or stability of the e2vO3 metal oxide sensor.
- L29: 'in a few' Quantify the number of UPODs with faulty RH sensors
- L31: 'nearby': Define the exact position relative to the faulty UPOD

When the U-Pods were initially deployed at the GRET site, on August 23$^{rd}$ of 2016, the RH sensors in all ten U-Pods malfunctioned, logging an error code of -99 instead of the relative humidity. This malfunction seemed to coincide with the implementation of radio communication from each U-Pod to a central node in an effort to reduce trips to the field site to download data and to identify issues with data acquisition promptly. RH signals in the U-Pods began logging correctly again

5   in November when we stopped remote communication. We replaced RH values for the U-Pods during this time period by utilizing data from the Picarro Cavity Ring-Down Spectrometer that was co-located at GRET with the U-Pods. Water mole fractions measured by the Picarro were converted into mass-based mixing ratios to match the units of the absolute humidity signal in the U-Pod data. We then replaced the absolute humidity signal in each U-Pod from August 23$^{rd}$ through October 1$^{st}$ in 2016 with the mixing ratios derived from Picarro measurements. Using the temperature and pressure logged in each U-

10   Pod along with the absolute humidity from the Picarro, relative humidity was calculated for each U-Pod during this period.

To perform regressions toward field calibration of sensors, the reference and U-Pod data needed to be aligned. When reference measurements with minute time resolution were available for both training and corresponding testing periods, minute median data from the U-Pods were used. Medians were used as opposed to averages in order to reduce the potential

15   influence of sensor noise as well as to remove short duration spikes in the reference and sensor data that resulted from air masses that may not have been well mixed across the reference instrument inlets and the U-Pod enclosures. When reference data were instead available with only 5-minute or 60-minute time resolution, U-Pod medians were calculated for the same time step. Medians were also calculated for reference measurements with finer time resolution to match the time resolution of corresponding training/testing data. The first 15 minutes of data after any period that the U-Pods had not recorded data for

20   the previous 5 minutes was removed in order to filter transient behavior associated with sensor warm-up.

- Did the implementation of radio communication for the UPODs have any impact on any of the other measurements in the system, beyond RH?
- At the beginning of the paragraph, the authors state that the radio communication was active until November, but the substitute RH values from the Picarro were only applied up to October 1 (later part of the paragraph). This is confusing.
- Generally speaking, faulty or absent RH measurements on-board the UPOD (or any low-cost AQ sensor system that suffers from environmental interference) is a potentially widespread issue across the emerging field. I think the authors missed an opportunity to discuss their work-around in more detail and comment on the importance of maintaining stable RH measurements within any given low-cost AQ sensor system.
  - The completely unusable radio communication RH values and the drifting RH values mentioned in section 2.3 beg the question – do the authors think this is a failure on the RHT component itself or the circuitry of the UPODs. Again, if the evidence suggests the former, that is useful empirical data for others in the field.
  - Where is RH measured specifically within each UPOD. Is the measurement internal to the box or positioned in a manner to provide a true ambient RH measurement? What are

the implications of using alternative RH data sources that are not on-board the same UPOD?

- If median values were used for the co-located reference instruments, but the data from those instruments was 1-min averages, how did the authors obtain reference measurement medians at 1-min (the vast majority of temporal resolution used in the current work).
- L19: What % of the total data used in training/testing each UPOD was removed due to this 5-min null data condition?

**Section 2.4**

- L32 'using methods described previously', given the importance of the LMs and ANNs in the current work, each model should be described in more detail in the manuscript.
- P7L6 – need reference for Bayesian Regularization
- The concepts of early stopping, hidden neurons, and hidden layers need to be described

**3 Results and Discussion**

To evaluate the performance of each of the ANN and LM models that were generated using training data then applied to test datasets, we used residuals, the coefficient of determination ($r^2$), root mean squared error (RMSE), mean bias error (MBE),

15 and centered root mean squared error (CRMSE). The CRMSE is an indicator of the distribution of errors about the mean, or the random component of the error. The MBE, alternatively, is an indicator of the systematic component of the error. The sum of the squares of the CRMSE and the MBE is equal to the square of the total error, the square root of which is defined by the RMSE. As in our previous work (Casey et al., 2017), we compared performance of LMs and ANNs with a number of different sets of inputs for each train/test data pair. The $r^2$, RMSE, and MBE for each of these alternative models when

20 applied to test data are presented in the supplemental materials (SM) in Fig. S2 through Fig. S7, along with representative scatter plots and time series comparing the performance LMs and ANNs for a given set of inputs. In Fig. S2 through Fig. S7, the best performing model inputs for each train/test data pair are shaded in purple. The type of model that performed the best (ANN vs. LM) is indicated in the caption of each figure. Presented below is an analysis and comparison of the best-performing model for each species as determined in our previous work, as well as performance metrics for the best

25 performing model associated with each new training/testing dataset pairs described in section 2.2.

- Highlighted sentence is confusing as written. How can there be multiple 'best' preforming models?
- Does section 2.2 really succinctly describe each training/testing dataset pair?

**Section 3.1**

- This is the first place in the text of the manuscript where the limited extent of co-location upon distributed field deployment is described and how the 10 UPODs are reconciled against such limitations.
- For the purposes of the current study, if there is no co-location with reference, is it still a relevant data point? Can the authors effectively 'test' their model under these circumstances?

- This section P8L3 is also the first mention of reducing/oxidizing interfering gas species – this potential deserves a more detailed explanation in the context of the specific micro-environment source contributions
- The overall discussion of factors impacting differences between the two Basin deployments is fairly scattered. It would be more beneficial to the reader if the authors could draw more specific lines of connectivity between environmental or pollution source contributions and the robustness (or lack of robustness) in the model.

[revised manuscript text omitted]

Highlighted above:
- L6-9:  Discussion is confusing and language is too casual:  "did not make a big difference" – too vague.  Quantify based on the statistical analysis of the model test data.  When considering the benefit of including extra sensor inputs in the training matrix for their models, again the Authors are drawing comparisons to their earlier work (Casey et al, 2017) but it's not really clear how this improves/informs the current work – besides stating that the inclusion of the parameters didn't make the data product worse.
- L10 e2vCO2 does not exist as a sensor metric in the UPODs.
- L15:  'all the UPODS'  how many is this again?

- L19: 'For a number of UPODs': state the number.
- L21: 'for some of the sites..': which sites?
- L15-22: this paragraph seems to say that the ANN training matrix determined to be optimal in Casey et al., 2017 was also found to be optimal in the current work, with inclusion of all peripheral sensors to the input training matrix for O3. But they also state that the LMs data products were just as good (or better) when compared to the ANN models. This result seems important, but not really discussed further. The results are left vague. Conclusions as to why this might be the case are absent.
- L27: 'had bad RH data' – as noted in a section that doesn't exist. What is bad RH data?
- L29: 'relatively far away' – how far? Again. These details matter.
- L30-31: Apparently one of the major results from Casey et al., 2017 is an extreme sensitivity to RH when using ANN's to quantify O3. Given the failure of the RH sensor throughout much of the work presented in the current work, it seems critically important that this RH-sensitivity be discussed in much greater detail in the current work, not simply stated in an off—handed matter with a reference to the prior work.
- L32: 'had a different reference instrument' what was the instrument and why do the authors think that this particular reference instrument was in error, subsequently disrupting the validity of their calibration model?
- L34 – carried into highlighted passage below: The authors indicate that the sampling sites or the circumstances discussed previously are the reason for the poor model performance, not the sensors comprising the UPODs. First, WHAT circumstances specifically, and what specifically about the sampling sites? This level of non-explanation is unacceptable.

seems to be connected to the sampling sites or the circumstances discussed previously as opposed to the quality of individual sensors in each of those U-Pods.

All SJ Basin U-Pod $O_3$ measurements systematically over estimate lower levels of $O_3$ each night, a trend apparent in the
5    scatter plots in Fig. 5 and in the residuals by time of day plot in Fig. 6. Upon examination of the scatter plots in Fig. 5, U-Pods at some sampling sites had positive bias for higher $O_3$ measurements as well (Shiprock, Ignacio, Sub Station, and Bloomfield), while for others, bias at the higher end of $O_3$ distributions did not appear to be present (Navajo Dam, Fort Lewis, and Bondad). The residuals by time of day plot in Fig. 6 shows that the two U-Pods at BAO did not have significant trends in their residuals according to the time of day, but that U-Pods deployed at SJ Basin sites consistently over estimated
10    nighttime $O_3$. The residuals are also plotted with respect to temperature in Fig. 6, where all U-Pods, even those at BAO to a lesser extent, appear to over predict $O_3$ at lower temperatures, which generally occurred at night. The times of day that generally correspond to the highest $O_3$ levels generally had the lowest residuals, with some exceptions at the Fort Lewis and Navajo Dam sites.

15    Fig. 6 includes a plot of the residuals across the duration of the deployment period, showing no significant sensor drift in measurements for any of the U-Pods. This plot also shows that the highest residuals observed generally occurred over short periods in time, particularly for the Fort Lewis (blue) and Sub Station (magenta) sites. In order to further explore the performance of field calibration models for $O_3$ at SJ Basin sites relative to BAO, the combined parameter space of temperature with $O_3$ reference mole fractions and temperature with absolute humidity are presented in Fig. 7. The combined
20    temperature and reference $O_3$ parameter space appears to be similar for all of the U-Pods, both at BAO and the SJ Basin sites. However, there appears to be some outlying combined temperature and humidity parameter space at the Sub Station site and at the Navajo Dam site. Brief excursions of high humidity may be connected to some of the large short-term residuals observed at these two sampling sites.

- L22: brief excursions of high humidity – how brief? How high?
- Can the authors comment on the role that humidity transients play in fundamental sensor response? The description of the high and low bias resulting from the models at different locations and different times of day is difficult to follow. What are the common response characteristics and failings of the model that manifest across the case studies featured here? What are the lessons learned and how can these lessons better inform ANN model development moving forward?

Section 3.2.1

- Extrapolation of the ANN and LM models is problematic. Why? If the full-span of O3 (or CO2) concentration encountered in the field deployment is not covered in the training set for the model, is the model incapable of reasonably extrapolating?

Section 3.2.2

- L15: post-test deployment co-locations: It's unclear what is meant by 'post-test', please clarify.
- L16: state the # of UPODs
- The concept of extrapolation in time is confusing. Please clarify what is meant by this? Generating a model at time X and then applying that same model to time X-Y?
- The authors identify coal-fired power plants as an important near-field ('close-by') pollutant source that could contribute a specific (unique) pollutant signature that could render the utility of the Figaro sensor useless. Did the CO2 response of the UPODs or reference instruments or CO

response of the sensor measurements indicate a near-field power plant plume across the deployment area?

Section 3.2.3

- How specifically was 'time' included as a raw input vector in the training matrix? Absolute time? Time since start of deployment? Time since calibration? Time since sensor manufacture?
- L11-12: "…LMs outperformed ANNs with notable instability associated with the performance of ANNs when time was included as an input." In the previous sentence the authors stated that time was useful predictor of CO2.. but the last sentence appears to contradict this assertion. The fact that LMs outperformed ANNs for CO2 also contradicts general assertions made in the abstract.
- The authors should comment on the notion that time-sensitive response patterns in sensors indicates that some level of time-decay. Is this the case with the CO2 sensor and that's why time as a input parameter in the model makes such a big difference? Is there some fundamental reason why the ANNs would be poorly suited to model time-decay patterns in the sensors?

Section 3.2.4

- L23-24 – final sentence in this section is very important. Where the faulty RH (and necessity of substituting RH from alternate sources) degraded the models, if enough RH variability was captured with the suite of peripheral metal oxides sensors, the RH-interference could be effectively modeled without explicit RH inputs. It would seem important to emphasize this point a bit more prominently and discuss further – especially in the context of overcoming some of the RH-measurement shortfalls elsewhere in the manuscript through similar means.

4. Conclusions

- Supervised learning techniques – generally, the manuscript lacks a description of what is meant by this -
- L19-20 the concepts of temporal and spatial extension are still a bit confusing here. Earlier statements to clarify exactly what is meant by each condition would be helpful.
- L24: how does one move something in terms of its temporal coverage?
- L1-3P16: LMs appear to be more robust when applied to a changing deployment condition – but then the authors hedge and say that they "… were not able to fully represent some of the complex nonlinear response behavior exhibited by the arrays of sensors." So a linear model can't model nonlinear behavior? The statement needs to be more specific.
- L7: "..data is almost a band running vertically in a range of CRMSEs." Data running in 'a band' doesn't aid in the interpretation of the data. Re-phrase to address the statistical product that results from the bias that was encountered.
- Final paragraph: how 'generalizable' are the models developed here? It would seem that despite having done an exhaustive amount of work, each individual UPOD system still required its own ANN or LM based on co-located data and raw sensor data from that individual sensor system. While the input matrix of raw sensor signals may be more generalizable, the models themselves appear to be very much node-specific, at least in so far as what has been shown in the paper.
- It is unclear how the extension of the model frameworks discussed in the current paper can be used in the context of low-cost electrochemical sensors

---

## Author Comment (AC1) · 4 Sep 2018

Dear Reviewer 1,

We would like to offer our sincere thanks for spending your time in the review of our work and helping us to significantly improve the quality and clarity of the manuscript with your very detailed comments and suggestions. Each of your comments is listed below in black text, followed by our response and edits in blue text.

**Review 1 Comments:**

**Overview:** In Casey et al., the authors investigated the performance of calibration models developed for ambient O3 and CO2 across time and space using field deployments spanning 2014 – 2017 as case studies. Specifically, they looked at the impact of post-deployment calibration vs pre- and post- calibration, and the impact of applying a calibration model developed in one location on U-Pods deployed in other locations. Calibration models investigated included linear models and artificial neural networks. The size and scope of the study is impressive, and I believe there is a significant quantity of insightful information within this paper.

However, in general, I found the narrative of the paper to be confusing (it is hard to effectively distill such a breadth of research) and the take home points could be made considerably clearer.

Additionally, I think this paper would benefit with a few more analyses of general model performance implications and a closer look at the impact of relative humidity. Following these corrections to comments identified below, I believe the publication is suitable to be published in Atmospheric Measurement Techniques.

**Response:** We have carefully addressed each of the comments below and carried out the analyses suggested by the reviewer to investigate the implications and impact of relative humidity and sensor drift in time. We have also worked to significantly improve and clarify the narrative of the paper as well as clarified and outlined take home points.

GENERAL COMMENTS

**Comment:** In general, I found this manuscript a little hard to read, because I felt like a cohesive narrative was missing. A lot of information is presented in a rapid-fire manner such that the results and discussion section reads more like a results section, with limited discussion. Although there is no straight forward solution to this problem, I would suggest that the authors think about the three to five key messages they wish to convey in the manuscript and that they tune and streamline the text to support this narrative.
**Response:** Thanks very much to the Reviewer for helping point out a way for us to improve a cohesive and clear narrative in this work.
**Edits:** We have added the following five key points to the conclusion as well as supporting edits in the abstract and throughout the results and discussion section:

[revised manuscript text omitted]

**Comment:** Many of the figures are needlessly complicated by an overload of case studies, unintelligible sensor signal labels, and colours. If there is any way of summarizing this data more cohesively, it would significantly improve the paper.

**Response:** Thank you very much for the feedback and helping us to simplify and clarify figures.

**Edits:** According to the specific comments below, we have split what was previously Figure 8 into two figures (now Figure 10 and 11) in order to simplify the graphics and highlight the content of each and simplified and clarified Figure 9 (now Figure 12). We have added definitions for the sensor inputs in the Figure captions for what were Figures 8 and 9 (now Figures 10, 11, and 12). We have also updated Figure 1 (now Figure 2) in order to clarify model training and test periods for each case study, as well as how many U-Pods were included in each case study.

[Figure]

**Figure 10**: Target diagrams demonstrating performance of a previously determined best-performing model across all new test datasets. (a) CO2 and (b) O3 LM performance when only the primary gas sensor, temperature and humidity are inputs. (c) CO2 and (d) O3 ANN performance with inputs that were found to perform best at the GRET site in the spring of 2017 (Casey et al., 2017). Model input definitions: eltCO2 (ELT S300 CO2 sensor), e2vO3 (e2v MiCs-2611 sensor), temp (temperature) , and absHum (absolute humidity).

[Figure]

**Figure 11**:  **Target diagrams demonstrating performance of a previously determined best-performing model across all new test datasets (a) CO2 and (b) O3 ANN performance with inputs that were found to perform best at the GRET site in the spring of 2017** (Casey et al., 2017).  **Model input definitions:  eltCO2 (ELT S300 CO2 sensor), e2vCO (e2v MiCs-5525 sensor), e2vVOC (e2v MiCs-5521 sensor), e2vO3 (e2v MiCs-2611 sensor), figCH4 (Figaro TGS 2600 sensor), figCxHy (Figaro TGS 2602 sensor), temp (temperature) , and absHum (absolute humidity).**

[Figure]

**Figure 12: Target diagrams for (a) CO₂ and (b) O₃ calibration model performance for the best performing model for each particular case when tested on data from a number of field deployments. Model input definitions: eltCO2 (ELT S300 CO2 sensor), e2vCO (e2v MiCs-5525 sensor), e2vVOC (e2v MiCs-5521 sensor), e2vO3 (e2v MiCs-2611 sensor), figCH4 (Figaro TGS 2600 sensor), figCxHy (Figaro TGS 2602 sensor), alphaCO (Alphasense CO-B4 sensor) temp (temperature), absHum (absolute humidity), rh (relative humidity), and time (absolute time).**

**Comment:** It would be good if the authors could elaborate on which U-Pods were where over all these campaigns. Given that temporal degradation / time was investigated in detail in this paper, some assessment of UPod changes over the three years of campaigns would be helpful if possible.

**Response:** Thanks very much to the reviewer for this helpful comment.

**Edits:** Figure 2 (previously Figure 1) has been updated to clearly state the number of U-Pods included in each case study, for both O₃ and CO₂, as well as names of the specific U-Pods that were used during each case study. We have also performed an assessment of U-Pod sensor drift from the summer of 2015 through the summer of 2017, shown below in Figure s26, that we have added to the Supplemental Materials. Figure s26 had been sited in section 3.2.2 of the manuscript, and described in the following text: "While we did not measure and record metal oxide sensor heater resistance for sensors included in U-Pods, we have investigated eltCO2 and e2vO3 sensor signal drift from the summer of 2015 through the summer of 2017. These data are presented in Fig. S26. Systematic

downward drift in all eltCO2 sensor signals is apparent over this time frame.  A clear and consistent pattern of systematic drift over this time period is less apparent for e2vO3 sensors.  Since the training data was collected immediately after, the test data period, and since the test data period was relatively short (approximately one month) sensor drift could be negligible across the combined training/testing time frame."

**(a)**

| Case Study | Year | \multicolumn{12}{c|}{Model Training and Test Deployment Timelines} |
|---|---|---|
| | | Jan · Feb · Mar · Apr · May · Jun · Jul · Aug · Sep · Oct · Nov · Dec |
| Dawson Summer 2014 | 2014 | |
| SJ Basin Spring 2015 | 2015 | |
| SJ Basin Summer 2015 | 2015 | |
| BAO Summer 2015 | 2015 | |
| BAO Summer 2016 | 2016 / 2017 | |
| GRET Fall 2016 | 2016 / 2017 | |
| GRET Spring 2017 | 2017 | |

**(b)**

| Case Study | Training Location | Test Location | $O_3$ # U-Pods | $O_3$ U-Pod Names | $CO_2$ # U-Pods | $CO_2$ U-Pod Names |
|---|---|---|---|---|---|---|
| Dawson Summer 2014 | CAMP | Dawson | 1 | BE | NA | NA |
| SJ Basin Spring 2015 | BAO | SJ Basin | 4 | BB, BD, BF, BJ | NA | NA |
| SJ Basin Summer 2015 | BAO | SJ Basin | 7 | BA, BB, BD, BE, BF, BH, BI | 2 | BB, BD |
| BAO Summer 2015 | BAO | BAO | 2 | BC, BJ | 2 | BC, BJ |
| BAO Summer 2016 | GRET | BAO | 2 | BH, BI | 2 | BH, BI |
| GRET Fall 2016 | GRET | GRET | 2 | BH, BI | 2 | BH, BI |
| GRET Spring 2017 | GRET | GRET | 2 | BH, BI | 2 | BF, BI |

**Figure 2**: **(a) ANN and LM training and test deployment timelines.  The Dawson, BAO, and GRET sampling sites are all located in the DJ Basin.  Model training periods for each test deployment are shown in blue, and model test periods are shown in magenta.  For the BAO Summer 2016 case study, the period outlined in blue shows data that was used to train $O_3$ model, but not $CO_2$ models since $CO_2$ reference data was not available during winter months. (b) Information about each of the case studies presented in the above timelines, including model training and testing locations, as well as the number and names of U-Pods included in each case study for both $O_3$ and $CO_2$ models.  The U-Pods with names shown in grey were constructed and deployed starting in May of 2014.  The U-Pods with names shown in black were constructed and deployed starting in April of 2015.**

[Figure]

**Figure S26 U-Pod sensor drift from 2015 – 2017 for (a) e2v MiCs-2611 $O_3$ sensors and (b) ELT S300 $CO_2$ sensors. Data presented are from 23-day periods when U-Pods were co-located together from the summers of 2015, 2016, and 2017. Raw ADC sensor signals were smoothed with rolling hourly medians during these periods, in order to track representative sensor responses across this time period, without the influence of exceptional events. Measurements from summer each year were used to capture sensor response under similar weather conditions.**

**Comment:** Also, when comparing sensor performance spatially, are the U-Pods that are compared the same age?

**Response:** Thanks to the Reviewer for bringing up this important point.

**Edits:** We have added the following text accordingly: "Some U-Pods used included in these case studies (indicated in grey font in Fig. 2) were constructed, populated with sensors, and deployed at field sites in the spring of 2014, approximately a year before the rest of the U-Pods were constructed, populated with sensors, and deployed at field sites in the spring of 2015. The relative age of sensor systems included in some case study comparisons could have contributed to some discrepancy in model performance, though systematic differences based on U-Pod age is not apparent."

**Comment:** Directly addressing the size of the training and testing windows should be included. It is hard to make generalizable conclusions from the study when there is so much variability in training and testing window size. Is there a reason why some training windows are shorter than others? This should be directly addressed in the manuscript.

**Response:** Thanks to the Reviewer for the helpful feedback.

**Edits:** We have added the following explanatory text accordingly: "As available data from each case study allowed, we used approximately one month of training data before and after (pre and post of) a given approximately month-long test period. When training data was not available within several months of a test period, significantly longer training datasets were used in order to attempt capture and effectively represent trends in sensor drift over time, as well as to avoid extrapolation of model parameters (particularly temperature) during the test data period. As a result, model-training durations varied across case studies and sometimes significantly exceeded model-testing durations. Each case study is similar in representing approximately one month-long deployment of sensor systems. This study design serves a primary goal of this work, which is to help support the quantification atmospheric trace gases from low-cost gas sensor data in new locations, relative to model training locations, for periods of approximately one month at a time."

**Comment:** I found the discussion of ANN and LM model building to be significantly under-developed, especially considering that this is a measurement techniques journal. This paper relies too heavily on the prior 2017 study, and has too much assumed knowledge that should be summarized in Section 2.4. The resulting LM and coefficients should be provided. As well as some mention of model performance metrics like MAE or r2.

**Response:** Thanks very much to the Reviewer for this helpful feedback. We have developed the discussion of ANN and LM model building significantly, through the addition of the following text in section 2.4 and have added a new subsection 2.5 describing the calibration model evaluation and testing we implemented in this work. We have also added a table summarizing model performance metrics for our previous work, as a case study, among other case studies. Since LM coefficients are unique to individual case studies, and within those groups, unique to gas sensors in individual U-Pods, and since we carried out an analysis of model sensitivity to inputs in our previous work, we have not included LM coefficients in this work.

[revised manuscript text omitted]
 work. The third group of models tested for each case study includes models that were optimized specifically for each case study. Tables 5 and 6 show the mean and standard deviation of model performance metrics for each of the case studies presented."

**Table 5**: $O_3$ model performance metrics.

| Case Study | N | $R^2$ | RMSE (ppb) | MBE (ppb) | Standard Deviation $R^2$ | Standard Deviation RMSE | Standard Deviation MBE |
|---|---|---|---|---|---|---|---|
| O₃ Models | | | | | | | |
| **Best O₃ Model (Casey et al., 2017)**
 **ANN with inputs:  e2vO3 temp absHum e2vVOC e2vCO FigCH4 FigCxHy** | | | | | | | |
| Dawson Summer 2014 | 1 | 0.83 | 6.46 | -0.91 | 0.00 | 0.00 | 0.00 |
| SJ Basin Spring 2015 | 4 | 0.86 | 7.74 | 3.69 | 0.05 | 3.82 | 5.78 |
| SJ Basin Summer 2015 | 7 | 0.85 | 7.03 | 4.89 | 0.10 | 1.10 | 1.73 |
| BAO Summer 2015 | 2 | 0.93 | 4.26 | 1.45 | 0.00 | 0.31 | 0.07 |
| BAO Summer 2016 | 2 | 0.92 | 12.21 | -11.14 | 0.00 | 0.31 | 0.07 |
| GRET Fall 2016 | 2 | 0.96 | 12.87 | 12.02 | 0.01 | 2.30 | 2.35 |
| GRET Spring 2017 | 2 | 0.98 | 2.59 | 1.49 | 0.00 | 0.69 | 1.02 |
| **Simple Model (Single Gas Sensor)**
 **LM with inputs:  e2vO3 temp absHum** | | | | | | | |
| Dawson Summer 2014 | 1 | 0.95 | 3.59 | -0.46 | 0.00 | 0.00 | 0.00 |
| SJ Basin Spring 2015 | 4 | 0.83 | 17.95 | 16.09 | 0.06 | 6.10 | 5.83 |
| SJ Basin Summer 2015 | 7 | 0.86 | 6.30 | 3.53 | 0.06 | 1.40 | 2.06 |
| BAO Summer 2015 | 2 | 0.87 | 5.50 | 0.94 | 0.00 | 0.78 | 1.56 |
| BAO Summer 2016 | 2 | 0.89 | 5.78 | -2.71 | 0.00 | 0.78 | 1.56 |
| GRET Fall 2016 | 2 | 0.93 | 12.73 | 11.92 | 0.01 | 0.62 | 0.88 |
| GRET Spring 2017 | 2 | 0.89 | 6.00 | -3.19 | 0.00 | 0.73 | 1.38 |
| **Models Optimized For Case Studies** | | | | | | | |
| Dawson Summer 2014 | 1 | 0.95 | 3.59 | -0.46 | 0.00 | 0.00 | 0.00 |

| | | | | | | | |
|---|---|---|---|---|---|---|---|
| SJ Basin Spring 2015 | 4 | 0.86 | 7.74 | 3.69 | 0.05 | 3.82 | 5.78 |
| SJ Basin Summer 2015 | 7 | 0.85 | 7.03 | 4.89 | 0.10 | 1.10 | 1.73 |
| BAO Summer 2015 | 2 | 0.93 | 4.26 | 1.45 | 0.02 | 0.51 | 1.54 |
| BAO Summer 2016 | 2 | 0.87 | 6.25 | -0.20 | 0.02 | 0.51 | 1.54 |
| GRET Fall 2016 | 2 | 0.95 | 3.99 | 2.14 | 0.00 | 0.28 | 0.89 |
| GRET Spring 2017 | 2 | 0.98 | 2.59 | 1.49 | 0.00 | 0.69 | 1.02 |

Table 6: $CO_2$ model performance metrics.

| Case Study | N | $R^2$ | RMSE (ppm) | MBE (ppm) | Standard Deviation $R^2$ | Standard Deviation RMSE | Standard Deviation MBE |
|---|---|---|---|---|---|---|---|
| CO$_2$ Models | | | | | | | |
| **Best CO$_2$ Model from (Casey et al., 2017)** **ANN with inputs:  eltCO2 temp absHum** | | | | | | | |
| SJ Basin Summer 2015 | 2 | 0.65 | 8.42 | -0.62 | 0.00 | 1.81 | 1.41 |
| BAO Summer 2015 | 2 | 0.75 | 9.98 | -2.60 | 0.05 | 13.00 | 13.89 |
| BAO Summer 2016 | 2 | 0.69 | 54.38 | 48.37 | 0.05 | 13.00 | 13.89 |
| GRET Fall 2016 | 2 | 0.74 | 42.37 | 39.58 | 0.02 | 2.44 | 2.57 |
| GRET Spring 2017 | 2 | 0.83 | 6.31 | 0.59 | 0.03 | 0.13 | 2.61 |
| **Simple Model (Single Gas Sensor)** **LM with inputs:  eltCO2 temp absHum** | | | | | | | |
| SJ Basin Summer 2015 | 2 | 0.71 | 7.84 | 0.27 | 0.01 | 1.43 | 0.42 |
| BAO Summer 2015 | 2 | 0.69 | 10.62 | -1.26 | 0.06 | 1.52 | 10.67 |
| BAO Summer 2016 | 2 | 0.73 | 11.82 | 0.73 | 0.06 | 1.52 | 10.67 |
| GRET Fall 2016 | 2 | 0.82 | 8.62 | -3.46 | 0.00 | 0.69 | 1.45 |
| GRET Spring 2017 | 2 | 0.55 | 9.88 | -0.33 | 0.03 | 0.29 | 1.91 |
| **Models Optimized For Case Studies** | | | | | | | |
| SJ Basin Summer 2015 | 2 | 0.72 | 7.45 | -0.11 | 0.04 | 2.06 | 0.31 |
| BAO Summer 2015 | 2 | 0.80 | 8.85 | -2.29 | 0.10 | 6.47 | 7.08 |
| BAO Summer 2016 | 2 | 0.73 | 11.82 | 0.73 | 0.06 | 1.52 | 10.67 |
| GRET Fall 2016 | 2 | 0.82 | 8.62 | -3.46 | 0.00 | 0.69 | 1.45 |
| GRET Spring 2017 | 2 | 0.83 | 6.31 | 0.59 | 0.03 | 0.13 | 2.61 |

**Comment:** It would be good to include an explicit discussion of % reduction in error by using established models vs. "best fit" models. Can we generalize? What is the quantitative impact of using your prior models vs making a new model every time. My interpretation from this paper is that we need a new model for every U-Pod for every deployment – is there any way around this? I feel there is a significantly missed opportunity to be quantitative here. Section 3.3 could be substantially enhanced using some sort of summary figure/table (other than a target diagram) that gives percent change in bias, random error, r2, mae etc. by switching from pre/post to just post, or by switching location. Given that there are many pairs of sensors looking at impact of pre/post vs. just post or impact of location switching, you could show average % change in model fitting statistics as well as confidence intervals or standard deviations to show the spread across the case studies. This might be a helpful way of streamlining the paper.

**Response:** Thanks very much to the Reviewer for this helpful comment, which will help us be more quantitative as well as clarify and focus the narrative and results we present.

**Edits:** Accordingly, we have added a table showing the percent change in R2, RMSE, and MBE, when one set of models is used instead of another, as well as the following text: "Table 7 shows the percent change in model performance metrics when one model-training paradigm is used in place of another, highlighting relative benefits associated with the implementation of different models for $O_3$ and $CO_2$."

**Table 7**: **Relative benefits associated with the implementation of different models for $O_3$ and $CO_2$.**

| Case Study | Mean % Increase in $R^2$ | Mean % Decrease in RMSE | Mean % Decrease in MBE | Mean % Increase in $R^2$ | Mean % Decrease in RMSE | Mean % Decrease in MBE |
|---|---|---|---|---|---|---|
| | $CO_2$ Models | | | $O_3$ Models | | |
| **Benefit of Models Optimized For Case Studies Over The Best Models from (Casey et al., 2017)** | | | | | | |
| Dawson Summer 2014 | | | | 14.51 | 44.42 | 50.00 |
| SJ Basin Spring 2015 | | | | 0.00 | 0.00 | 0.00 |
| SJ Basin Summer 2015 | 10.56 | 11.52 | 82.60 | 0.00 | 0.00 | 0.00 |
| BAO Summer 2015 | 5.84 | 11.27 | 11.95 | 0.00 | 0.00 | 0.00 |
| BAO Summer 2016 | 5.72 | 78.27 | 98.49 | -5.01 | 48.82 | 98.19 |
| GRET Fall 2016 | 11.17 | 79.66 | 108.73 | -0.54 | 68.99 | 82.22 |
| GRET Spring 2017 | 0.00 | 0.00 | 0.00 | 0.00 | 0.00 | 0.00 |
| **Benefit of The Best Models from (Casey et al., 2017) Over Simple Linear Models** | | | | | | |
| Dawson Summer 2014 | | | | -12.67 | -79.92 | -99.99 |
| SJ Basin Spring 2015 | | | | 3.20 | 56.88 | 77.09 |
| SJ Basin Summer 2015 | -8.41 | -7.29 | 331.39 | -1.34 | -11.53 | -38.41 |
| BAO Summer 2015 | 8.70 | 6.05 | -106.48 | 6.79 | 22.48 | -53.85 |
| BAO Summer 2016 | -5.41 | -360.09 | -6543.84 | 2.57 | -111.22 | -310.71 |
| GRET Fall 2016 | -10.05 | -391.73 | 1244.99 | 2.88 | -1.12 | -0.86 |
| GRET Spring 2017 | 51.92 | 36.13 | 278.55 | 10.00 | 56.90 | 146.65 |
| **Benefit of Models Optimized For Case Studies Over Simple Linear Models** | | | | | | |
| Dawson Summer 2014 | | | | 0.00 | 0.00 | 0.00 |
| SJ Basin Spring 2015 | | | | 3.20 | 56.88 | 77.09 |
| SJ Basin Summer 2015 | 1.26 | 5.06 | 140.25 | -1.34 | -11.53 | -38.41 |
| BAO Summer 2015 | 15.04 | 16.64 | -81.80 | 6.79 | 22.48 | -53.85 |
| BAO Summer 2016 | 0.00 | 0.00 | 0.00 | -2.57 | -8.10 | 92.59 |

| | | | | | | |
|---|---|---|---|---|---|---|
| GRET Fall 2016 | 0.00 | 0.00 | 0.00 | 2.33 | 68.64 | 82.07 |
| GRET Spring 2017 | 51.92 | 36.13 | 278.55 | 10.00 | 56.90 | 146.65 |

**Comment:** This is mentioned in the specific comments, but I would like to see a quantitative assessment of the impact of swapping out RH data if a U-Pod failed. You could accomplish this by taking a U-Pod with valid RH data, replacing it with the Picarro or nearby station RH data, and quantitatively assessing the impact on model performance. That way, you could transition from hypotheticals about the impact of this data swapping to some actual numbers.

**Response:** Thanks very much to the reviewer for helping us to be less hypothetical about the impact of this data swapping. We have carried out a dummy experiment, testing the effect of this humidity data swapping on data collected during the GRET Spring 2017 case study. A figure, showing the relative performance of models when the humidity data was taken from the U-Pods directly and replaced with measurements from the Picarro CRDS, has been added to the Supplemental Materials. This figure, and associated implication have been cited in the main text.

**Edits:** "In our previous work, we showed that $O_3$ models were very sensitive to the humidity signal input (Casey et al., 2017). In this case study, it seems that replacing actual humidity signals with closely approximated humidity signals, negatively influenced model performance. In order to investigate this observation further, we tested the influence of replacing humidity data in the same manner, using mixing ratios from the same co-located Picarro, on test data from the GRET Spring 2017 case study. A comparison of model performance under normal and this 'borrowed RH' circumstance are presented in Fig. S27 in the SM. $O_3$ model performance was negatively impacted when 'borrowed' RH values based on Picarro data replaced U-Pod RH sensor signals. From these findings, it seems likely that the inclusion of multiple metal oxide type sensors as inputs in the model, which all respond strongly to humidity fluctuations, helped the ANN to effectively represent the influence of humidity in the system, more so than including a 'borrowed RH' signal from another instrument. We tested models with multiple gas sensor signals and no humidity signal as inputs for a number of other case studies as well (as seen in Fig. S2, Fig. S4, and Fig. S5), when good humidity data from U-Pod enclosures was available, but they did not turn out to be the best performing model in any of these other tests."

[Figure]

[Figure]

**Figure S27 A comparison of model performance when humidity inputs are taken from sensor measurements collected within a given U-Pod sensor system enclosure, vs the performance of models when humidity inputs are replaced using data from a Picarro CRDS for (a) O$_3$ (b) CO$_2$**

SPECIFIC COMMENTS
**Comment:** P1 - L13-14: Seems like an oxymoron to say "Generally" if the circumstances for best model performance are case study specific. Recommendation to remove the word "generally".
**Response:** Thank you for the helpful feedback.
**Edits:** We have removed the word 'Generally'.

**Comment:** P3 - L19-24: Please discuss why ozone is elevated near O&G production.

**Response:** Thank you for helping us clarify why ozone can be elevated near oil and gas production activities.

**Edits:** We have added the following text to augment this discussion: "NO$_X$ and VOC emissions, including those from oil and gas production activities, react in the atmosphere in the presence of sunlight to form tropospheric O$_3$."

"Emissions of industry related air pollutants, including O$_3$ precursors, NO$_X$ and VOCs are expected to occur on spatially distributed scales, across multiple individual components on individual well pads, transmission lines, transportation routes, and gathering stations that are each distributed throughout production basins (Litovitz et al. 2013; Mitchell et al. 2015; Allen et al. 2013). Spatially distributed networks of low-cost sensors have the potential to better inform spatial variability of air quality than existing Regulatory air quality monitoring stations which feasibly cover such spatially resolved measurements continuously, and may not be representative of air quality across smaller spatial scales (Bart et al., 2014; Jiao et al., 2016; Moltchanov et al., 2015)."

**Comment:** P3 – L27: Can you quantify "small spatial scales" in this context? Is well pad combustion and diesel traffic really contributing so much that it is universally increasing ozone? Most of the construction traffic would occur during active drilling and less so during production when well pad sites are very quiet. I think some further thinking or elaboration on this train of thought it warranted.

**Response:** Thank you for the helpful comment.

**Edits:** We have added the following detail, regarding spatial scales that ozone may be influenced near oil and gas emissions sources: " a modeling study concluded that oil and gas production activities could significantly impact ozone near emissions sources, beginning 2 and 8 km downwind of compressor engine and flaring activities, respectively [3]."

We have also added the following text to address how emissions may change across the lifetime of a given oil and gas production well: "While emissions from truck traffic (and in some cases drilling rig generators), at a given well pad are highest during the drilling, stimulation, and completion phases, industry truck traffic often persists as produced water and condensate tanks are collected from storage tanks on a well pad throughout the life a the well, as do emissions from flaring and compressor engines."

**Comment:** P4 – L1-2: What do you mean by "pooling" of compounds - I am not sure I understand this sentence.

**Response:** Thank you for helping us clarify this statement.

**Edits:** We have made the following edits accordingly: "While elevated ambient CO$_2$ levels are not directly harmful to human health, continuous CO$_2$ measurement can provide information about nearby combustion-related pollution and atmospheric dynamics that lead to the accumulation of potentially harmful compounds associated with the oil and gas production industry during periods of atmospheric stability."

**Comment:** P4 – L5-8: I think some short discussion of the operating principles of the sensors would be helpful here.

**Response:** Thank you for the helpful feedback.

**Edits:** We have added a discussion of the operating principles of the sensors to section 1.1 accordingly:

"While low-cost sensors have been emerging on the market with sufficient sensitivity to resolve variations in ambient mole fractions of target gases of interest, they are also sensitive to temperature and humidity variations that occur in the ambient environment. NDIR sensors, like the ELT s300 CO$_2$ sensor employed in this study, have good selectivity, but, since pressure and temperature are not controlled in the optical cavity of ELT s300 CO$_2$ sensors, the influence of temperature on sensor signals plays an important role. The influence of humidity is also important to address because changes in water vapor are known to influence NDIR measurements of CO$_2$ in terms of spectral cross-sensitivity due to absorption band broadening (Licor, 2010).

Both metal oxide and electrochemical type sensors operate on the principle of oxidizing or reducing reactions at sensor surfaces. For electrochemical sensors, like the Alphasense CO-B4 sensor

employed in this study, oxidizing or reducing compounds react at the working electrode, resulting in the transfer of ions across an electrolyte solution from the working electrode to the counter electrode, balanced by the flow of electrons across the circuit connecting the working electrode to the counter electrode.  A linear relationship is expected between this current and the target gas mole fraction.   Electrochemical sensors can be tuned to respond more or less strongly to specific gases by adjusting the materials properties of the working electrode. A membrane is located between the working electrode and the exterior of the sensor in order to control redox reaction rates.  Gases diffusion through the membrane to reach the working electrode and the electron transfer rates have been shown to increase at higher temperatures (Xiong and Compton, 2014), and since chemical reaction rates are also influenced by temperature, electrochemical sensor responses can be influenced by sensor operating temperature.  Changes in ambient humidity levels can cause sensors to loose or gain of the electrolyte solution, by mass, also influencing electrochemical sensor response (Xiong and Compton, 2014).

For metal oxide sensors, and to a lesser extent for electrochemical sensors, resolving the response of a sensor attributable to the target gas species can also pose a challenge in the presence of interfering gas species.  Metal oxide sensors, like those used in this study, have a resistive heater circuit that warms up the sensor surface, causing $O_2$ molecules to adsorb to the sensor surface, which leads to increased resistance across the surface of the sensor.  In the presence of an oxidizing compound, like $O_3$, more oxygen molecules are adsorbed to the sensor surface and the resistance across the sensor surface in increased further.  In the presence of a reducing compound, like CO, oxygen molecules are removed from the sensor surface, allowing electrons to flow more freely, resulting in decreased resistance across the sensor surface. For metal oxide sensors, the resistance across the sensor surface can then be used to determine the mole fraction of a given oxidizing or reducing compound, often according to a nonlinear relationship.  Exposure to humidity has been shown to significantly lower the sensitivity of metal oxide gas sensors making it an important parameter to address in a gas quantification model (Wang et al., 2010).  Metal oxide sensor operating temperature has also been shown to strongly influence sensor sensitivity and selectivity to different gas species (Wang et al., 2010).  Metal oxide type sensors can be tuned to respond differently from one another to oxidizing and reducing gas species by using different metal oxide materials and doping agents for the sensor surface, but selectivity is difficult to achieve."

We have also added a section to the introduction, section 1.2, entitled "Low-Cost Air Quality Sensor Quantification:

"Because low-cost gas sensor signals are influenced, sometimes significantly, by interfering gas species and changing weather conditions in the ambient environment, field normalization methods to quantify atmospheric trace gases using low-cost sensors have been found to be more effective than lab calibration (Cross et al., 2017; Piedrahita et al., 2014; Sun et al., 2016).  Our previous study and several others have compared the efficacy field calibration models generated using LMs (simple and multiple linear regression) relative to supervised learning methods (including ANNs and random forests), all finding that ANNs (Casey et al., 2017; Spinelle et al., 2015, 2017) and random forests (Zimmerman et al., 2017) outperformed LMs in the ambient field calibration of low-cost sensors.  Like earlier laboratory based studies (Brudzewski, 1999; Gulbag and Temurtas, 2006; Huyberechts and Szeco, 1997; Martín et al., 2001; Niebling, 1994; Niebling and Schlachter, 1995; Penza and Cassano, 2003; Reza Nadafi et al., 2010; Srivastava, 2003; Sundgren et al., 1991), ANN-based calibration models, incorporating signals from an array of gas sensors with overlapping sensitivity as inputs, have been able to effectively compensate for the influence of interfering gas species and resolve the target gas mole fraction.

ANNs are known to be able to very effectively represent complex, nonlinear, and collinear relationships among input and output variables in a system (Larasati et al., 2011).  ANNs are useful in the field calibration of low-cost sensors because, through pattern recognition of a training dataset, they are able to effectively represent the complex processes and relationships among sensors and the ambient environment that would be very challenging to represent analytically or based on empirical

representation of individual driving relationships. In practice though, the reason multiple gas sensors are able to improve the performance of calibration models may be in part the result of correlation between mole fractions of target gases themselves that hold for one model training location, but might not remain effective at alternative sampling sites or during other time periods."

**Comment:** P4 – L25: Not sure what is meant by "toward" here
**Response:** Thank you for the helpful feedback.
**Edits:** We have replaced "toward" with "that were used for"

**Comment:** P5 – L27: Is this "clean air" normalization done dynamically/in real-time in parallel with the actual measurement? Or is the clean air measurement established during some calibration/maintenance? Please clarify.
**Response:** Thank you for helping us clarify.
**Edits:** We have added the following text accordingly: "For metal oxide type sensors, voltage signals were converted into resistance, and then normalized by the resistance of the sensor in clean air, $R_0$. A single value for $R_0$ was used for each sensor across the study duration. This $R_0$ value was taken as the resistance of each sensor at the GRET field deployment site when the target pollutant had approached background levels (at night for the metal oxide $O_3$ sensors and midday for all other metal oxide sensors), and when the ambient temperature was approximately 20° C and relative humidity of approximately 25%."

**Comment:** P5 – L30-32: Is there expected to be spatial variability of RH?
**Response:** Thank you for helping us to clarify.
**Edits:** We have added the following text to section 2.3: "The closest U-Pod with good humidity sensors ranged from several feet, when U-Pods were co-located during deployments in the DJ Basin, to approximately fifty miles during deployments in the San Juan Basin."

In Section 3.1 of we have added this text also: "Since the Ignacio site was located approximately twenty-two and fifty miles away from the Navajo Dam and Sub Station sites respectively, this could have introduced some additional error into the application of a calibration equation, particularly since we showed earlier that $O_3$ ANNs like the ones we employed here are very sensitive to humidity inputs (Casey et al., 2017). Spatial variability in humidity across tens of miles could be significant as isolated storms (which are on average 15 miles in diameter) propagate throughout the region in the summer."

**Comment:** P5 – L30-32: Why not just replace the RH sensors directly?
**Response:** Good question.
**Edits:** In answer, we have added the following text: "RH sensors were not replaced during field deployments in order to preserve consistency across different deployment periods, allowing for the possibility of a single comprehensive model to apply to all data from a single U-Pod. After some experimentation in generating a 'master model' that could be applied to data from a given U-Pod for all collected field measurements, across several years, we determined that individual models for each deployment would be more effective, and replacing RH sensors that had drifted down would have been appropriate in support of the methods presented here. We have since upgraded to Sensirion AG SHT25 sensors, which appear to be more robust and consistent over the course of long-term field deployments."

**Comment:** P7 – Section 3.0 first paragraph – Are there some general conclusions from the SM that you can discuss here? Some discussion of model performance is warranted vs just describing what figures are in the SM.
**Response:** Thanks very much for the feedback.
**Edits:** We have moved the paragraph in question to the methods section and have added the following sentence, letting the reader know that these plots are discussed in the results and

discussion section in context with each case study presented: "The best-performing model for each case study are highlighted below in the Results and Discussion section."

**Comment:** P8 – L17: What is eltCO2?? Can you better define all the model parameter inputs? This comes up in Figure 9 as well.
**Response:** Yes, thank you for the feedback.
**Edits:** Description added here and at the first mention of other model input codes in the manuscript in the text: "eltCO2 (ELT S300 CO2 sensor)"
We have also defined these model input codes in the caption for Figure 9 (now Figure 12) as well as Figure 8 (now Figure 10 and Figure 11): "Model input definitions: eltCO2 (ELT S300 CO2 sensor), e2vCO (e2v MiCs-5525 sensor), e2vVOC (e2v MiCs-5521 sensor), e2vO3 (e2v MiCs-2611 sensor), figCH4 (Figaro TGS 2600 sensor), figCxHy (Figaro TGS 2602 sensor), alphaCO (Alphasense CO-B4 sensor), temp (temperature), absHum (absolute humidity), rh (relative humidity), and time (absolute time)."

**Comment:** P8 – L30-33: Is this early morning under prediction really true? Bloomfield doesn't look like it is exhibiting any diurnal variation in residual error at all… I feel like given the small number of U-Pods, it is hard to make this conclusion definitively.
**Response:** We agree, thank you for pointing the trend in Bloomfield out.
**Edits:** We have edited the text to say "three of four U-Pods" instead of "all four U-Pods"

**Comment:** P9 – L4-13: I am confused now – why did you use the model with three inputs (eltCO2, abshum, and temp) if the best performing model had more variables? I feel like the model selection discussion is substantially underdeveloped. There could be many good reasons to not choose a more complex model, but any discussion of this seems to be completely omitted, or the reasoning is too difficult to follow.
**Response:** Thank you for the feedback and for helping us to clarify. You have helped us see that some important details were missing from the methods section regarding model selection and testing procedures.
**Edits:** We have added a subsection 2.5 to the end of the methods section entitled "Calibration Model Evaluation and Testing". In this section, we first define the r2, RMSE, MBE, and CRMSE metrics that are used to evaluate the performance of a given model when it is applied to a test dataset. Next we added a paragraph describing how we first tested models that were found to perform best for each gas species in our previous work, and then evaluated the performance of the best model for each specific case study. We then describe the methodology behind model selection and testing for each case study, in the following text and in the newly added Table 4:

"First, we generated and applied the best performing model, as determined in our previous work (presented in Table 3), to data from each new case study. Each new case study was selected to challenge models in different ways in order to evaluate the resiliency of the findings from our previous study when challenged by different circumstances.

**Table 3: Best performing models, as determined for each gas species, in the previous study (Casey et al., 2017)**

| Gas Species | Model Type | Sensor Signal Model Inputs | |
|---|---|---|---|
| CO$_2$ | ANN | eltCO2
temp
absHum | (ELT S300 CO2 Sensor)
(temperature)
(absolute humidity) |
| O$_3$ | ANN | e2vO3
e2vCO
e2vVOC
figCH4
figCxHy | (e2v MiCs-2611)
(e2v MiCs-5525)
(e2v MiCs-5521)
(Figaro TGS 2600)
(Figaro TGS 2602) |

| | | temp | (temperature) |
|---|---|---|---|
| | | absHum | (absolute humidity) |

Next we tested LMs for CO2 and O3 that contained only the primary target gas sensor for each species, as well as temperature and absolute humidity as inputs.  Finally, we generated, applied, and evaluated the performance of a number of LMs and ANNs with different sets of inputs for each case study in order to see which specific model performed the best for each individual case study.  The $r^2$, RMSE, and MBE for each of these alternative models when applied to test data are presented in the supplemental materials (SM) in Fig. S2 through Fig. S7, along with representative scatter plots and time series comparing the performance LMs and ANNs for a given set of inputs.  In Fig. S2 through Fig. S7, the best performing model inputs for each train/test data pair are shaded in purple.  The type of model that performed the best (ANN vs. LM) is indicated in the caption of each figure.  We discuss both the performance of the previously determined best fitting model (generated using data from the GRET Spring 2017 case study) when applied and generated to data from new case studies, and the performance of models that were tuned to perform the best for each individual case study.  From these comparisons, we draw insight into circumstances that challenge model performance in terms of relative local emissions characteristics, location, and timing between model training and testing pairs.  Table 4 lists the relative timing and parameter coverage between model training and testing periods for dataset pairs, highlighting instances of incomplete coverage during training that led to model extrapolation during testing."

**Comment:**  P9 – L28-30: It would be good to do a more comprehensive assessment of the impact of replacing RH sensor signal on model performance. Could you conduct a dummy experiment where you replace RH data you actually logged with that of a nearby or alternate monitor and then quantify the impact on model outcome? Given that it seems that a) RH/abshum is an important variable and b) that you had significant data loss issues, I feel a more quantitative assessment of the impact of these data substitutions is needed.

**Response:** Thanks very much to the reviewer for helping us to be less hypothetical about the impact of this data swapping.  We have carried out a dummy experiment, testing the effect of this humidity data swapping on data collected during the GRET Spring 2017 case study.  A figure, showing the relative performance of models when the humidity data was taken from the U-Pods directly and replaced with measurements from the Picarro CRDS, has been added to the Supplemental Materials.  This figure, and associated implication have been cited in the main text.

**Edits:**  "The fall 2016 GRET test period coincided with the time period U-Pod absolute humidity was replaced using mixing ratios from a co-located Picarro due to missing humidity sensor data.  Interestingly, when this 'borrowed' humidity signal was not included as an input, the model performance markedly increased and became competitive with other 'same location' test deployment case studies.  In our previous work, we showed that $O_3$ models were very sensitive to the humidity signal input  (Casey et al., 2017).  In this case study, it seems that replacing actual humidity signals with closely approximated humidity signals, negatively influenced model performance.  In order to investigate this observation further, we tested the influence of replacing humidity data in the same manner, using mixing ratios from the same co-located Picarro, on test data from the GRET Spring 2017 case study.  A comparison of model performance under normal and this 'borrowed RH' circumstance are presented in Fig. S27 in the SM.  $O_3$ model performance was negatively impacted when 'borrowed' RH values based on Picarro data replaced U-Pod RH sensor signals.  From these findings, it seems likely that the inclusion of multiple metal oxide type sensors as inputs in the model, which all respond strongly to humidity fluctuations, helped the ANN to effectively represent the influence of humidity in the system, more so than including a 'borrowed RH' signal from another instrument.  We tested models with multiple gas sensor signals and no humidity signal as inputs for a number of other case studies as well (as seen in Fig. S2, Fig. S4, and Fig. S5), when good humidity data from U-Pod enclosures was available, but they did not turn out to be the best performing model in any of these other tests."

[Figure]

**Figure S27 A comparison of model performance when humidity inputs are taken from sensor measurements collected within a given U-Pod sensor system enclosure, vs the performance of models when humidity inputs are replaced using data from a Picarro CRDS for (a) O₃ (b) CO₂**

**Comment:** P11 – L2-13: Can you comment on the quality of the fit at Dawson vs CAMP in addition to the ideal model. The discussion is fairly qualitative. Also, the LM should be much better at extrapolating vs the ANN (which cannot extrapolate I think…not sure) – can you comment on this difference? Does LM perform better because it can extrapolate?
**Response:** Thank you for this useful comment. We agree it is true that LMs should accommodate extrapolation more effectively than ANNs, and that a quantitative description of model performance is warranted.

**Edits:** We have added the following text, accordingly: "The fact that LMs performed better than ANNs in this case (with an r2 of .95 and RMSE of 0.35 ppb for LMs, as opposed to an r2 of .9 and an RMSE of 5.1 ppb for ANNs) may have to do with the general expectation that LMs be more resilient to extrapolation than ANNs."

**Comment:** P11 – Section 3.2.2: If the calibration is immediately after deployment, I am not surprised that there wasn't much of an effect. Do you anticipate there should be a significant time effect on such short time scales? What is the lifespan of the U-Pods?
**Response:** Thank you for the useful comment.
**Edits:** We have added the following text accordingly: "Gas sensor manufactures don't clearly define sensor lifetimes, but sensors are generally expected to loose sensitivity over time. For example, Alphasense CO-B4 electrochemical sensors are expected to have 50% of their original sensitivity after two years (Alphasense, 2015). The heater resistance in a give metal oxide type sensor is expected to drift over time, influencing sensor measurements (e2v Technologies Ltd., 2007). Masson and colleagues observed a significant drift in a metal oxide sensor heater resistance over the course of a 250 day sampling period in a laboratory setting (Masson et al., 2015). While we did not measure and record metal oxide sensor heater resistance for sensors included in U-Pods, we have investigated eltCO2 and e2vO3 sensor signal drift from the summer of 2015 through the summer of 2017. These data are presented in Fig. S26. Systematic downward drift in all eltCO2 sensor signals is apparent over this time frame. A clear and consistent pattern of systematic drift over this time period is less apparent for e2vO3 sensors. Since the training data was collected immediately after, the test data period, and since the test data period was relatively short (approximately one month) sensor drift could be negligible across the combined training/testing time frame."

**Comment:** P11 – L25: I am confused by the introduction of discussion around figCxHy – should we expect this sensor to play an important role?
**Response:** Thank very much to the reviewer for helping us to clarify.
**Edits:** We have added the following text: "Again the model for $O_3$ that was found to perform best in our previous (Casey et al., 2017), an ANN with temp, absHum and all metal oxide sensor signals as inputs, performed the best at sites included in this case study, with one exception. At the Sub Station site the inclusion of the figCxHy sensor signal decreased model performance. Additionally, the performance of all models tested at the Sub Station site during the SJ Basin Spring 2015 deployment was significantly worse in terms of MBE than model performance at other sites, both LMs and ANNs with different sets of inputs. Since this sensor signal input augmented model performance at the same sampling location during the summer deployment period, this finding could be attributable to the extrapolation with respect to temperature that occurred during the test period of this case study. As discussed in the introduction, metal oxide sensor sensitivity to different gas species can vary along with sensor surface temperature. Models were trained to use the figCxHy sensor signal, across the ambient temperatures in encompassed by the training data, to help account for the influence of confounding gas species at the BAO site. We think it is possible that the different temperatures in combination with the unique mix of gas species present at the Sub Station site, which the figCxHy sensors are highly sensitive to, caused the ANN to perform worse."

**Comment:** P12 – L5: How long is "so long"? This is related to my comment on
**Response:** Thank you for helping us to be more specific.
**Edits:** We have changed "so long" to "several months".

P11 – Section 3.2.2.

**Comment:** P12 – L21-24: I am confused – did you switch to humidity measured by Picarro or omit humidity entirely? It is not clear to me what happened here.
**Response:** Thank you for pointing out this confusion.
**Edits:** We have more clearly described the humidity replacement process (if any) for each individual case study dataset pair in the methods section. We have additionally added to the text in section 2.3 as follows: "Water mole fractions measured by the Picarro were converted into mass-based mixing

ratios to match the units of the absolute humidity signal in the U-Pod data. We applied an adjustment to this absolute humidity signal so that it matched observations in U-Pods during the following month when good RH sensor data was available, to account for the fact that temperatures were higher in U-Pod enclosures than the ambient environment. We then replaced the relative humidity signal in each U-Pod from August 23rd through October 1st in 2016 with the mixing ratios derived from Picarro measurements. Using the temperature and pressure logged in each U-Pod along with the absolute humidity from the Picarro, relative humidity was calculated for each U-Pod during this period."

And to section 3.2.4: "Interestingly, when humidity this 'borrowed humidity signal was not included as an input, the model performance markedly increased and became competitive with other 'same location' test deployment case studies."

**Comment:** P13 – L13: I don't really understand what is meant by "relative circumstances" – could you be more explicit about each of the case studies? Perhaps a table that outlines case study, with a one sentence description, and a column describing limitations would be more appropriate (and should be introduced at the beginning of the paper).
**Response:** Thank you for the feedback.
**Edits:** We have changed "relative circumstances" to "relative timing and parameter coverage". We have also adapted Table 4 according to this feedback and described the 'relative circumstances' present in each case study much more thoroughly in the methods section.

**Comment:** P13 – L18: What is meant by "extrapolated significantly?" Can you be specific?
**Response:** Sure, thank you for the comment.
**Edits:** We have changed "extrapolated significantly" to "extrapolated more than several months".

**Comment:** Table 1: I find Table 1 almost impossible to follow. It is not very clear which sensor measures which pollutant, as the input codes are frequently indecipherable. I am honestly not sure what I am supposed to get out of this table.
**Response:** Thank you for this useful feedback that will help us improve the quality and clarity of Table 1.
**Edits:** We have added a row at the top of the table indicating the target gases for each sensor. We have added to the Table caption an explanation of the input codes for the sensors: "Gas sensors included in U-Pods along with the model input codes we assigned each. The input code for each gas sensor is simply an abbreviation for the make of the sensor, followed by the target gas species(s)."

**Comment:** Table 3: I find it difficult to interpret this table. What do the black diamonds mean? What do you mean by "relative circumstances"??
**Response:** Thank you for the helpful comment.
**Edits:** We have updated Table 4 by replacing the first column so it is more clearly an indication of which case studies covered which target gases ($O_3$ and $CO_2$, or just $O_3$). We have also updated the caption of Table 4 to be more descriptive and informative, and less confusing: "Relative timing and parameter coverage between model training and test deployment dataset pairs. Incomplete coverage of time occurred if training only took place before or after the test data period and not before and after (pre and post). Incomplete coverage of location occurred when training took place in one location and testing took place in another. Incomplete coverage of parameters, including the target gas mole fraction, temperature, time, and pressure occurred when the values observed during training did not encompass the values observed during testing."

**Comment:** Figure 8: There is way too much going on in this figure, it is almost difficult to look at. Is there a more streamlined way of presenting the findings that is less complicated? I feel there is valuable information in the Figure, but it's hard to determine what that is, due to information overload. Ditto for

Figure 9, though it isn't as bad. Also you would do well to remind the reader what each variable represents.

**Response:** Thank you very much for the feedback and helping us to simplify and clarify figures.

**Edits:** We have split what was previously Figure 8 into two figures (now Figure 10 and 11) in order to simplify the graphics and highlight the content of each. We have also added definitions for the sensor inputs in the Figure captions for what were Figures 8 and 9 (now Figures 10, 11, and 12).

TECHNICAL CORRECTIONS

**Comment:** P1 – L18: "in time than to…" vs. in time that to

**Response:** Thanks very much for catching this typo.

**Edits:** We have changed 'that' to 'than' accordingly.

**Comment:** P2 – L18: change informal language "hold up" to something more scientific

**Response:** Thank you for the helpful feedback.

**Edits:** We have replaced 'hold up' with 'remain effective'.

**Comment:** P2 – L20: Delete word "Specifically"

**Response:** Thank you for the helpful edit.

**Edits:** We have deleted the word "Specifically"

**Comment:** P7 – L8: Rephrase "…showed successfully reduced over fitting"

**Response:** Thank you for the helpful comment.

**Edits:** We have replaced "…showed successfully reduced over fitting" with proved to be effective in the reduction of over fitting"

**Comment:** Figure 1: Enhance figure caption to explicitly state that blue is training, pink is testing

**Response:** Thank you, we will do this.

**Edits:** "Model training periods for each test deployment are shown in blue, and model test periods are shown in magenta."

---

## Author Comment (AC2) · 4 Sep 2018

Dear Reviewer 2,

We would like to offer our sincere thanks for spending your time in the review of our work and helping us to significantly improve the quality and clarity of the manuscript with your very detailed comments and suggestions. Each of your comments is listed below in black text, followed by our response and edits in blue text.

**Review 2 Comments:**

**Overview:** Casey and Hannigan explore the spatial and temporal transferability of field calibration models (specifically linear models (LMs) and artificial neural networks (ANNs)) for two sensors, O3 (e2vO3) and CO2 (eltCO2), reported by the integrated U-POD sensor package. By 'spatial/temporal transferability' they mean a determination as to whether a calibration model trained from sensor colocation (with reference instrumentation measuring target species) at one location works effectively when that same sensor system is then deployed at a different location. As the authors point out, changing the micro-environment (and local air pollution source contributions to that unique environment) may pose additional complications/challenges when trying to reconcile quantitative measurements with low-cost sensors. The authors make some attempt to separately describe temporal and spatial extension to better understand whether time-alone undermines the accuracy of the calibration models or change of location.

While the topic of sensor calibration and extension of calibration models across a diverse set of deployment scenarios is of fundamental importance to the field of low-cost AQ sensing, the paper, as written, largely fails to pull together a coherent narrative from which active participants in the low-cost AQ measurement space could easily glean useful, actionable information. To be clear, the topic of sensor quantification is inherently complex, and the authors undertake an ambitious analysis spanning 3 years of data from 10 U-POD systems deployed across 4 micro-environments. There are important lessons to be learned from their efforts, but at present these lessons are not brought to the fore of the paper and as a result are easily lost to the reader.
**Response:** Owing to Reviewer comments and through careful reconstruction, we think the updated version of the paper does a much better job of pulling together a coherent narrative that will be useful for others in the field of low-cost AQ sensing, in terms of useful, actionable information. We hope the Reviewer and the editor will find that important elements of the paper and take-away lessons are now brought to the fore of the paper so that readers can more easily note and make use of our findings.
**Edits:** We have clarified and added significant detail to the methods section. We have added more context, explanation, and discussion of specific findings in the results and discussions section, and we have explicitly highlighted a number of take away points and recommendations connected to specific findings in the conclusions, as well as highlighted these points in the abstract.

**Comment:** Throughout the manuscript the authors refer back to their published work (Casey et al., 2017). In the vast majority of instances in which this reference is provided, there is little to no contextual detail explicitly drawing the lines of connectivity between the current work and the previous work. Seeking out the exact evidence that exists in the earlier work and relating its relevance to the current work is left entirely up to the reader. Overall, this referencing needs to be done in a manner that is not vague and does not require that the reader be intimately familiar with the previous work.
**Response:** Thanks very much to the Reviewer for this helpful comment highlighting confusion between the contributions and citations of our previous work and the unique contributions of the current work.
**Edits:** In each instance that we have cited our previous work, we have added text to provide contextual detail explicitly drawing the lines of connectivity between the current work and the previous work.

**Comment:** The paper would also be strengthened if the unique and novel insights that result from the current work were more clearly differentiated from the Casey et al., 2017 effort.

**Response:** Thanks very much to the Reviewer for this very helpful comment and suggestion.

**Edits:** We have added text to the conclusions section explicitly summarizing unique and novel insights that result form the current work so that current findings are clearly differentiated form our current work.

**Comment:** There are seemingly contradictory statements throughout the text. These tend to originate from the authors' desire to provide a clear-cut answer as to whether or not a given model 'worked' in a given case study under a given environmental sampling condition. The fact of the matter is, low-cost AQ sensor quantification is extremely convoluted and often times the validity of data can be somewhat ambiguous. Faced with this level of complexity, the current manuscript fails to provide a succinct and systematic evaluation/reporting approach, and as such main (and important) take-home lessons from their work are lost.

**Response:** Thanks very much to the Reviewer for this helpful feedback. Many of the specific comments below have helped us to clarify what previously appeared to be contradictory statements throughout the text.

**Edits:** Throughout the manuscript, we have made edits to clarify what could have been perceived as contradictory statements. We have added significant detail to better match the level of complexity in the findings we present and have attempted to more systematically and succinctly report these findings and associated take away messages for readers.

Specific comments:

**Abstract.** We assessed the performance of ambient ozone ($O_3$) and carbon dioxide ($CO_2$) sensor field calibration techniques when they were generated using data from one location and then applied to data collected at a new location. We also explored the sensitivity of these methods to the timing of field calibrations relative to deployments they are applied to.

10   Employing data from a number of field deployments in Colorado and New Mexico that spanned several years, we tested and compared the performance of field-calibrated sensors using both linear models (LMs) and artificial neural networks (ANNs) for regression. Sampling sites covered urban, rural/peri-urban, and oil and gas production influenced environments. Generally, we found that the best performing model inputs and model type depended on circumstances associated with individual case studies. In agreement with findings from our previous study that was focused on data from a single location

15   (Casey et al., 2017), ANNs remained more effective than LMs for a number of these case studies but there were some exceptions. In almost all cases the best $CO_2$ models were ANNs that only included the NDIR $CO_2$ sensor along with temperature and humidity. The performance of $O_3$ models tended to be more sensitive to deployment location than to extrapolation in time while the performance of $CO_2$ models tended to be more sensitive to extrapolation in time that to deployment location. The performance of O3 ANN models benefited from the inclusion of several secondary metal oxide

20   type sensors as inputs in many cases.

**Comment:** L9. Avoid ending sentence with 'to'

**Response:** Thank you for catching this mistake.

**Edits:** We changed the sentence structure accordingly. "We also explored the sensitivity of these methods in response to the timing of field calibrations relative to deployments periods."

**Comment:** L13: this is one of the core conclusions: the resilience of a given calibration model depends on the circumstances of the deployment for that same sensor system. As such, the paper would be strengthened if the authors focused the narrative on succinctly describing such dependences and circumstances relating these factors back to the sensitivity, selectivity, and stability of each sensor system and sensor type.

**Response:** Thanks very much to the Reviewer for this helpful comment.

**Edits:** Throughout the paper, we have added text to help focus the narrative in the context of relative circumstances present in individual case studies, and how model performance in each case study relates to sensor sensitivity, selectivity, and stability.

**Comment:** This language is far too vague, especially for an abstract. What circumstances?

**Response:** Thank you for noting how we can make the abstract more informative and less vague.

**Edits:** We have added descriptions of specific circumstances: "We found that the best performing model inputs and model type depended on circumstances associated with individual case studies, such as differing characteristics of local dominant emissions sources, relative timing of model training and application, and the extent of extrapolation outside of parameter space encompassed by model training. "

**Comment:** L15: 'a number' - again, this is too vague. Define exactly how many of the case studies were characterized has having superior AAN models and how many were just as well served with an LM model

**Response:** Thanks to the Reviewer for the helpful comment.

**Edits:** We have added the following detail to the abstract: "Among models that were tailored to cases studies on an individual basis, $O_3$ ANNs performed better than $O_3$ LMs in 6 out of 7 case studies, while $CO_2$ ANNs performed better than $CO_2$ LMs in 3 out of 5 case studies."

**Comment:** L16: This line suggests that people should model CO2 with ANNs not LMs. The more detailed discussion in the body of the paper contradicts this assertion.

**Response:** Thanks to the Reviewer for the helpful comment.

**Edits:** After further consideration, we determined that this statement was an oversimplification and so have removed it from the abstract.

**Comment:** L19: subscript O3

**Response:** Thanks to the Reviewer for catching this.

**Edits:** O3 subscripted: $O_3$

5    **1.1    Low-Cost Sensors For Air Quality Measurements**

The use of low-cost metal oxide, electrochemical and non-dispersive infrared sensors to characterize air quality is becoming increasingly common across the globe (Clements et al., 2017; Kumar et al., 2015). Field normalization methods to quantify atmospheric trace gases using low-cost sensors have been found to be more effective than lab calibration when sensors are deployed in the ambient environment, and subject to changing temperature and humidity (Cross et al., 2017; Piedrahita et al.,

10   2014; Sun et al., 2016). Our previous study and several others have compared the efficacy of LMs (simple and multiple linear regression) relative to supervised learning methods, all finding that ANNs (Casey et al., 2017; Spinelle et al., 2015, 2017) and random forests (Zimmerman et al., 2017) outperformed LMs in the ambient field calibration of low-cost sensors. These effective supervised learning techniques often incorporate multiple gas sensor signals as inputs in order to quantify each target gas, in addition to environmental variable sensor signals, with the goal of compensating for the effects of

15   interfering gas species and environmental factors. In practice though, the reason multiple gas sensors are able to improve the performance of supervised learning type regression, and linear models for that matter, may be in part the result of correlation, or correlation for some time periods, between mole fractions of target gases themselves that hold for one model training location, but might not hold up at alternative sampling sites or during other time periods.

**Comment:** L11: What is the difference between supervised learning methods and ANNs? This warrants a more detailed description / definition.

**Response:** Thanks to the Reviewer for the helpful comment.

**Edits:** We have added the following text to help clarify that ANNs are an example of a supervised learning method, as are random forests: "supervised learning methods (including ANNs and random forests)"

**Comment:** L15: This sentence (bracketed in red) - is very important, but also very wordy and hard to follow.

**Response:** Thanks to the Reviewer for the helpful comment.

**Edits:**  We have simplified and clarified the sentence accordingly:  "In practice though, the reason multiple gas sensors are able to improve the performance of calibration models may be in part the result of correlation between mole fractions of target gases themselves that hold for one model training location, but might not remain effective at alternative sampling sites or during other time periods."

**Comment:**  Related to this assertion, it is not clear how the authors disentangle the temporal and spatial domain from one another, particularly the temporal domain. Time-decay patterns in the data are going to be present whether or not the sensor system has been moved to a different location. How would one ascribe difference in that case to a spatial domain and not temporal domain?
**Response:**  In this work, we attempt to disentangle the temporal and spatial domain by including some case studies where models were only extended outside of their training spatial domain or only extended outside of their temporal domain, but not both.  We attempt to represent time-decay patterns effectively in models by using pre/post training data for some case studies.
**Edits:**  We have clarified this strategy by more clearly defining what is meant by 'extrapolation in time', and by more clearly identifying which case studies were subject to extrapolation in time.  We have added the following text to section 2.3 accordingly:  "A model was extrapolated in time when ever training data does not take place both before and after a given test deployment period.  In several case studies we present, model training only took place after the test deployment period, comprising a 'post only' calibration.  In Colorado, and more broadly in the western United States, ambient temperatures change significantly across the seasons throughout the year, so if a model is extrapolated in time, extrapolation in temperature often results as well."

**Comment:**  'hold up' this language is too casual and used throughout the text. Consider re-wording.
**Response:**  Thank you for the feedback.
**Edits:**  We have replaced 'hold up' with 'remain effective'.

Section 1.2

**Comment:**  L28: 'A number of enclosures..' define the number.
**Response:**  Thank you for the feedback and helping us to clarify how many U-Pods were included in each case study and in our previous work.
**Edits:**  In section 1.3 (previously section 1.2) we have added the following text: "The study tested and compared calibration models using data from two U-Pod sensor systems".  We have also updated the text in the methods section and Fig. 2 (previously Fig.1) to explicitly list the number of U-Pods included in each case study.

**Comment:**  If Casey et al., 2017 demonstrated the ANN results for CO2 and O3 in the Spring of 2017 in Greeley, CO; is that same data being presented as a portion of this paper (as Figure 1 suggests).
**Response:**  Yes, it is the same data.  Thank you for pointing out that this needs clarity.
**Edits:**  We have clarified this, adding sub subsections within section 2.2 describing each case study individually, including our previous study in section 2.2.7:  "We include findings from our previous work as a case study in order to provide context.  Models for $CO_2$ and $O_3$ were tested using data from two U-Pods collected over the course of approximately one month at the GRET site in the spring of 2017.  Data from two U-Pods during approximately month-long periods pre and post of the test period were used to train $O_3$ and $CO_2$ models.  This case study provides another example of model performance when training took place both pre and post of the test period, and testing took place in the same location as training."

**Comment:**  The concluding sentences of this section nicely frame the motivation/need for the current work, consider bringing this to the fore of the paper / abstract, etc.
**Response:** Thanks to the Reviewer for the helpful comment.
**Edits:**  We have augmented the abstract with this piece of motivation by adding the following text: "This was motivated by a previous study (Casey et al., 2017) which highlighted the importance of

determining the extent to which field calibration regression models could be aided by relationships among atmospheric trace gases at a given training location, which may not hold if a model is applied to data collected in a new location. We also explored the sensitivity of these methods in response to the timing of field calibrations relative to deployments periods."

We have also augmented the for of the paper by added the following text to the end of the first paragraph in the introduction: "ANNs, as powerful pattern recognition tools, were found to perform better than both inverted and direct LMs in our previous study, but concerns arose when findings suggested that the performance of ANNs was being augmented by the relationships among gas mole fractions in the atmosphere at a given location. Low-cost gas sensor systems have the potential to inform spatial and temporal variability of pollution, when calibration equations for each sensor system are generated in one location based on co-located measurements with reference instruments, then moving the sensor systems into a spatially distributed network. Since the relationships among gas mole fractions at different sampling sites across a spatially distributed network, calibration models may not hold at new sampling sites. In this work, we test calibration model performance when extended to new locations."

Section 1.3

**Comment:** Final sentence: It's unclear why, if all of the U-POD sensor systems were equipped to measure CO and CH4 alongside CO2 and O3, analogous training/test matrix pairs are unavailable for these other species.
**Response:** Thanks to the Reviewer for the helpful comment.
**Edits:** We have clarified that while the sensors for CO and CH4 were included in the U-Pods during all the presented case studies, reference measurements for these species were not available: "In previous work (Casey et al., 2017) we have additionally addressed the quantification of CO and $CH_4$ using arrays of low-cost sensors together with field normalization methods, but these species are not included in the present analysis because reference data for model training and testing deployment pairs, diverging in location and timing and analogous to those we present for $O_3$ and $CO_2$, were not available CO and $CH_4$."

Section 1.4

**Comment:** L20: 'Very high levels of ozone' – specify the actual concentration or concentration range
**Response:** Thanks to the Reviewer for the helpful comment.
**Edits:** We have replaced 'Very high levels of ozone' with 'Mole fractions of ozone in as high as 140 ppb and 117 ppb during winter months have also been observed and attributed directly to oil and gas production emissions in the Upper Green River Basin of Wyoming and Utah's Uinta Basin, respectively"

**Comment:** L23: 'a modeling study' – is there really only one modeling study that shows this?
**Response:** Thanks for this comment. This is the only modeling study we know of that was focused on the effects of oil and gas production emissions on ozone, with potentially high spatial near emissions sources.

**Comment:** Final sentence: 'pooling' avoid using words with common association different from the intended meaning. Consider re-wording. 'accumulating'?
**Response:** Thanks to the Reviewer for the helpful comment.
**Edits:** We replaced 'pooling' with accumulation'

Section 2.1

**Comment:** L5: "with a number of low-cost gas sensors" – specify the actual number of sensors integrated in each U-POD. The authors identify that 10 U-POD systems were used in the previous and current work, but the vast majority of case studies (outlined in Figure 1.) utilize just 2 U-PODs at each

location. The authors need to more clearly describe in the text how the U-PODs were distributed throughout the work and whether all 10 U-PODs used in the current work had the same characteristic O3 and CO2 response when measuring the same air.

**Response:** Thanks to the Reviewer for the helpful comment.
**Edits:** We have replaced added subsections to section 2.2 describing each case study, including how many U-Pods were included in each:
"2.2.1 **Dawson Summer 2014**
The first distributed measurement campaign took place during the summer of 2014 when five U-Pods were sited at locations around Boulder County, with four distributed along the eastern boundary of the county, adjacent to Weld County where dense oil and gas production activities were underway.  A background site, further from oil and gas production activities was also included to the west, near a busy traffic intersection on the north end of the City of Boulder. Co-locations with reference measurements that were used for field calibration of sensors took place at the Continuous Ambient Monitoring Program (CAMP) Colorado Department of Health and Environment (CDPHE) air quality monitoring site in downtown Denver. One of the distributed sampling sites, Dawson School, was also equipped with an $O_3$ reference instrument (a Thermo Electron 49) ,operated by Detlev Helmig's research group from the Institute for Artic and Alpine Research (INSTAAR). In this work, a case study is developed using data from one U-Pod located at the CAMP site in downtown Denver for $O_3$ model training, and data from one U-Pod, located at the Dawson School for $O_3$ model testing. This case study is used to test model performance when extrapolated in terms of $O_3$ mole fractions and applied in a new location, transferred from an urban to a peri-urban environment.
**2.2.2 SJ Basin Spring 2015**
In the spring of 2015 we augmented our original fleet of five U-Pods (BA, BB, BD, BE, and BF) with five more (BC, BG, BH, BI, and BJ) and deployed these sensor systems in the SJ Basin while a targeted field campaign was underway to understand more about a $CH_4$ 'hot spot' that was discovered from satellite based remote sensing measurements (Frankenberg et al., 2016; Kort et al., 2014).  The primary goal of this sensor deployment was to inform spatial and temporal patterns in atmospheric trace gases like $CH_4$, $O_3$, CO, and $CO_2$ across the SJ Basin.  Most U-Pods were located at existing air quality monitoring sites operated by the New Mexico Air Quality Bureau (NM AQB), the Southern Ute Indian Tribe Air Quality Program (SUIT AQP), and the Navajo Environmental Protection Agency (NEPA), which supported validation of sensor measurements for $O_3$ After this deployment period, all U-Pods were moved to the BAO site in the DJ Basin for approximately one month, and were co-located with reference instruments there that were operated by National Oceanic and Atmospheric Administration (NOAA) researchers.  A case study is developed with data from the BAO site to train $O_3$ models for four U-Pods, and data from SJ Basin sites to test $O_3$ models for four U-Pods.  This case study is used to test model performance when extrapolated in temperature and time, and extended to a new location, extended from one oil and gas production basin to another across Colorado
**2.2.3 SJ Basin Summer 2015**
In the summer of 2015, after an approximately month-long co-location with reference instruments at the BAO site, seven U-Pods were deployed again at existing regulatory monitoring sites for approximately one month, after which they were moved back to the BAO site for another month of co-location with reference instruments there.  We equipped two of the regulatory monitoring sites in the SJ Basin with LI-COR LI-840A $CO_2$ analysers to provide reference measurements for $CO_2$.  A case study is developed with data from the BAO site, pre and post of the SJ Basin summer 2015 deployment to train models, and data from SJ Basin sites during the summer deployment period, to test models.  Data from seven U-Pods were used to train and test $O_3$ models and data from two U-Pods were used to train and test $CO_2$ models.  This case study is used to test model performance when training took place both pre and post of the test period, and when extended to a new location, from one oil and gas production basin to another across Colorado
**2.2.4 BAO Summer 2015**
During the SJ Basin Summer 2015 deployment period, two U-Pods remained at the BAO site. A case study is developed using data from two U-Pods the BAO site, pre and post of the summer 2015 deployment to train models for $O_3$ and $CO_2$, and data from two U-Pods the BAO site during the summer deployment period to test models for $O_3$ and $CO_2$.  This case study is used to test model

[revised manuscript text omitted]

**Comment:** The sensor system age (time since manufacture date) and environmental-hysteresis (lifetime environmental exposure of a given UPOD system) is not mentioned anywhere in the text. Do these factors not matter when analyzing the temporal extension of a given calibration model? When considering the fundamental measurement principles of these particular gas sensors, does degradation occur due to gradual (or rapid) deposition of material onto active catalytic sites within the sensors? If so, then the age of a given sensor and what's it's been exposed to over its lifetime, ought to factor in.. or at least deserve a mention.

**Response:** We agree with the importance and relevance of sensor challenges highlighted in this comment.

**Edits:** We have added the following text in section 2.2 accordingly: "Making quantitative measurements of atmospheric trace gases with low-cost sensors is challenged by unique variations in individual sensor responses associated with variations in the manufacturing process, sensor age, and sensor exposure history. For these reasons, we generated unique calibration models using data from sensors in each individual U-Pod sensor system. The closest available data prior and or subsequent to a test data period was used for model training to avoid complications associated with significant sensor drift and degradation in sensor sensitivity to target gas species over time if possible."

We have additionally added the following text has been added to section 3.2.2: "Gas sensor manufactures don't clearly define sensor lifetimes, but sensors are generally expected to loose sensitivity over time. For example, Alphasense CO-B4 electrochemical sensors are expected to have 50% of their original sensitivity after two years (Alphasense, 2015). The heater resistance in a give metal oxide type sensor is expected to drift over time, influencing sensor measurements (e2v Technologies Ltd., 2007). Masson and colleagues observed a significant drift in a metal oxide sensor heater resistance over the course of a 250 day sampling period in a laboratory setting (Masson et al., 2015). While we did not measure and record metal oxide sensor heater resistance for sensors included in U-Pods, we have investigated eltCO2 and e2vO3 sensor signal drift from the summer of

2015 through the summer of 2017.  These data are presented in Fig. S26.  Systematic downward drift in all eltCO2 sensor signals is apparent over this time frame.  A clear and consistent pattern of systematic drift over this time period is less apparent for e2vO3 sensors.  Since the training data was collected immediately after, the test data period, and since the test data period was relatively short (approximately one month) sensor drift could be negligible across the combined training/testing time frame. "

**Comment:**  The explanation of the training vs test sampling periods is confusing as written. Given the nature of the experiment, doesn't each UPOD system have to be co-located with reference instrumentation for the full duration of the period of study? It sounds as though the authors aimed to bookend the distributed network measurements ('testing period' with a period of colocation at a reference site in the general vicinity of the deployment ('training period') – but in order evaluate their models, they would have to retain a co-located reference measurement of O3 and CO2 at all times in all locations.

**Response:**  Yes, we present data, opportunistically, from test periods when sensor systems were co-located with $O_3$ and $CO_2$ reference instruments.

**Edits:**  We have added text to section 2.2 to try to clarify these details:   "Five to ten U-Pods were deployed at sampling sites in and around the DJ and SJ Basins over the course of several years, from 2014 - 2017. Deployments generally consisted of co-location with reference measurements prior to and following approximately one-month periods of spatially distributed measurements.  During some of the distributed measurement periods, a subset of U-Pods remained co-located with reference instruments where the field calibrations took place.  As well, during some distributed measurement periods, some U-Pods were deployed in new locations that were equipped with reference measurements.  In between periods of distributed sensor system deployments, sensor systems were co-located with reference instruments for as long as possible, as logistics, and coordination with other regulatory agencies and researchers would allow.  In this way, we hoped to maximize our ability to encompass full ranges of temperature, humidity, and trace gases that occur across seasons, in order to minimize extrapolation with respect to these parameters when models were applied to measurements from distributed deployment periods.  The locations where all or a subset of U-Pods were co-located with reference instruments are indicated in Fig. 1.  In this exploratory study, we opportunistically employ data from these sensor deployments, treating them as case studies in order to characterize the performance of field calibration models when they are extended to new locations. For each case study, described below, data was divided into training and test periods.  Timelines for these dataset pairs detailed in Fig. 2.  Some U-Pods used included in these case studies (indicated in grey font in Fig. 2) were constructed, populated with sensors, and deployed at field sites in the spring of 2014, approximately a year before the rest of the U-Pods were constructed, populated with sensors, and deployed at field sites in the spring of 2015.  The relative age of sensor systems included in some case study comparisons could have contributed to some discrepancy in model performance, though systematic differences based on U-Pod age is not apparent.

As available data from each case study allowed, we used approximately one month of training data before and after (pre and post of) a given approximately month-long test period. When training data was not available within several months of a test period, significantly longer training datasets were used in order to attempt capture and effectively represent trends in sensor drift over time, as well as to avoid extrapolation of model parameters (particularly temperature) during the test data period. As a result, model-training durations varied across case studies and sometimes significantly exceeded model-testing durations.  Each case study is similar in representing approximately one month-long deployment of sensor systems.  This study design serves a primary goal of this work, which is to help support the quantification atmospheric trace gases from low-cost gas sensor data in new locations, relative to model training locations, for periods of approximately one month at a time.

Making quantitative measurements of atmospheric trace gases with low-cost sensors is challenged by unique variations in individual sensor responses associated with variations in the manufacturing process, sensor age, and sensor exposure history.  For these reasons, we generated unique calibration models using data from sensors in each individual U-Pod sensor system.  The closest

available data prior and or subsequent to a test data period was used for model training to avoid complications associated with significant sensor drift and degradation in sensor sensitivity to target gas species over time if possible. Table 2 lists the $O_3$ and $CO_2$ reference instruments that were co-located with U-Pods at each sampling site, along with instrument operators, calibration procedures, and reference data time resolution. The selected case studies, described in sections 2.2.1 through 2.2.7 below are aimed at supporting methods to quantify atmospheric trace gases during the distributed deployments we carried out from 2014 through 2017 as well as future distributed sensor network measurements. Fig. 1 shows sampling site locations in context with urban areas and oil and gas production wells. Fig. 2 shows the timeline of each of these deployments, highlighting the training and testing periods defined for both $O_3$ and $CO_2$."

**Comment:** Looking at the deployment timelines displayed in Figure 1, it is also evident from the Figure (but not from the text) that the vast majority (~75% or greater) of the total deployment time was used to train the nodes not test the resultant calibration models (~25% of the total time). These train-to-test ratios appear to undermine the general applicability of the models to longer duration, distributed sensor measurements in which no co-located reference measurements are available. The authors should make an effort to bridge the gap between how they were able to execute their experiments and how distributed low-cost AQ sensor systems will ultimately be deployed.
**Response:** Thank you for helping us clarify why varying and sometime long durations of training data were used for each case study, and how we hope this study design can help support future sensor measurement efforts.
**Edits:** We have added the following text to section 2.2 toward this end:
"In between periods of distributed sensor system deployments, sensor systems were co-located with reference instruments for as long as possible, as logistics, and coordination with other regulatory agencies and researchers would allow. In this way, we hoped to maximize our ability to encompass full ranges of ambient temperature, humidity, and trace gases that occur across seasons, in order to minimize extrapolation with respect to these parameters when models were applied to measurements from distributed deployment periods."

"In an effort to fully encompass the parameter space present and during each individual test deployment case study, as well as sensor drift over time, model-training durations varied across case studies and sometimes significantly exceeded model-testing durations. Each case study is similar in representing approximately one month-long deployment of sensor systems. This study design serves a primary goal of this work, which is to help support the quantification atmospheric trace gases from low-cost gas sensor data in new locations, relative to model training locations, for periods of approximately one month at a time."

From section 2.2.5: "A significantly longer training duration is implemented in this case study because the training period took place more than several months after the model testing period. We reasoned that a longer training duration would be better able to represent patterns in sensor drift over time, as well as encompass the temperature range of test dataset period. Significantly less training time is needed when training occurs directly pre and/or post of a given model application period.

Highlighted in passage above:

**Comment:** L10: define the number of sampling sites. Eliminate vague language in the text. L15: same comment.
**Response:** The number of sampling sites during each case study varied, so to help clarify we directly reference the map showing each of the sampling sites included in the study.
**Edits:** We have renamed this 'Figure 1' and renamed the timeline 'Figure 2' accordingly.

**Comment:** L21: 5 UPOD systems are purportedly used in the Boulder / CAMP 2014 work. Figure 1 lists 1 UPOD system as being active during that test. Reconcile this.

**Response:** Thank you for helping us clarify that while 5 U-Pods were deployed during the Boulder County study, only one of the U-Pods was deployed at a location that had co-located reference measurements for O₃.
**Edits:** The number of U-Pods used in the Dawson Summer 2014 case study and others has been clearly updated in sections 2.2.1 – 2.2.7.

**Comment:** L27: Identify the actual ref. O3 measurement in the text here
**Response:** Thank you, we agree indicating the specific instrument used would be useful to the reader.
**Edits:** "Thermo Electron 49"

**Comment:** Last sentence: is this relevant to the current paper/study? Not clear what 'study' the authors are referring to in this sentence.
**Response:** We agree this sentence lacked specific relevance to the current study.
**Edits:** We have removed this sentence.

[revised manuscript text omitted]

**Comment:** L9: The authors claim that the SJ Basin network was similarly executed for the DJ Basin.
DJ Basin is absent from Figure 1., replaced presumably by BAO. It is unclear how many UPODs were
deployed to the DJ Basin. It's very confusing trying to track in time and location the distribution of
the 10 UPODs. If I try and decipher the information in Figure 1, either 2 or 4 UPOD units were
deployed to the DJ Basin, which on the face of it, does not constitute a similar network deployment of
10x UPODs deployed to the SJ Basin (although, it seems that only 4 and/or 7 UPOD units were
deployed to the SJ Basin.
**Response:** Thanks to the Reviewer for the helpful comment.
**Edits:** We have added the to the caption of Figure 2 (previously Figure 1) to help clarify that all
sampling sites outside of the SJ Basin group were in the DJ Basin. "The Dawson, BAO, and GRET
sampling sites are all located in the DJ Basin."

**Comment:** L13: The authors identify the BAO site as the relevant co-location site for the DJ Basin-
deployed UPODS, but then point out that there were NO co-located reference instrumentation
accessible for any of the distributed sampling sites. What does this mean for evaluating / testing their
models in the distributed network application?
**Response:** Thanks to the Reviewer for the helpful comment.
**Edits:** We have made significant edits in section 2.2, more clearly defining which data is included in
this work and in each case study. All discussion about the distributed deployment sites that did not
have reference measurements has been removed from the text, since these deployments helped to
motivate, but are not directly relevant to the present work.

**Comment:** L14-16: The authors state the GRET site housed all 10x UPOD systems for a year, but Figure 1 indicates that only 2-6? UPOD systems were used at this location and only for shorter periods of time. Again, the text is extremely hard to follow and the information in Figure 1 does not make it any clearer.

**Response:** Thanks very much to the Reviewer for pointing out the confusing nature of how the information is presented.

**Edits:** We have updated Figure 2 (previously Figure 1) to help clarify which U-Pods were included in each case study.

| (a) Case Study | Year | Jan | Feb | Mar | Apr | May | Jun | Jul | Aug | Sep | Oct | Nov | Dec |
|---|---|---|---|---|---|---|---|---|---|---|---|---|---|
| Dawson Summer 2014 | 2014 | | | | | | | | | | | | |
| SJ Basin Spring 2015 | 2015 | | | | | | | | | | | | |
| SJ Basin Summer 2015 | 2015 | | | | | | | | | | | | |
| BAO Summer 2015 | 2015 | | | | | | | | | | | | |
| BAO Summer 2016 | 2016 2017 | | | | | | | | | | | | |
| GRET Fall 2016 | 2016 2017 | | | | | | | | | | | | |
| GRET Spring 2017 | 2017 | | | | | | | | | | | | |

| (b) Case Study | Training Location | Test Location | O$_3$ # U-Pods | O$_3$ U-Pod Names | CO$_2$ # U-Pods | CO$_2$ U-Pod Names |
|---|---|---|---|---|---|---|
| Dawson Summer 2014 | CAMP | Dawson | 1 | BE | NA | NA |
| SJ Basin Spring 2015 | BAO | SJ Basin | 4 | BB, BD, BF, BJ | NA | NA |
| SJ Basin Summer 2015 | BAO | SJ Basin | 7 | BA, BB, BD, BE, BF, BH, BI | 2 | BB, BD |
| BAO Summer 2015 | BAO | BAO | 2 | BC, BJ | 2 | BC, BJ |
| BAO Summer 2016 | GRET | BAO | 2 | BH, BI | 2 | BH, BI |
| GRET Fall 2016 | GRET | GRET | 2 | BH, BI | 2 | BH, BI |
| GRET Spring 2017 | GRET | GRET | 2 | BH, BI | 2 | BF, BI |

**Figure 2**: (a) ANN and LM training and test deployment timelines. The Dawson, BAO, and GRET sampling sites are all located in the DJ Basin. Model training periods for each test deployment are shown in blue, and model test periods are shown in magenta. For the BAO Summer 2016 case study, the period outlined in blue shows data that was used to train O$_3$ model, but not CO$_2$ models since CO$_2$ reference data was not available during winter months. (b) Information about each of the case studies presented in the above timelines, including model training and testing locations, as well as the number and names of U-Pods included in each case study for both O$_3$ and CO$_2$ models. The U-Pods with names shown in grey were constructed and deployed starting in May of 2014. The U-Pods with names shown in black were constructed and deployed starting in April of 2015.

**Comment:** L26: The only metal oxide sensor that's relevant to the current work is the e2vO3 sensor. The operational fundamentals of this sensor should be described: the raw signal processing, circuitry considerations, and known theoretical operational conditions that undermine the sensitivity, selectivity, and/or stability of the e2vO3 metal oxide sensor.

**Response:** Thanks to the Reviewer for the helpful comment.

**Edits:** Since models in this work included signals from multiple gas sensors, we have added a discussion of the operating principles of metal oxide, electrochemical, NDIR the sensors accordingly,

as well ad discuss these sensor properties in context with model development in section 1.1, and 1.2. Additionally, we discuss these sensor considerations in context with unique challenges associated with measurements in oil and gas production basins in section 1.5:

[revised manuscript text omitted]

**Comment:** L29: 'in a few' Quantify the number of UPODs with faulty RH sensors
**Response:** Thanks to the Reviewer for the helpful comment.
**Edits:** We have replaced 'in a few' with 'in four'.

**Comment:** L31: 'nearby': Define the exact position relative to the faulty UPOD
**Response:** Thanks to the Reviewer for the helpful comment.
**Edits:** We have updated this passage to include specific information about the relative positions of U-Pods when faulty humidity signals were replaced: "The closest U-Pod with good humidity sensors ranged from several feet, when U-Pods were co-located during deployments in the DJ Basin, to approximately fifty miles during deployments in the San Juan Basin."

When the U-Pods were initially deployed at the GRET site, on August 23rd of 2016, the RH sensors in all ten U-Pods malfunctioned, logging an error code of -99 instead of the relative humidity. This malfunction seemed to coincide with the implementation of radio communication from each U-Pod to a central node in an effort to reduce trips to the field site to download data and to identify issues with data acquisition promptly. RH signals in the U-Pods began logging correctly again

5    in November when we stopped remote communication. We replaced RH values for the U-Pods during this time period by utilizing data from the Picarro Cavity Ring-Down Spectrometer that was co-located at GRET with the U-Pods. Water mole fractions measured by the Picarro were converted into mass-based mixing ratios to match the units of the absolute humidity signal in the U-Pod data. We then replaced the absolute humidity signal in each U-Pod from August 23rd through October 1st in 2016 with the mixing ratios derived from Picarro measurements. Using the temperature and pressure logged in each U-

10   Pod along with the absolute humidity from the Picarro, relative humidity was calculated for each U-Pod during this period.

To perform regressions toward field calibration of sensors, the reference and U-Pod data needed to be aligned. When reference measurements with minute time resolution were available for both training and corresponding testing periods, minute median data from the U-Pods were used. Medians were used as opposed to averages in order to reduce the potential

15   influence of sensor noise as well as to remove short duration spikes in the reference and sensor data that resulted from air masses that may not have been well mixed across the reference instrument inlets and the U-Pod enclosures. When reference data were instead available with only 5-minute or 60-minute time resolution, U-Pod medians were calculated for the same time step. Medians were also calculated for reference measurements with finer time resolution to match the time resolution of corresponding training/testing data. The first 15 minutes of data after any period that the U-Pods had not recorded data for

20   the previous 5 minutes was removed in order to filter transient behavior associated with sensor warm-up.

**Comment:** Did the implementation of radio communication for the UPODs have any impact on any of the other measurements in the system, beyond RH?
**Response:** We have added the following text to help address this question for the Reviewer and other readers.
**Edits:** "No other impacts to sensor systems were observed in connection with radio communications."

**Comment:** At the beginning of the paragraph, the authors state that the radio communication was active until November, but the substitute RH values from the Picarro were only applied up to October 1 (later part of the paragraph). This is confusing.
**Response:** Thank you for catching this conflict.
**Edits:** We have corrected it by changing "November" to "October" in the first instance.

**Comment:** Generally speaking, faulty or absent RH measurements on-board the UPOD (or any low-cost AQ sensor system that suffers from environmental interference) is a potentially widespread issue across the emerging field. I think the authors missed an opportunity to discuss their work-around in more detail and comment on the importance of maintaining stable RH measurements within any given low-cost AQ sensor system.
**Response:** Thanks to the reviewer for this helpful comment. We added to the text to help clarify the work around that we implemented for faulty RH sensor data. Over the course of multiple field deployments of U-Pod sensor systems, including those described in this work, RHT03 sensors signals were found to drift down over time, and "bottom out" in some cases. Following this observation, we have since upgraded to Sensirion AG SHT25 sensors which appear to be more robust and consistent over the course of long-term field deployments. Hopefully this information will be as helpful to readers as the more through discussion of the work around we have added.
**Edits:** We have added the following text accordingly:
"Over the course of multiple field deployments, relative humidity sensors in four of the U-Pods drifted down, causing the lower humidity levels to be cut off or 'bottomed out'. RH sensors were not replaced during field deployments in order to preserve consistency across different deployment

periods, allowing for the possibility of a single comprehensive model to apply to all data from a single U-Pod. After some experimentation in generating a 'master model' that could be applied to data from a given U-Pod for all collected field measurements, across several years, we determined that individual models for each deployment would be more effective, and replacing RH sensors that had drifted down would have been appropriate in support of the methods presented here. We have since upgraded to Sensirion AG SHT25 sensors, which appear to be more robust and consistent over the course of long-term field deployments."

**Comment:** The completely unusable radio communication RH values and the drifting RH values mentioned in section 2.3 beg the question – do the authors think this is a failure on the RHT component itself or the circuitry of the UPODs. Again, if the evidence suggests the former, that is useful empirical data for others in the field.
**Response:** Thanks to the Reviewer for bringing up this important question. We have not yet determined whether the failure of the RHT sensor signals during periods of active radio communications were connected to the sensors themselves or to the circuitry of the UPODs. This will be important to determine for the sensor community and before we try to implement radio communications again. As indicated in the previous comment, the drift of RHT03 sensors over time appeared to be an issue associated with the sensor model itself.

**Comment:** Where is RH measured specifically within each UPOD. Is the measurement internal to the box or positioned in a manner to provide a true ambient RH measurement? What are the implications of using alternative RH data sources that are not on-board the same UPOD?
**Response:** RH is measured within each U-Pod enclosure, in the microenvironment where the gas sensors are located. Using an alternative source for RH data that are not onboard and individual U-Pod has the potential to increase uncertainty of quantified gas mole fractions.
**Edits:** We have added the following text accordingly: "Temperature and RH sensor measurements are usually collected from within each U-Pod sensor system, in order to gain representative information about the environment the gas sensors are being operated in. Using an alternative source for RH data that are not onboard and individual U-Pod has the potential to increase uncertainty of quantified gas mole fractions. We used replacement RH data from the closest available U-Pod instead of ambient measurements in order to more closely match operating temperature within a U-Pod enclosure."

**Comment:** If median values were used for the co-located reference instruments, but the data from those instruments was 1-min averages, how did the authors obtain reference measurement medians at 1- min (the vast majority of temporal resolution used in the current work).
**Response:** Thank you for pointing out that this passage was confusing. We have changed the text to help clarify.
**Edits:** "In order to test models using the same time resolution they were trained with, the time resolution of reference and sensor measurements for corresponding training/testing datasets were matched, if necessary, by taking medians of the dataset with higher time resolution to match the data with the longer time resolution."

**Comment:** L19: What % of the total data used in training/testing each UPOD was removed due to this 5-min null data condition?
**Response:** We agree this information is useful for readers.
**Edits:** Accordingly, we have added the following text: "During a given deployment, the data removed to avoid sensor warm-up transients constituted less than 1%."

Section 2.4

**Comment:** L32 'using methods described previously', given the importance of the LMs and ANNs in the current work, each model should be described in more detail in the manuscript.
**Response:** Thanks to the Reviewer for the feedback. We are happy to provide more detail.

**Edits:** Two useful figures from our previous paper, showing ANN architecture, have been added and cited (now Figures 2 and 3) to help clarify. The following text has also been added:

". As in (Casey et al., 2017), direct LMs and ANNs were trained with a number of different sensor input sets to map those inputs to target gas mole fractions measured by reference instruments. Direct LMs implemented were multiple linear regression models given by

$$r = p_1 + p_2 s_1 + p_3 s_2 + \ldots + p_n s_{n-1} \tag{1}$$

where $r$ is the target gas mole fraction (measured by a reference instrument) $s_1 - s_{n-1}$ are sensor signals from U-Pods that are included as model predictor variables, and $p_1 - p_n$ are corresponding predictor coefficients.

ANNs designed for regression tasks, like those employed in this work, generally consist of artificial neuron nodes that are connected with weights. Weights are initiated with randomly assigned values. An optimization algorithm is then employed to map a given set input values to corresponding target values. An example of a very simple feed forward neural network, and how weights are propagated through it are depicted in Fig. 3.

[Figure]

**Figure 3. Example of a simple feed forward neural network, showing how inputs are propagated through the network during each of the training iterations (Casey et al., 2017)**

In this work, ANNs were designed by assigning U-Pod sensor signals to artificial neurons in an input layer and assigning target gas mole fractions for an individual gas species, measured by a reference instrument to a single output neuron. Nonlinear, tansig, artificial neurons in one hidden layer for $O_3$ or two hidden layers for $CO_2$ (accordance with our earlier findings for each target gas species (Casey et al., 2017)) were then added between input layer and the network output neuron. Additionally, bias neurons, each assigned a value of 1, were connected to neurons in the hidden layer(s) so that individual connecting weights could be activated or deactivated during the optimization process. The number of neurons in each hidden layer was set equal to the number of inputs included in a given ANN. Fig. 4 shows a diagram of an ANN architecture employed in this work, when there were five inputs.

[Figure]

**Figure 4. Diagram of an example ANN with the same color-coded components as are presented in Figure SM3 in section 2.2 of the SM. This ANN has 5 inputs, 1 hidden layer with 5 tansig hidden neurons, and 1 linear output layer leading to 1 output. The network is fully connected with weights and biases (Casey et al., 2017).**

For ANN training we employed the Levenberg Marquardt optimization algorithm with Bayesian Regularization (Hagan et al., 1997). The Levenberg-Marquardt algorithm combines the Gauss-Newton and Gradient Decent methods, towards incremental minimization of a cost function (the summed squared error between the ANN output and target values as a function of all of the weights in the network). Training begins according to the Gauss-Newton method, in which the Hessian matrix (the second order Taylor series representation of the error surface) is approximated as a function of the Jacobian matrix and its transpose, significantly reducing required training time. Network weights are adjusted accordingly each training step to reduce error. If the cost function is not reduced in a given training step, an algorithm parameter is adjusted so that optimization more closely approximates the gradient decent method (a first order Taylor series representation of the cost function), providing a guarantee of convergence on a cost function minimum. Since local minima may exist across the error surface, it is important to train the same network multiple times (with different randomly assigned starting weights), in order to access the stability of ANN performance. In this work each ANN was trained 5 times."

**Comment:** P7L6 – need reference for Bayesian Regularization
**Response:** We agree this would be useful for readers.
**Edits:** Test added: "In the implementation of Bayesian Regularization, a term is added to the sum of squared error cost function as a penalty for increased network complexity in order to guard against over fitting. A two level Bayesian inference framework is employed, operating on the assumptions the noise in the training data is independent, normally distributed, and also that all of the weights in the ANN are small, normally distributed, and unbiased (Hagan et al., 1997)."

**Comment:** The concepts of early stopping, hidden neurons, and hidden layers need to be described
**Response:** Thanks for this useful comment. Hidden neurons and hidden layers have been depicted in diagrams and described in more detail, embedded in the new text describing ANNs in general cited two comments above.
**Edits:** We have added some text to describe the concept of early stopping: "In preliminary ANN tests we found that over fitting occurred even when Bayesian Regularization was used, so we additionally implemented early stopping, which proved to be effective in the reduction of over fitting. To implement early stopping, a portion of training data is set aside as validation dataset, and during training, an ANN is applied to this validation data after each training step. Training continues so long as the error associated with the validation dataset is reduced. When the error associated with the validation dataset is no longer being reduced, training stops early. For ANNs, training datasets were divided in half on an alternating 24-hr basis, with half used for training and half used as validation data for early stopping."

**3    Results and Discussion**

To evaluate the performance of each of the ANN and LM models that were generated using training data then applied to test datasets, we used residuals, the coefficient of determination ($r^2$), root mean squared error (RMSE), mean bias error (MBE),

15    and centered root mean squared error (CRMSE). The CRMSE is an indicator of the distribution of errors about the mean, or the random component of the error. The MBE, alternatively, is an indicator of the systematic component of the error. The sum of the squares of the CRMSE and the MBE is equal to the square of the total error, the square root of which is defined by the RMSE.   As in our previous work (Casey et al., 2017), we compared performance of LMs and ANNs with a number of different sets of inputs for each train/test data pair. The $r^2$, RMSE, and MBE for each of these alternative models when

20    applied to test data are presented in the supplemental materials (SM) in Fig. S2 through Fig. S7, along with representative scatter plots and time series comparing the performance LMs and ANNs for a given set of inputs.  In Fig. S2 through Fig. S7, the best performing model inputs for each train/test data pair are shaded in purple.  The type of model that performed the best (ANN vs. LM) is indicated in the caption of each figure.  Presented below is an analysis and comparison of the best-performing model for each species as determined in our previous work, as well as performance metrics for the best

25    performing model associated with each new training/testing dataset pairs described in section 2.2.

**Comment:**  Highlighted sentence is confusing as written. How can there be multiple 'best' preforming models?

**Response:** Thanks to the Reviewer for the helpful comment.

**Edits:**  We have added section 2.5 entitled "Calibration Model Evaluation and Testing in order to help clarify:

"To evaluate the performance of each of the ANN and LM models that were generated using training data then applied to test datasets, we used residuals, the coefficient of determination ($r^2$), root mean squared error (RMSE), mean bias error (MBE), and centered root mean squared error (CRMSE).  The CRMSE is an indicator of the distribution of errors about the mean, or the random component of the error.  The MBE, alternatively, is an indicator of the systematic component of the error.  The sum of the squares of the CRMSE and the MBE is equal to the square of the total error, the square root of which is defined by the RMSE.

First, we generated and applied the best performing model, as determined in our previous work (presented in Table 3), to data from each new case study.  Each new case study was selected to challenge models in different ways in order to evaluate the resiliency of the findings from our previous study when challenged by different circumstances.  Next we tested LMs for CO2 and O3 that contained only the primary target gas sensor for each species, as well as temperature and absolute humidity as inputs.  Finally, we generated, applied, and evaluated the performance of a number of LMs and ANNs with different sets of inputs for each case study in order to see which specific model performed the best for each individual case study.  The $r^2$, RMSE, and MBE for each of these alternative models when applied to test data are presented in the supplemental materials (SM) in Fig. S2 through Fig. S7, along with representative scatter plots and time series comparing the performance LMs and ANNs for a given set of inputs.  In Fig. S2 through Fig. S7, the best performing model inputs for each train/test data pair are shaded in purple.  The type of model that performed the best (ANN vs. LM) is indicated in the caption of each figure.  We discuss both the performance of the previously determined best fitting model (generated using data from the GRET Spring 2017 case study) when applied and generated to data from new case studies, and the performance of models that were tuned to perform the best for each individual case study.  From these comparisons, we draw insight into circumstances that challenge model performance in terms of relative local emissions characteristics, location, and timing between model training and testing pairs.  Table 4 lists the relative timing and parameter coverage between model training and testing periods for dataset pairs, highlighting instances of incomplete coverage during training that led to model extrapolation during testing."

**Comment:** Does section 2.2 really succinctly describe each training/testing dataset pair? This is the first place in the text of the manuscript where the limited extent of co-location upon distributed field deployment is described and how the 10 UPODs are reconciled against such limitations.

**Response:** Thanks so much for pointing out that this is needed. Instead of describing why measurements were planned and carried out, we change the focus in section 2.2 to describe measurements and how they are used in this work.

**Edits:** We have added subsections to section 2.2, in which we describe each case study (training/testing dataset pair) in the context of the work presented here.

Section 3.1

**Comment:** For the purposes of the current study, if there is no co-location with reference, is it still a relevant data point? Can the authors effectively 'test' their model under these circumstances?

**Response:** Thanks to the reviewer for pointing out this confusion. We only have the ability to evaluate models when we have co-located reference instruments, and we only include data in this work that had co-located reference instruments.

**Edits:** We have added details in section 2.2 about how many U-Pods are included in each case study presented and which reference instruments were co-located with each.

**Comment:** This section P8L3 is also the first mention of reducing/oxidizing interfering gas species – this potential deserves a more detailed explanation in the context of the specific micro-environment source contributions

**Response:** Thanks for this important comment.

**Edits:** We have added a discussion of the operating principles of the sensors to section 2.1 accordingly, detailed in response to an earlier comment above.

**Comment:** The overall discussion of factors impacting differences between the two Basin deploymnets is fairly scattered. It would be more beneficial to the reader if the authors could draw more specific lines of connectivity between environmental or pollution source contributions and the robustness (or lack of robustness) in the model.

**Response:** Thanks very much to the Reviewer for this helpful comment.

**Edits:** We have improved the manuscript by describing differences between the gas basins in more detail, and in the context of sensor sensitivity and selectivity. Here is text from one place in the manuscript where we have made these improvements: "In this work, we present and compare models designed to address the unique challenges that come with using low-cost sensors, in the quantification of atmospheric trace gases of interest in oil and gas production basins, where ambient hydrocarbon mole fractions are potentially elevated, exerting uniquely cofounding influence on low-cost gas sensors. We investigate how well models can be transferred from one microenvironment to another, with different dominant local emissions source characteristics, and different relative abundance of oxidizing and reducing compounds. Microenvironments explored in this work include an oil and gas basin where both natural gas and heavier hydrocarbons are produced (the DJ Basin), and an oil and gas production basin where prominently natural gas is produced (the SJ Basin), with much smaller proportional emissions of heavier hydrocarbons, and in tern, lower atmospheric concentrations of alkanes. Within and bordering the DJ Basin, additional microenvironments include an urban location, with significant mobile sources emissions ($NO_X$, CO, and VOCs), and a peri-urban site with fewer mobile emissions and closer proximity to oil and gas production activities. We explore how robust model performance is when a model is trained in one microenvironment and transferred to another; challenged by different relative abundance of oxidizing and reducing gas species. Additionally we test how well models can represent and address sensor stability over time and the potential for drift."

The U-Pod $CO_2$ data presented in Fig. 3 and Fig. 4 were quantified using ANNs that were trained using data from the BAO Tower with the following inputs from each U-Pod: eltCO2, temp, and absHum. This set of model inputs were found to be the best ANN inputs that we highlighted in our previous study, using data from the GRET site in the spring of 2017 (Casey et al., 2017). Fig. 3 shows scatter plots of U-Pod $CO_2$ vs. reference $CO_2$ during the test data period for sensors located at

20 BAO as well as sensors that were located at distributed sampling sites throughout the SJ Basin. The scatter plots show that while there was generally a smaller dynamic range of $CO_2$ at the SJ Basin sites relative to BAO, model performance did not appear to be impacted or degraded by spatial extension to these locations in the SJ Basin. The line of best fit for Fort Lewis site (periwinkle) is even closer to the 1:1 than the lines of best fit for two U-Pods located at BAO (black and grey). Overlaid histograms of residuals in the bottom right corner of Fig. 3 show that $CO_2$ residuals from each of the SJ Basin U-Pods are

25 generally centered and evenly distributed about zero with similar spread.

U-Pod $CO_2$ average residuals from the same data presented in Fig. 3, quantified using ANNs with eltCO2, temp, and absHum signals as inputs, are plotted according to time of day and date in Fig. 4. While the use of ANNs in place of LMs was shown to reduce U-Pod $CO_2$ residuals significantly with respect to temperature, some daily periodicity in the residuals

30 for all four U-Pods is apparent in the upper plot in Fig. 4 that shows residuals by date. The lower plot in Fig. 4, showing residuals by time of day, demonstrates that $CO_2$ from all four U-Pods was generally under predicted during early hours of the morning and generally over predicted during afternoon and evening hours. Interestingly, this trend in residuals by time of day is more pronounced for the two U-Pods that remained at BAO. The majority of U-Pods stopped logging data, unfortunately, at one point or another during these deployments. The periods of missing data are reflected in the plots of

**Comment:** eltCO2, temp, absHum should be human readable, this is the first time these parameters appear in the text. I understand that they were listed in the table describing UPOD guts, but they should be spelled out here.
**Response:** Thank you for the feedback.
**Edits:** Descriptions added here and at the first mention of other model input codes in the manuscript in the text: "eltCO2 (ELT S300 CO2 sensor) , temp (temperature) , and absHum (absolute humidity)"

**Comment:** L18: it is unclear to what extent the current work and the previous work are duplicated here? Does the previous work form the basis for determining the optimal set of input parameters to train the ANN model and those same set of input parameters were found to be optimal again in this second application or are the actual applications overlapping and therefore the result is redundant? This is an example where I find the self-referential context to Casey et al., 2017 confusing (and lacking specific differentiating information).
**Response:** Thank you for the useful comment. We applied the model that we found to perform best in our previous work to new data. The application circumstances did not overlap and are not redundant.
**Edits:** We have added the following text to help clarify: "We began by testing the best-performing $CO_2$ model, as determined in our previous work (Casey et al., 2017), on this data, collected under a different set of circumstances, during the summer of 2015."

**Comment:** The under-prediction / over-prediction behavior of all four UPODs warrants more discussion. What environmental conditions are pushing the model beyond its limits? What is the fundamental (under-the-hood) reason for the interference in the first place (based on sensor fundamentals)?
**Response:** Thank you for this interesting comment. After analysis and careful consideration, we have added the following text:
**Edits:** "Upon examination of overlaid histograms showing distributions of parameters during model testing and training periods, in Fig. S12, and model time series and residuals plots in Fig. S3, there is no indication of model extrapolation at the BAO site, and no significant trends of concern with respect to residuals. Bias introduced to mole fraction estimates are likely attributable to differences in hydrocarbon mixtures in the SJ Basin relative to the DJ Basin."

**Comment:** Why did the majority of UPODs stop logging data during the deployment? Did the system overheat? What fraction of the total possible sample time was missed?

**Response:** Thank you for the comment and helping us to improve the details of the study.

**Edits:** The following text has been added accordingly: "Periods of missed data during the month-long deployment included approximately 1 day at the Shiprock site, 2 days at the Bloomfield site, 4 days at the Sub Station site, 9 days at the Fort Lewis site, and 17 days at the Navajo Dam site. We carried out frequent sampling site visits (on a weekly or biweekly basis as logistics and travel to remote locations in some cases allowed) in order to identify and fix problems as they arose during field deployments. Operational issues were predominantly attributable to power supply problems associated with BNC bulkhead fittings and corrupted micro SD cards."

Section 3.1 continued..

5    Differing from our previous findings, for this group of training and testing data pairs from the summer of 2015 at the BAO and SJ Basin sites, the inclusion of the e2vVOC and alphaCO signals noticeably improved the RMSE in the quantification of $CO_2$. While the inclusion of these two secondary sensor signals didn't result in the best performance in our previous study, using data from the GRET site (Casey et al., 2017), we found that their inclusion did not degrade performance relative to the models that included just eltCO2, temp, and absHum signals as inputs. Generally, using rh vs. absHum signals as ANN inputs did not seem to make a big difference in model performance, though linear models were sometimes found perform

10    better when the absHum signal is used instead of the rh signal. From Fig. S2, it is apparent that inputs including e2vCO2, temp, rh, e2vVOC, and alphaCO sensor signals as model inputs resulted in the lowest RMSE for U-Pods at BAO as well as at the two SJ Basin sites. Plots analogous to those presented in Fig. 3 and Fig. 4, but with this best performing set of inputs for the present data set pairs are presented in the SM, in Fig. S24 and Fig. S25 respectively.

15    $O_3$ was quantified for all the U-Pods deployed at BAO and SJ Basin sampling sites using an ANN with the following inputs: e2vO3, temp, absHum, e2vCO, e2vVOC, figCH4, and figCxHy. ANNs with this configuration were found to perform best in the quantification of $O_3$ in our previous study (Casey et al., 2017). These same inputs and model configuration were also found to be the best performing among others tested for SJ Basin 2015 dataset pairs as noted in Fig. S2. Interestingly though, LMs with this same set of inputs performed competitively well for a number of the U-Pods in the SJ Basin. For

20    three of seven U-Pods in the SJ Basin, LMs even outperformed ANNs in terms of RMSE. When the BAO trained U-Pods field calibrations for $O_3$ were extended to sites in the SJ Basin, we found that U-Pods at some of the sites performed better than others across all models that were tested, as seen in Fig. S2.

   Scatter plots and trends in residuals are presented in Fig. 5 and Fig. 6 for $O_3$. These plots show the performance of U-Pods at

25    BAO relative to those at SJ Basin sites in the quantification of $O_3$ during the test data period. U-Pod $O_3$ measurements at Fort Lewis, Navajo Lake, and the Sub Station did not agree with reference measurements as well as U-Pod $O_3$ measurements from the other four SJ Basin sites. U-Pods at Navajo Lake and Sub Station had bad humidity sensor data, as noted in section 6.2.3 and Table S1, so humidity from the U-Pod located at the Ignacio site was used in place of their humidity signals. Since the Ignacio site was located relatively far away from the Navajo Dam and Sub Station sites, this could have introduced some

30    additional error into the application of a calibration equation, particularly since we showed earlier that $O_3$ ANNs like the ones we employed here are very sensitive to humidity inputs (Casey et al., 2017). The Fort Lewis site had a different reference instrument than those used at other sites, which may have contributed to observed discrepancies. Fig. S1 shows that differences among U-Pod $O_3$ performance during the test deployment period were larger than those observed during the training phase among the same U-Pods; therefore, the incongruous field calibration performance phenomena we observed

Highlighted above:

**Comment:** L6-9: Discussion is confusing and language is too casual: "did not make a big difference" – too vague. Quantify based on the statistical analysis of the model test data. When considering the

benefit of including extra sensor inputs in the training matrix for their models, again the Authors are drawing comparisons to their earlier work (Casey et al, 2017) but it's not really clear how this improves/informs the current work – besides stating that the inclusion of the parameters didn't make the data product worse.

**Response:** Thank you for this useful comment. With this work, we are testing methods that we developed in our previous work under new circumstances that have the potential to challenge and degrade model performance. The finding we are highlighting in this instance is that in the current work, two additional sensor signals result in improved performance of a model under different circumstances, relative to our previous work. Since the addition of these two signals do not reduce the performance of models in our previous work, the addition of these two sensor signals in models for the quantification of $CO_2$ may be warranted more broadly.

**Edits:** We have changed 'did not make a big difference' to 'did not have a measurable affect'. Additionally we have added the following text: "so including these sensor signals may be appropriate as a general rule, in areas that are strongly influenced by oil and gas production activities."

**Comment:** L10 e2vCO2 does not exist as a sensor metric in the UPODs.
**Response:** Thanks so much to the Reviewer for catching this mistake. It should be e2vCO.
**Edits:** We have changed 'e2vCO2' to 'e2vCO'.

**Comment:** L15: 'all the UPODS' how many is this again?
**Response:** Thank for the clarifying comment.
**Edits:** The following edits have been made: "$O_3$ was quantified for the 2 U-Pods deployed at BAO and 7 of the U-Pods deployed at SJ Basin sampling sites"

**Comment:** L19: 'For a number of UPODs': state the number.
**Response:** Thanks to the reviewer for helping us clarify.
**Edits:** The text has been edited accordingly: "Interestingly though, LMs with this same set of inputs performed competitively well for 3 of the 7 U-Pods in the SJ Basin in terms of RMSE and $r^2$"

**Comment:** L21: 'for some of the sites..': which sites?
**Response:** Thanks to the Reviewer for helping us clarify.
**Edits:** The following edits have been made to the text: "When the BAO trained U-Pods field calibrations for $O_3$ were extended to sites in the SJ Basin, we found that U-Pods at the Bloomfield, Bondad, Shiprock and Ignacio sites performed better than others across all models that were tested, as seen in Fig. S2."

**Comment:** L15-22: this paragraph seems to say that the ANN training matrix determined to be optimal in Casey et al., 2017 was also found to be optimal in the current work, with inclusion of all peripheral sensors to the input training matrix for O3. But they also state that the LMs data products were just as good (or better) when compared to the ANN models. This result seems important, but not really discussed further. The results are left vague. Conclusions as to why this might be the case are absent.
**Response:** Thank you very much for helping us to make our conclusions more detailed and less vague.
**Edits:** We have added the following text accordingly: "The observation that LMs performed competitively well at a subset of SJ Basin sites is likely connected to the relative abundance of hydrocarbons and other potentially interfering oxidizing and reducing gas species at individual sampling sites, diverging from conditions present during model training at the BAO site. ANNs can better represent the influence of these interfering species than LMs during training, but appear to have lost their ability to do so for this subset of microenvironments in the SJ Basin."

**Comment:** L27: 'had bad RH data' – as noted in a section that doesn't exist. What is bad RH data? L29: 'relatively far away' – how far? Again. These details matter.

**Response:** Thank you for helping us clarify the text and catching the error regarding the section referenced.

**Edits:** We have made the following edits to the text: "As noted earlier, U-Pods at the Navajo Dam and Sub Station sites had faulty relative humidity sensor data, so humidity from the U-Pod located at the Ignacio site was used in place of their humidity signals. Since the Ignacio site was located approximately twenty-two and fifty miles away from the Navajo Dam and Sub Station sites respectively, this could have introduced some additional error into the application of a calibration equation, particularly since we showed earlier that $O_3$ ANNs like the ones we employed here are very sensitive to humidity inputs (Casey et al., 2017). Spatial variability in humidity across tens of miles could be significant as isolated storms (which are on average 15 miles in diameter) propagate throughout the region in the summer."

**Comment:** L30-31: Apparently one of the major results from Casey et al., 2017 is an extreme sensitivity to RH when using ANN's to quantify O3. Given the failure of the RH sensor throughout much of the work presented in the current work, it seems critically important that this RHsensitivity be discussed in much greater detail in the current work, not simply stated in an off—handed matter with a reference to the prior work.

**Response:** Thanks very much to the Reviewer for this helpful comment. We have added significant detail throughout the text describing humidity influences on sensors in the context of model development and testing.

**Edits:** Here is an example of some text we have added accordingly in section 3.2: "In our previous work, we showed that $O_3$ models were very sensitive to the humidity signal input (Casey et al., 2017). In this case study, it seems that replacing actual humidity signals with closely approximated humidity signals, negatively influenced model performance. In order to investigate this observation further, we tested the influence of replacing humidity data in the same manner, using mixing ratios from the same co-located Picarro, on test data from the GRET Spring 2017 case study. A comparison of model performance under normal and this 'borrowed RH' circumstance are presented in Fig. S27 in the SM. $O_3$ model performance was negatively impacted when 'borrowed' RH values based on Picarro data replaced U-Pod RH sensor signals. From these findings, it seems likely that the inclusion of multiple metal oxide type sensors as inputs in the model, which all respond strongly to humidity fluctuations, helped the ANN to effectively represent the influence of humidity in the system, more so than including a 'borrowed RH' signal from another instrument. We tested models with multiple gas sensor signals and no humidity signal as inputs for a number of other case studies as well (as seen in Fig. S2, Fig. S4, and Fig. S5), when good humidity data from U-Pod enclosures was available, but they did not turn out to be the best performing model in any of these other tests."

**Comment:** L32: 'had a different reference instrument' what was the instrument and why do the authors think that this particular reference instrument was in error, subsequently disrupting the validity of their calibration model?

**Response:** Thank you for the useful comment. We only want to acknowledge that discrepancies among different reference instruments that are operated according to different protocols and by different agencies are possible.

**Edits:** The following text has been added to help clarify: "At the Fort Lewis site, a 2b Technologies model 202 $O_3$ analyser was employed as a reference instrument, differing from the Thermo Scientific 49i, Thermo Scientific 49is, and Teledyne API T400 instruments utilized for reference measurements, elsewhere in the SJ Basin, and the Thermo Scientific 49c that was operated at the BAO site and used for model training. Of all the reference instruments, only the 2b Technologies model 202 $O_3$ at the Fort Lewis site was operated in a room that was not temperature controlled. Some bias may have been introduced to the Fort Lewis $O_3$ reference measurements as the temperature in the room it was housed in varied. Different instruments, operators, calibration and data quality checking procedures could have contributed to observed discrepancies. It is also possible that the microenvironment at each of these three sites contributed lower model performance."

**Comment:** L34 – carried into highlighted passage below: The authors indicate that the sampling sites or the circumstances discussed previously are the reason for the poor model performance, not the sensors comprising the UPODs. First, WHAT circumstances specifically, and what specifically about the sampling sites? This level of non-explanation is unacceptable.

**Response:** Thanks to the Reviewer for helping us clarify.

**Edits:** The following edits have been made to the text: "therefore, the incongruous field calibration performance phenomena we observed seems to be connected to unique characteristics associated with individual sampling sites; possibly the relative abundance of oxidizing and reducing molecules in the local atmosphere, which could interfere with sensor responses to their target gas species, as opposed to the quality of individual sensors in each of those U-Pods."

seems to be connected to the sampling sites or the circumstances discussed previously as opposed to the quality of individual sensors in each of those U-Pods.

5    All SJ Basin U-Pod $O_3$ measurements systematically over estimate lower levels of $O_3$ each night, a trend apparent in the scatter plots in Fig. 5 and in the residuals by time of day plot in Fig. 6. Upon examination of the scatter plots in Fig. 5, U-Pods at some sampling sites had positive bias for higher $O_3$ measurements as well (Shiprock, Ignacio, Sub Station, and Bloomfield), while for others, bias at the higher end of $O_3$ distributions did not appear to be present (Navajo Dam, Fort Lewis, and Bondad). The residuals by time of day plot in Fig. 6 shows that the two U-Pods at BAO did not have significant trends in their residuals according to the time of day, but that U-Pods deployed at SJ Basin sites consistently over estimated

10    nighttime $O_3$. The residuals are also plotted with respect to temperature in Fig. 6, where all U-Pods, even those at BAO to a lesser extent, appear to over predict $O_3$ at lower temperatures, which generally occurred at night. The times of day that generally correspond to the highest $O_3$ levels generally had the lowest residuals, with some exceptions at the Fort Lewis and Navajo Dam sites.

15    Fig. 6 includes a plot of the residuals across the duration of the deployment period, showing no significant sensor drift in measurements for any of the U-Pods. This plot also shows that the highest residuals observed generally occurred over short periods in time, particularly for the Fort Lewis (blue) and Sub Station (magenta) sites. In order to further explore the performance of field calibration models for $O_3$ at SJ Basin sites relative to BAO, the combined parameter space of temperature with $O_3$ reference mole fractions and temperature with absolute humidity are presented in Fig. 7. The combined

20    temperature and reference $O_3$ parameter space appears to be similar for all of the U-Pods, both at BAO and the SJ Basin sites. However, there appears to be some outlying combined temperature and humidity parameter space at the Sub Station site and at the Navajo Dam site. Brief excursions of high humidity may be connected to some of the large short-term residuals observed at these two sampling sites.

**Comment:** L22: brief excursions of high humidity – how brief? How high?

**Response:** Thank you for helping us improve clarity.

**Edits:** The following details have been added: "Brief excursions, lasting approximately 2 – 4 hours, of high humidity (up to 0.025 kg/kg, relative to the upper bound of absolute humidity observed at other sampling sites of 0.013 kg/kg) may be connected to some of the large short-term residuals observed at these two sampling sites."

**Comment:** Can the authors comment on the role that humidity transients play in fundamental sensor response? The description of the high and low bias resulting from the models at different locations and different times of day is difficult to follow. What are the common response characteristics and failings of the model that manifest across the case studies featured here? What are the lessons learned and how can these lessons better inform ANN model development moving forward?

**Response:** Thanks very much to the Reviewer for this helpful comment.

**Edits:** We have added the following details about fundamental sensor response in section 2.1, including the role that humidity transients play.

Section 3.2.1

**Comment:** Extrapolation of the ANN and LM models is problematic. Why? If the full-span of O3 (or CO2) concentration encountered in the field deployment is not covered in the training set for the model, is the model incapable of reasonably extrapolating?
**Response:** Yes, thank you for the helpful comment. The Dawson Summer 2014 case study suggested that, when a model is transferred to a new location, with different dominant local emission sources, both ANNs and LMs fail to extrapolate effectively with respect to high $O_3$ mole fractions.
**Edits:** We have added the following text has been added accordingly: "Across applications, ANNs have been found to be unreliable when extrapolated, due to the nonlinear nature and complexity of the relationships they represent. Though they are generally expected to be more robust to extrapolation that ANNs, increased uncertainty in measurements can also be introduced to LMs when parameters are extrapolated. In order to have high confidence in measurements of uncommonly high mole fractions of a target gas, the model -raining period has to encompass the full possible range. Combining both field calibration and lab calibration data together in a training dataset could accomplish this type of coverage. If extrapolation is expected to occur with respect to the target gas mole fraction, as in this case study, the use of an inverted LM may yield better results than LMs or ANNs. We describe inverted LMs and their potential advantages in our previous work (Casey et al., 2017)."

Section 3.2.2

**Comment:** L15: post-test deployment co-locations: It's unclear what is meant by 'post-test', please clarify.
**Response:** We are happy to clarify this concept.
**Edits:** The following text has been added to the end of section 2.3: "A model was extrapolated in time when ever training data does not take place both before and after a given test deployment period. In several case studies we present, model training only took place after the test deployment period, comprising a 'post only' calibration. In Colorado, and more broadly in the western United States, ambient temperatures change significantly across the seasons throughout the year, so if a model is extrapolated in time, extrapolation in temperature often results as well."

**Comment:** L16: state the # of UPODs
**Response:** Thank you for helping us to add clarity.
**Edits:** The following edits have been carried out: "We present data from four U-Pods that were co-located with reference instruments in the SJ Basin in the spring of 2015, at the Navajo Dam, Sub Station, and Bloomfield sites. Two U-Pods at the Bloomfield site provide a set of duplicate measures."

**Comment:** The concept of extrapolation in time is confusing. Please clarify what is meant by this? Generating a model at time X and then applying that same model to time X-Y?
**Response:** Thank you for helping us to clarify what is meant by extrapolation in time.
**Edits:** The edits have been made in the manuscript in section 2.3: "A model is extrapolated in time when ever training data does not take place both before and after a given test deployment period. In this case study, model training only took place after the test deployment period, comprising a 'post only' calibration. In Colorado, and more broadly in the western United States, ambient temperatures change significantly across the seasons throughout the year, so if a model is extrapolated in time, extrapolation in temperature often results as well."

**Comment:** The authors identify coal-fired power plants as an important near-field ('close-by') pollutant source that could contribute a specific (unique) pollutant signature that could render the utility of the Figaro sensor useless. Did the CO2 response of the UPODs or reference instruments or CO response of the sensor measurements indicate a near-field power plant plume across the deployment area?

**Response:** Thanks to the Reviewer for this useful comment.  We did observe evidence of a near-field power plant plume in the raw $CO_2$ and CO sensor signals as well as the NO and $NO_2$ reference measurements (the site was not equipped with a CO reference instrument).
**Edits:**  We have added the following text accordingly:  "Several-hour long enhancements or spikes are apparent in the raw eltCO2 and alphaCO sensor signals in the U-Pod deployed at the Sub Station site, indicating the presence of a near-by combustion-related emissions source.  Another indication of indicate a near-field power plant plume across the deployment area is apparent, in the form of several-hour long enhancements reference measurements of NO and $NO_2$ at the site."

Section 3.2.3

**Comment:**  How specifically was 'time' included as a raw input vector in the training matrix? Absolute time? Time since start of deployment? Time since calibration? Time since sensor manufacture?
**Response:** Thank you for the helpful comment.
**Edits:**  We have added the following text to the end of section 2.3 to help clarify, since the time model input is discussed there first:  "When time was included in a model as an input, the absolute time was used.  Specifically, we used the datenum value from the MATLAB environment, which is defined by the number of days that have elapsed since the start of January 1st, in the year 0000."

**Comment:**  L11-12: "...LMs outperformed ANNs with notable instability associated with the performance of ANNs when time was included as an input." In the previous sentence the authors stated that time was useful predictor of CO2.. but the last sentence appears to contradict this assertion. The fact that LMs outperformed ANNs for CO2 also contradicts general assertions made in the abstract.
**Response:** Thanks very much to the Reviewer for making these important points.
**Edits:**  We have added the following text to section 3.2.3 to help clarify:  "In the case of $CO_2$, LMs outperformed ANNs, which could be largely attributable to notable instability associated with the performance of ANNs when time was included as an input."

We have also added the following text to the abstract to help clarify:  "For $CO_2$ models, exceptions included, case studies in which training data used took place more than several months subsequent to the test data period.  For $O_3$ models, exceptions included studies in which the characteristics of dominant local emissions sources (oil and gas vs. urban) were significantly different at model training and testing locations."

**Comment:**  The authors should comment on the notion that time-sensitive response patterns in sensors indicates that some level of time-decay. Is this the case with the CO2 sensor and that's why time as a input parameter in the model makes such a big difference? Is there some fundamental reason why the ANNs would be poorly suited to model time-decay patterns in the sensors?
**Response:** Thanks very much to the reviewer for this suggestion as well as interesting and relevant questions.
**Edits:**  We have added the following text to address each:  "For $CO_2$, we expected the inclusion of time as an input to be a useful to model performance across this time frame, owing to observed trends of decreased $CO_2$ sensor sensitivity in time.  To keep the power requirements for the U-Pod sensor systems low, and to keep systems quiet, fans were used to exchange air in the enclosures as opposed to pumps.  As a result, the air entering the enclosures was not filtered, and sensors were exposed to some dust over time.  This dust exposure is likely largely responsible for observed decreases in $CO_2$ sensors sensitivity over time, shown in Fig. S26.  Decreases in infrared lamp intensity over time may also play a role.  In the case of $CO_2$ sensors, the implementation of pumps to draw new, filtered air into sensor enclosures could likely significantly reduce lose rates in the sensitivity of an individual sensor over periods of continuous deployment in ambient environment.  While we are not sure why ANN performance tended not to benefit from the addition of a time input, while LM performance did, it is likely attributable to the extrapolation of the time input, since only data that was collected significantly subsequent to the test data period was used for training.  ANNs are expected to be able

to better represent time decay trends if data from measurements both prior and subsequent to the test period are used in training, so that there is no extrapolation with respect to the time input."

Section 3.2.4

**Comment:** L23-24 – final sentence in this section is very important. Where the faulty RH (and necessity of substituting RH from alternate sources) degraded the models, if enough RH variability was captured with the suite of peripheral metal oxides sensors, the RH-interference could be effectively modeled without explicit RH inputs. It would seem important to emphasize this point a bit more prominently and discuss further – especially in the context of overcoming some of the RH-measurement shortfalls elsewhere in the manuscript through similar means.
**Response:** Thanks very much for this helpful comment.   We found this to be an interesting result also.
**Edits:**  We have added the following text accordingly:  "We tested models with multiple gas sensor singals and no humidity signal as inputs for a number of other case studies as well (as seen in Figures S2, S4, and S5), when good humidity data from U-Pod enclosures was available, but they did not turn out to be the best performing model in any of these other tests."

4. Conclusions

**Comment:**  Supervised learning techniques – generally, the manuscript lacks a description of what is meant by this -
**Response:** Thanks very much to the Reviewer for pointing this out.
**Edits:**  We have added the following text to the introduction and the conclusions to help clarify that ANNs are an example of a supervised learning method, as are random forests:  "We investigated how well a supervised learning technique (ANNs) hold up when sensors are moved to a new location, different from where calibration model training took place."

**Comment:**  L19-20 the concepts of temporal and spatial extension are still a bit confusing here. Earlier statements to clarify exactly what is meant by each condition would be helpful.
**Response:** Thanks to the Reviewer for pointing out this confusion.
**Edits:**  We have added the following text, early in the manuscript, at the end of section 1.4:  "In the present work, we test model performance under conditions of spatial extension, wherein a model is trained using data from one location then applied to a test dataset using data from a new location.  In testing spatial extension of a model we investigate how well the field calibration of low-cost sensors can inform target gas mole fractions when sensors are deployed in a new location and a new microenvironment of oxidizing and reducing compounds.  We also test model performance under conditions of temporal extension, wherein a model is trained using data that was collected only prior or subsequent to the model application period.  In testing temporal extension of models, we investigate how model performance is influence by sensor drift over time."

**Comment:**  L24: how does one move something in terms of its temporal coverage?
**Response:** Thanks to the Reviewer for pointing out that this statement is confusing and unclear.
**Edits:**  We have updated the text accordingly:  "While ANNs and other supervised learning techniques have been shown to consistently out perform linear models in previous studies when training and testing took place in the same location, we find that this trend does not always hold when field calibration models are applied in a new location, with significantly different local emissions source signatures for $O_3$ models, or when model training data takes more than several months subsequent to the model application period for $CO_2$ models."

**Comment:**  L1-3P16: LMs appear to be more robust when applied to a changing deployment condition – but then the authors hedge and say that they "… were not able to fully represent some of the complex nonlinear response behavior exhibited by the arrays of sensors." So a linear model can't model nonlinear behavior? The statement needs to be more specific.

**Response:** Thanks to the Reviewer for pointing out the vague nature of the statement. After some consideration, we realize that a more important point to make at the end of this paragraph has less to do with nonlinear response behavior, and more to do with extrapolation of observed ozone mole fraction.
**Edits:** We have updated the text accordingly: "While these LMs seemed to be more stable under circumstances of significant extrapolation in terms of local air chemistry and timing, we found that they did not extrapolate well in terms of the $O_3$ mole fraction, resulting in underproduction of $O_3$ values during the test period that exceeded those encompassed in the training data."

**Comment:** L7: "..data is almost a band running vertically in a range of CRMSEs." Data running in 'a band' doesn't aid in the interpretation of the data. Re-phrase to address the statistical product that results from the bias that was encountered.
**Response:** We agree re-phrasing this statement in terms of statistical attributes will help clarify.
**Edits:** The text has been updated accordingly: "As seen in Fig. 12, plot markers from all case studies have very similar CRMSE values, but plot markers from case studies in which models were tested in new locations have larger MBE values than models that were tested in the same location as they were trained. "

**Comment:** Final paragraph: how 'generalizable' are the models developed here? It would seem that despite having done an exhaustive amount of work, each individual UPOD system still required its own ANN or LM based on co-located data and raw sensor data from that individual sensor system. While the input matrix of raw sensor signals may be more generalizable, the models themselves appear to be very much node-specific, at least in so far as what has been shown in the paper.
**Response:** Thanks to the reviewer for highlighting this important point. We have added text to help address it.
**Edits:** Text added: "In order to account for unique variations in sensor responses, in each individual sensor system, due to variations in manufacturing along with elapses time and specific exposure subsequent to manufacturing, we present models that are generated for each sensor system on an individual basis. Future studies exploring the potential for universal calibration models would be very useful to the field."

**Comment:** It is unclear how the extension of the model frameworks discussed in the current paper can be used in the context of low-cost electrochemical sensors
**Response:** Thanks to the Reviewer for this very important comment. We have added five key take away points from this work and associated recommendations that we hope can be used by others in the field of low-cost gas sensors.

[revised manuscript text omitted]